# Lattice Models for Phases and Transitions with Non-Invertible Symmetries

Lakshya Bhardwaj[1], Lea E. Bottini[1], Sakura Schäfer-Nameki[1], Apoorv Tiwari[2]

[1] *Mathematical Institute, University of Oxford,*
*Andrew Wiles Building, Woodstock Road, Oxford, OX2 6GG, UK*

[2] *Niels Bohr International Academy, Niels Bohr Institute, University of Copenhagen,*
*Blegdamsvej 17, DK-2100, Copenhagen, Denmark*

Non-invertible categorical symmetries have emerged as a powerful tool to uncover new beyond-Landau phases of matter, both gapped and gapless, along with second order phase transitions between them. The general theory of such phases in (1+1)d has been studied using the Symmetry Topological Field Theory (SymTFT), also known as topological holography. This has unearthed the infrared (IR) structure of these phases and transitions. In this paper, we describe how the SymTFT information can be converted into an ultraviolet (UV) anyonic chain lattice model realizing in the IR limit these phases and transitions. In many cases, the Hilbert space of the anyonic chain is tensor product decomposable and the model can be realized as a quantum spin-chain Hamiltonian. We also describe operators acting on the lattice models that are charged under non-invertible symmetries and act as order parameters for the phases and transitions. In order to fully describe the action of non-invertible symmetries, it is crucial to understand the symmetry twisted sectors of the lattice models, which we describe in detail. Throughout the paper, we illustrate the general concepts using the symmetry category $\mathsf{Rep}(S_3)$ formed by representations of the permutation group $S_3$, but our procedure can be applied to any fusion category symmetry.

# 1 Introduction

A basic question when studying a quantum system (formulated on the lattice or in the continuum) pertains to the infrared (IR) phases realized— essentially the low energy phase diagram. Key questions arise concerning the organization, classification, and characterization of phases and transitions within the parameter space of a quantum system. Robust and universally applicable methods to address these questions are therefore of paramount importance.

Historically, global symmetries have served as the primary framework for organizing the understanding of these fundamental questions. Informed by Landau's seminal insights, much of the vast landscape of quantum phases and their universal properties found a cohesive description rooted in a symmetry-based understanding. Global symmetries facilitate the organization of states and operators into representations. Moreover, they constrain the kinds of IR phases/ground states that may arise, which in turn are classified by their symmetry breaking patterns and characterized by the ground state expectation values of charged operators or order parameters. Subtle symmetry features such as 't Hooft anomalies and symmetry fractionalization also impose strong constraints on the possible phases realizable in a quantum system. This is the content of the Landau paradigm of classification of phases of quantum matter.

The recent years, sparked by the work [1], have seen the development of a hugely generalized understanding of global symmetries. The central insight relates to identifying that topological operators/defects serve as global symmetries in quantum systems. This has led to the study of non-invertible or categorical symmetries [2–15] (for recent reviews with more complete references on this topic see [16, 17]). The natural mathematical framework to describe such symmetries, particularly in the context of finite symmetry structures, involves fusion (higher-)categories. Correspondingly, there has been a concerted effort to extend the Landau framework to incorporate such generalized categorical symmetries. This endeavor encompasses a systematic study of the generalized charges or representations [18–22], phases [3, 4, 23–31] and transitions [32, 33] for systems with categorical symmetries, which can be referred to as "categorical Landau paradigm" [28].

These studies have already revealed that such symmetries have physical implications that significantly go beyond their conventional group-like counterparts. For instance, their symmetry charges may involve combinations of local and extended operators [21, 29], and therefore their symmetry breaking patterns are distinctly new. Examples of generalized symmetry broken gapped phases in (2+1)d are topological orders, that are well outside the standard Landau paradigm [34, 35].

The exploration of the phase structure of systems with categorical symmetries have mostly been limited to the continuum, either in abstract conceptualizations or within continuum field theory frameworks. The main theoretical tool for these efforts is that of the SymTFT [36–39], also known as topological holography [40–45]. Let us briefly review the general framework which utilizes the SymTFT. Given a global categorical symmetry $\mathcal{S}$ and a spacetime dimension $d$, the SymTFT is a $(d+1)$-dimensional topological field theory $\mathfrak{Z}(\mathcal{S})$ with the property that any $d$-dimensional $\mathcal{S}$-symmetric quantum system can be recovered from its interval com-

pactification with a topological boundary condition, known as the symmetry boundary, on one end and a (generically non-topological) boundary condition on the other. The SymTFT is a fundamental structure, which encompasses all symmetry aspects of a quantum system. In particular, topological defects of the SymTFT label the generalized symmetry charges. For $d = 2$, phases associated to a symmetry are determined by condensable algebras in the category of topological defects associated to the SymTFT. Maximal condensable algebras (i.e. Lagrangian algebras) define topological boundary conditions of the SymTFT and classify gapped phases. Non-maximal condensable algebras correspond to gapless phases and transitions. The structure of condensable algebras, and thus phases, forms a partially ordered set, which can be arranged in a Hasse diagram [33].

**Lattice Models.** Quantum lattice models provide a concrete ultra-violet (UV) description for systems with interesting IR behavior. They illustrate general conceptual points, but also of course play a central role in the description of quantum matter: e.g. they encode complex emergent phenomena such as those found in frustrated magnets and strongly correlated electronic systems. They also are a promising avenue for providing toy models that serve to elucidate subtle phenomena such as topological order.

A natural question to ask is what kind of lattice systems realize non-invertible/categorical symmetries and how the different $\mathcal{S}$-symmetric phases and transitions are realized within such lattice systems (see [46–60] for recent works along these directions). A promising class of models in this regard are the anyon chain models [61–68] which can be defined using an input fusion category that determines the symmetry of the model. That being said, a systematic study of the phases and transitions as well as the organization of operators into representations of the categorical symmetry for anyon chain lattice models has not been carried out. In this paper, we describe how the SymTFT information can be converted into an ultraviolet (UV) anyonic chain lattice model which then lends itself to such a systematic analysis. Let us summarize the main aspects of this prescription.

**Anyon-chains from SymTFT data.** The input information entering the definition of this model involves:

1. **A symmetry boundary $\mathfrak{B}_{\mathcal{S}}^{\mathbf{sym}}$.** The fusion category of topological lines on this boundary is $\mathcal{S}$, which is the symmetry of the anyonic chain lattice model.

2. **An input boundary $\mathfrak{B}_{\mathcal{C}}^{\mathbf{inp}}$.** The fusion category of topological lines on the input boundary is $\mathcal{C}$. The two boundaries $\mathfrak{B}_{\mathcal{S}}^{\mathrm{sym}}$ and $\mathfrak{B}_{\mathcal{C}}^{\mathrm{inp}}$ are separated by an interface described by

a $\mathcal{C}$-module category $\mathcal{M}$.

3. **An object $\rho \in \mathcal{C}$.** This object is not necessarily simple and is arranged in a 1d lattice of length $L$ terminating on $\mathcal{M}$.

The model defined with these input data is a length $L$ anyon chain that lives on the interface defined by $\mathcal{M}$ between $\mathfrak{B}_{\mathcal{C}}^{\mathrm{inp}}$ and $\mathfrak{B}_{\mathcal{S}}^{\mathrm{sym}}$. In many cases, the lattice model thus defined admits a tensor decomposable Hilbert space and can therefore be translated into a familiar quantum spin model. This is an important point as typical microscopic condensed matter systems have such tensor product Hilbert spaces, even though such a condition is often relaxed dynamically at intermediate scales by energetic considerations.

**Twisted sectors and $\mathcal{S}$ symmetry action.** Within this construction of the anyonic lattice model, the twisted sector Hilbert spaces arises when a symmetry line living on $\mathfrak{B}_{\mathcal{S}}^{\mathrm{sym}}$ ends on $\mathcal{M}$. Based on general considerations in the SymTFT, various properties of these symmetry twist defects can be derived, such as how they can be transported, fused, split, associated etc. on the anyonic chain Hilbert space. This provides a constructive way to extract the complete $\mathcal{S}$ action on a concrete lattice model, including on all its symmetry twisted sectors.

**Generalized Charges for $\mathcal{S}$ on the Lattice.** In [19,21], it was shown that the symmetry multiplets or generalized charges for a non-invertible categorical symmetry $\mathcal{S}$ are labelled by topological defects in the corresponding SymTFT $\mathfrak{Z}(\mathcal{S})$. This understanding facilitates a concrete realization of all symmetry charges as realized in the anyon chain. The prescription is physically intuitive and can be summarized as follows. Consider the interface $\mathcal{M}$ between $\mathfrak{B}_{\mathcal{C}}^{\mathrm{inp}}$ to $\mathfrak{B}_{\mathcal{S}}^{\mathrm{sym}}$ in some configuration. This provides a state in the lattice model. Now pick any bulk line $\boldsymbol{Q}$ in the SymTFT and consider the configuration where the line has one end each on $\mathfrak{B}_{\mathcal{C}}^{\mathrm{inp}}$ and $\mathfrak{B}_{\mathcal{S}}^{\mathrm{sym}}$. Dragging the end on $\mathfrak{B}_{\mathcal{C}}^{\mathrm{inp}}$ through the interface $\mathcal{M}$ transforms the state. From these transformation properties we read off the action of the $\boldsymbol{Q}$-multiplet on the anyon lattice chain. Having concrete lattice expressions for such $\mathcal{S}$-charged operators is very desirable from both a theoretical and practical stand-point. From a theoretical point of view, it helps understand the representation theory of $\mathcal{S}$ on lattice models and consequently conceptually organize the phases and transitions in these models. From a practical point of view, these provide easy-to-use diagnostics for novel phases and transitions that can be numerically and potentially experimentally investigated.

**Gapped phases in the space of $\mathcal{S}$-symmetric lattice models.** The lattice models come equipped with the global symmetry $\mathcal{S}$ and a corresponding natural parameter space

of $\mathcal{S}$-symmetric Hamiltonians given by acting with arbitrary fusion graphs of the $\mathcal{C}$ lines on the lattice of $\rho$ lines on $\mathcal{M}$. Our approach systematically produces in particular commuting projector or fixed-point Hamiltonians. Such Hamiltonians are particularly convenient as they are easy to solve and yet capture all the universal properties of a given gapped phase. We leverage the understanding that gapped phases correspond to topological boundary conditions for the physical boundary in the SymTFT. Each such topological boundary condition can be obtained from any other reference topological boundary condition by an appropriate gauging on the physical boundary. Such a gauging is defined by a choice of Frobenius algebra in $\mathcal{C}$ which precisely enters the definition of the commuting projector Hamiltonian.

**Gapless phases and Transitions in the space of $\mathcal{S}$-symmetric lattice models.** It is natural to consider also Hamiltonians whose ground states realize gapless $\mathcal{S}$-symmetric phases. When these characterize second order phase transitions between two gapped $\mathcal{S}$-symmetric phases, the gapless models admit deformations to Hamiltonians whose ground states realize the corresponding gapped phases. Again, we use the continuum results for gapless phases to construct the associated Hamiltonians systematically from the SymTFT data, which is specified by a non-maximal condensable algebra of the SymTFT (the so-called club sandwich construction [32]) and which, together with an input phase transition, gives rise to a $\mathcal{S}$-symmetric gapless phase.

**Outlook.** An obvious question to consider is the extension to higher dimensions of the Landau paradigm for categorical symmetries. This can be done both in the continuum using the SymTFT, as well as the lattice. In dimensions higher than $(1+1)$d the phase structure of systems with categorical symmetries is expected to be significantly more complex. We hope to report on this interesting direction in the future.

**Outline of the paper.** The paper is organized as follows. In section 2 we describe the general setup of the anyonic chain models, describing their untwisted and symmetry twisted sectors, $\mathcal{S}$ symmetry action and multiplets of local operators charged under $\mathcal{S}$. In section 3, we detail how to obtain the different gapped phases realized in the lattice model, derive the symmetry action on the untwisted and twisted sector ground states, and describe the lattice realization of order parameters for these gapped phases. In section 4, we present a general approach to understand certain gapless phases and phase transitions realized in the lattice model that are obtainable via the club sandwich construction within the SymTFT. We describe the lattice realization of order parameters for the phase transitions. In each section, we present

a general theory followed by concrete examples with Abelian group symmetry and $\mathsf{Rep}(S_3)$ symmetry. $\mathsf{Rep}(S_3)$ is the non-invertible symmetry whose generators are the irreducible representations of the permutation group of 3 elements, with the composition given by tensor product of representations. The Abelian group symmetry examples serve to contextualize the general theory in a familiar and simple context, while the $\mathsf{Rep}(S_3)$ example is a simple non-invertible symmetry that is used to exemplify the general structure. An analysis of lattice models with $\mathsf{Rep}(S_3)$ symmetry in a Rydberg blockade ladder (on a constrained Hilbert space) has also appeared in the works [56, 69], where phases and transitions are discussed in terms of the $\mathsf{Rep}(S_3)$ symmetry. We also outline the analysis for the transition between gapped, SPT-phases for the non-invertible symmetry $\mathsf{Rep}(D_8)$ (which has as generators the irreducible representations of the dihedral group of order 8). The methods presented in this work can however be applied to any fusion category $\mathcal{S}$ symmetry.

**Note.** In our companion paper [70] we discuss a spin-chain defined on a tensor decomposable Hilbert space with alternating qubits and qutrits with generalized Ising-type Hamiltonians, which is $\mathsf{Rep}(S_3)$ symmetric and realize associated gapped phases and phase transitions between them. The model in [70] is motivated by (thought not the same as) the anyonic chain $\mathsf{Rep}(S_3)$ symmetric model discussed in this paper. However, many of the symmetry related aspects are analogous in both cases, and the present paper provides an extended and thorough treatment of the analysis reported in [70].

## 2 Anyon Chains with Fusion Category Symmetry

The goal of this work is to realize concrete (1+1)d lattice models acted upon by a categorical symmetry $\mathcal{S}$ that flow to gapped or gapless phases with symmetry $\mathcal{S}$, including the gapless phases serving as transitions between the gapped phases.

From general considerations, the symmetry restricts the phase structure in the IR, determines the (generalized) charges of order parameters for given phases, and provides insights into phase transitions. The key advantage of the anyon chain [61, 63, 65–68] is that it provides a spin model that has the action of the symmetry baked in from the get go. Perhaps a drawback is that it may seem less directly accessible (e.g. for numerics), but this is easily remedied by choosing specific basis of the Hilbert space (and representation of the Hamiltonian on this basis in terms of standard Heisenberg matrices).

## 2.1 General Construction

We discuss a class of lattice models in (1+1)d dubbed as **anyon chain models**. These models can carry any arbitrary fusion category symmetry $\mathcal{S}$. In this paper, we will study how these models are defined on a circle with periodic or twisted boundary conditions. In a later work, we will study how these models can be defined on a segment with various types of boundary conditions at the two ends.

**Input.** In order to define these models on a circle, we need the following input data:

- An input fusion category $\mathcal{C}$, which should in general be distinguished from the symmetry fusion category $\mathcal{S}$.

- A $\mathcal{C}$-module category $\mathcal{M}$. The symmetry $\mathcal{S}$ is determined in terms of $\mathcal{C}$ and $\mathcal{M}$ as

$$\mathcal{S} = \mathcal{C}^*_{\mathcal{M}} = \mathrm{Fun}_{\mathcal{C}}(\mathcal{M}, \mathcal{M}), \tag{2.1}$$

  which is the category formed by $\mathcal{C}$-module functors from $\mathcal{M}$ to $\mathcal{M}$. If $\mathcal{M}$ is an indecomposable module category, the symmetry $\mathcal{S}$ is a fusion category. On the other hand, if $\mathcal{M}$ is a decomposable module category, the symmetry $\mathcal{S}$ is a multi-fusion category[1].

- An object $\rho$ in $\mathcal{C}$, which in general is taken to be a non-simple object.

- A morphism $h: \rho \otimes \rho \to \rho \otimes \rho$.

**Untwisted Sector.** The basic constituent for the lattice model is a block of the following form

$$\tag{2.2}$$

where $m_i$ and $m_{i+1}$ are simple objects in the module category $\mathcal{M}$, and $\mu_{i+\frac{1}{2}} \in \mathrm{Hom}(m_i, \rho \otimes m_{i+1})$ is a basis vector in the morphism space formed by $\rho$ ending between $m_i$ and $m_{i+1}$.

---

[1] In this paper, we will mostly focus on the indecomposable case, but the decomposable case will also play a role in constructing lattice models for phase transitions between gapped phases with fusion category symmetries. In what follows here, we assume that $\mathcal{M}$ is an indecomposable module category and hence $\mathcal{S}$ is a fusion category.

Concatenating such blocks builds a basis vector in the Hilbert space of the model on a circle with periodic boundary conditions, referred to as the untwisted sector Hilbert space $\mathcal{V}_u$,

$$
\begin{array}{c}
\rho \qquad \rho \qquad\qquad \rho \qquad \rho \\
\\
m_1 \ \ \mu_{\frac{3}{2}} \ \ m_2 \ \ \mu_{\frac{5}{2}} \quad \cdots \quad \mu_{L-\frac{1}{2}} \ \ m_L \ \ \mu_{\frac{1}{2}} \ \ m_1
\end{array}
\tag{2.3}
$$

Here $L$ is the length of the system and we consider periodic boundary conditions, i.e. the two $m_1$ on the left and right are identified with each other.

The building block for the Hamiltonian is given by the move:

$$
\begin{array}{c}
\rho \qquad\qquad \rho \\
h \\
m_{i-1} \ \ \mu_{i-\frac{1}{2}} \ \ m_i \ \ \mu_{i+\frac{1}{2}} \ \ m_{i+1}
\end{array}
\ = \ {\sum}' \ \boldsymbol{h}^{\mu'_{i-\frac{1}{2}},m'_i,\mu'_{i+\frac{1}{2}}}_{\mu_{i-\frac{1}{2}},m_i,\mu_{i+\frac{1}{2}}}
\begin{array}{c}
\rho \qquad\qquad \rho \\
\\
m_{i-1} \ \ \mu'_{i-\frac{1}{2}} \ \ m'_i \ \ \mu'_{i+\frac{1}{2}} \ \ m_{i+1}
\end{array}
\tag{2.4}
$$

where we sum over simple objects and a basis of morphisms (labeled by primes) in

$$
{\sum}' : \quad
\begin{cases}
m'_i \in \mathcal{M} \\
\mu'_{i-\frac{1}{2}} \in \mathrm{Hom}(m_{i-1}, \rho \otimes m'_i) \\
\mu'_{i+\frac{1}{2}} \in \mathrm{Hom}(m'_i, \rho \otimes m_{i+1}) \,,
\end{cases}
\tag{2.5}
$$

and $\boldsymbol{h}^{\mu'_{i-\frac{1}{2}},m'_i,\mu'_{i+\frac{1}{2}}}_{\mu_{i-\frac{1}{2}},m_i,\mu_{i+\frac{1}{2}}} \in \mathbb{C}$ are coefficients appearing in the sum that depend on the morphism $h \in \mathrm{Hom}(\rho \otimes \rho, \rho \otimes \rho)$ and $(\mu_{i-\frac{1}{2}}, m_i, \mu_{i+\frac{1}{2}})$.

Let us define an operator $h_i$ which takes a state of the form (2.3) to a state where the local information around site $i$ is modified to the RHS of (2.4). Then the Hamiltonian is

$$
H = -\sum_i h_i \,.
\tag{2.6}
$$

**Lattice SymTFT picture and Action of Symmetry.** There is a natural physical home for such a lattice model: it arises on the boundaries of a (2+1)d TQFT $\mathfrak{Z}(\mathcal{C})$ obtained by performing the Turaev-Viro-Barrett-Westbury state-sum construction with $\mathcal{C}$ being the input fusion category. This 3d TQFT admits a topological boundary $\mathfrak{B}^{\mathrm{inp}}_{\mathcal{C}}$ (input boundary condition), whose topological line defects form the fusion category $\mathcal{C}$. The module category $\mathcal{M}$

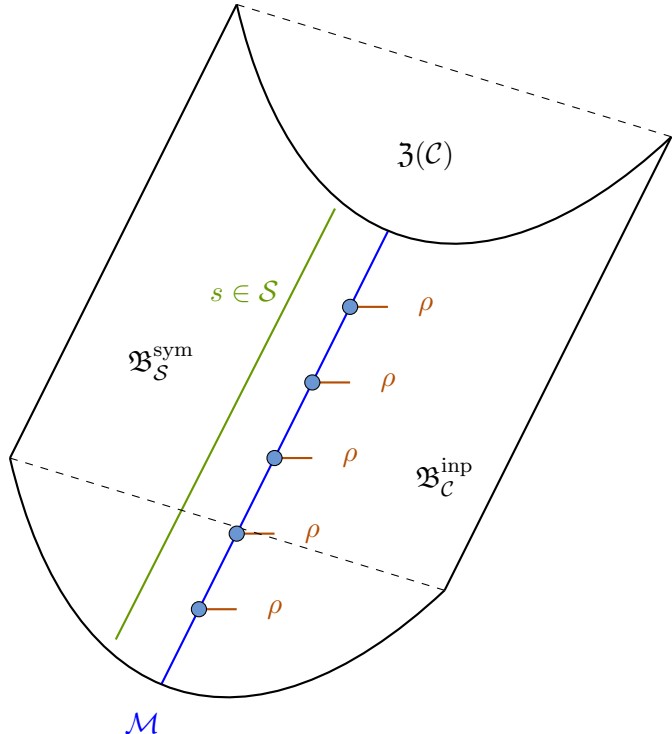

Figure 1: Three-dimensional sketch of the SymTFT picture: the (2+1)d TQFT $\mathfrak{Z}(\mathcal{C})$ has two boundaries, $\mathfrak{B}_{\mathcal{S}}^{\mathrm{sym}}$ and $\mathfrak{B}_{\mathcal{C}}^{\mathrm{inp}}$. The interface between these is the module category $\mathcal{M}$ (blue). The topological lines on $\mathfrak{B}_{\mathcal{C}}^{\mathrm{inp}}$ form the category $\mathcal{C}$, whereas the ones on $\mathfrak{B}_{\mathcal{S}}^{\mathrm{sym}}$ form $\mathcal{S} = \mathcal{C}_{\mathcal{M}}^{*}$. The latter is the symmetry of the spin-chain. We can think of the spin-chain as located along the interface specified by the module category, with $\rho \in \mathcal{C}$ extending into $\mathfrak{B}_{\mathcal{C}}^{\mathrm{inp}}$. The symmetry acts from the left (i.e. taking a topological defect (green) of $\mathfrak{B}_{\mathcal{S}}^{\mathrm{sym}}$ and pushing it parallel to $\mathcal{M}$).

describes topological line defects acting as interfaces from the topological boundary $\mathfrak{B}_{\mathcal{C}}^{\mathrm{inp}}$ of $\mathfrak{Z}(\mathcal{C})$ to another topological boundary $\mathfrak{B}_{\mathcal{S}}^{\mathrm{sym}}$ of $\mathfrak{Z}(\mathcal{C})$. The topological line defects of $\mathfrak{B}_{\mathcal{S}}^{\mathrm{sym}}$ form the fusion category $\mathcal{S} = \mathcal{C}_{\mathcal{M}}^*$.

Thus, the lattice model under discussion arises on a boundary of the 3d TQFT $\mathfrak{Z}(\mathcal{C})$ that is partitioned into two halves. The top half carries the boundary condition $\mathfrak{B}_{\mathcal{C}}^{\mathrm{inp}}$ while the bottom half carries the boundary condition $\mathfrak{B}_{\mathcal{S}}^{\mathrm{sym}}$. The entire 2d boundary is wrapped along a circle whose direction is such that the interface between the two boundary conditions is wrapped along this circle. A state of the form (2.3) arises from a collection of topological line operators $m_i$ inserted along the interface, plus a collection of topological local operators $\mu_{i+\frac{1}{2}}$ between $m_i$ and $m_{i+1}$, arising at a junction with a topological line $\rho$ of $\mathfrak{B}_{\mathcal{C}}^{\mathrm{inp}}$. The building block (2.4) of the Hamiltonian $H$ is a topological manipulation of these topological line and local operators.

So far the boundary condition $\mathfrak{B}_{\mathcal{S}}^{\mathrm{sym}}$ placed on the bottom half has not played any role except providing an interface where the module category $\mathcal{M}$ lives. The topological lines of $\mathfrak{B}_{\mathcal{S}}^{\mathrm{sym}}$ can act on the operators placed along the interface, which provides an action of the fusion category $\mathcal{S}$ on the Hilbert space $\mathcal{V}_u$ of the lattice model

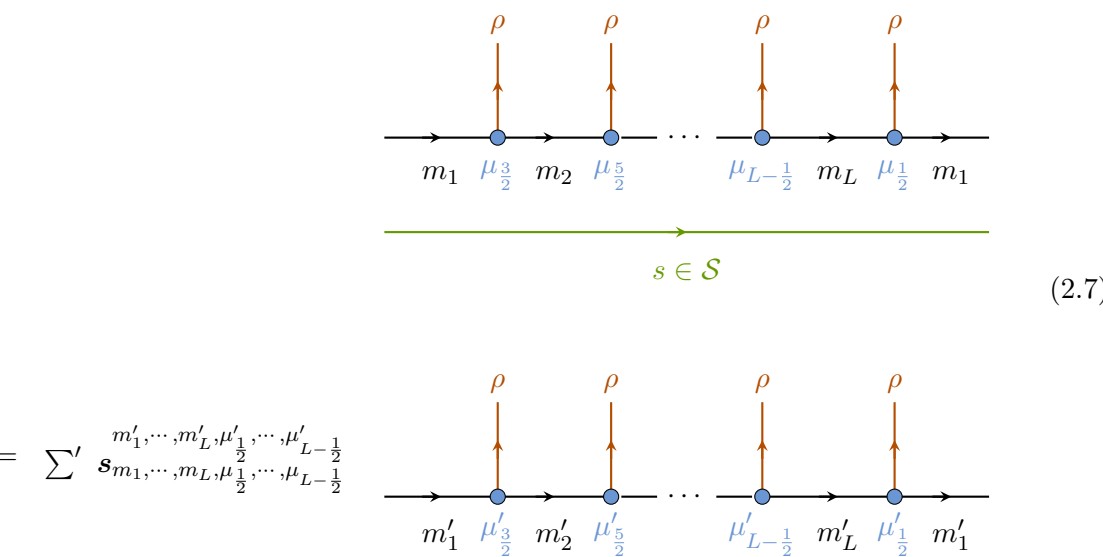

$$(2.7)$$

where the sum is over simple objects $m_i' \in \mathcal{M}$ and basis morphisms $\mu_{i+\frac{1}{2}}' \in \mathrm{Hom}(m_i', \rho \otimes m_{i+1}')$, and $s_{m_1, \cdots, m_L, \mu_{\frac{1}{2}}, \cdots, \mu_{L-\frac{1}{2}}}^{m_1', \cdots, m_L', \mu_{\frac{1}{2}}', \cdots, \mu_{L-\frac{1}{2}}'} \in \mathbb{C}$ are coefficients appearing in the sum that depend on the simple object $s \in \mathcal{S}$ and $(m_1, \cdots, m_L, \mu_{\frac{1}{2}}, \cdots, \mu_{L-\frac{1}{2}})$.

This action commutes with the Hamiltonian move (2.4) due to the topological nature of the lines and local operators involved. To put it simply, the action of symmetry is from the bottom, while the Hamiltonian acts from the top, and hence the two commute. As a

consequence of this, $\mathcal{S}$ is a symmetry of any such lattice model built using input data $(\mathcal{C}, \mathcal{M})$.

Note that we can also perform Turaev-Viro construction with $\mathcal{S}$ being the input fusion category, and the resulting 3d TQFT $\mathfrak{Z}(\mathcal{S})$ can be identified with the Turaev-Viro TQFT based on $\mathcal{C}$

$$\mathfrak{Z}(\mathcal{S}) = \mathfrak{Z}(\mathcal{C}) \, . \tag{2.8}$$

Both $\mathcal{S}$ and $\mathcal{C}$ arise on topological boundary conditions of this TQFT. Since $\mathcal{S}$ is the symmetry of the lattice models, this TQFT is also referred to as the Symmetry TFT (SymTFT) for $\mathcal{S}$ [21, 38]. Thus, $\mathcal{S}$-symmetric lattice models being discussed here find their natural home on the boundaries of the SymTFT for $\mathcal{S}$. See figure 1 for a *three-dimensional* sketch of the whole setup.

**Gauging.** Changing the input $\mathcal{C}$-module category to $\mathcal{M}'$ while keeping $(\rho, h)$ the same leads to another lattice model, which is obtained by gauging the symmetry $\mathcal{S}$ of the original lattice model based on $\mathcal{M}$. The symmetry of the new system is $\mathcal{S}' = \mathcal{C}^*_{\mathcal{M}'}$. This is because changing $\mathcal{M} \to \mathcal{M}'$ changes the boundary condition $\mathfrak{B}^{\text{sym}}_{\mathcal{S}} \to \mathfrak{B}^{\text{sym}}_{\mathcal{S}'}$, and such a change of boundary condition is implemented precisely by a gauging of the topological lines of $\mathfrak{B}^{\text{sym}}_{\mathcal{S}}$.

We can describe the precise gauging of $\mathcal{S}$ that is involved. Recall that different possible gaugings of a fusion category symmetry $\mathcal{S}$ correspond to indecomposable module categories of $\mathcal{S}$ [2]. The gauging under discussion corresponds to a $\mathcal{S}$-module category $\mathcal{N}$ satisfying

$$\mathcal{M} \boxtimes_{\mathcal{S}} \mathcal{N} = \mathcal{M}' \, , \tag{2.9}$$

where $\mathcal{M}$ is regarded as a right module category (and hence a bimodule category for $(\mathcal{C}, \mathcal{S})$) over $\mathcal{S}$ and $\mathcal{N}$ a left module category over $\mathcal{S}$, and $\boxtimes_{\mathcal{S}}$ is the relative Deligne product over $\mathcal{S}$ defined between right and left module categories of $\mathcal{S}$. The relative Deligne product intertwines the left $\mathcal{C}$-module structures on $\mathcal{M}$ and $\mathcal{M}'$.

**Twisted Sectors.** Let us now discuss the model with symmetry twisted boundary conditions as one goes around the circle. Let the twist be by a simple object $s \in \mathcal{S}$. The corresponding Hilbert space of states is referred to as $s$-twisted sector Hilbert space $\mathcal{V}_s$ of the

lattice model. A basis vector in $\mathcal{V}_s$ is

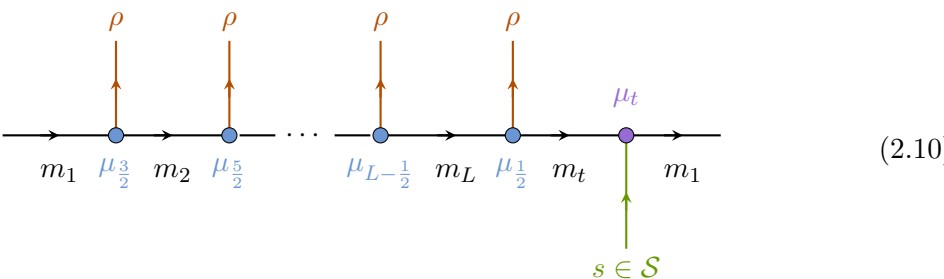

$$(2.10)$$

where we utilize the fact that $\mathcal{M}$ is also a right $\mathcal{S}$-module category and pick a basis morphism $\mu_t \in \mathrm{Hom}(m_t \otimes s, m_1)$. The Hamiltonian acting on this Hilbert space is

$$H^s = -h_1^s - \sum_{i \neq 1} h_i \,, \qquad (2.11)$$

where $h_i$ are the same pieces appearing in the untwisted sector Hamiltonian (2.6), and $h_1^s$ is an operator based on the $s$-twisted $h$-move

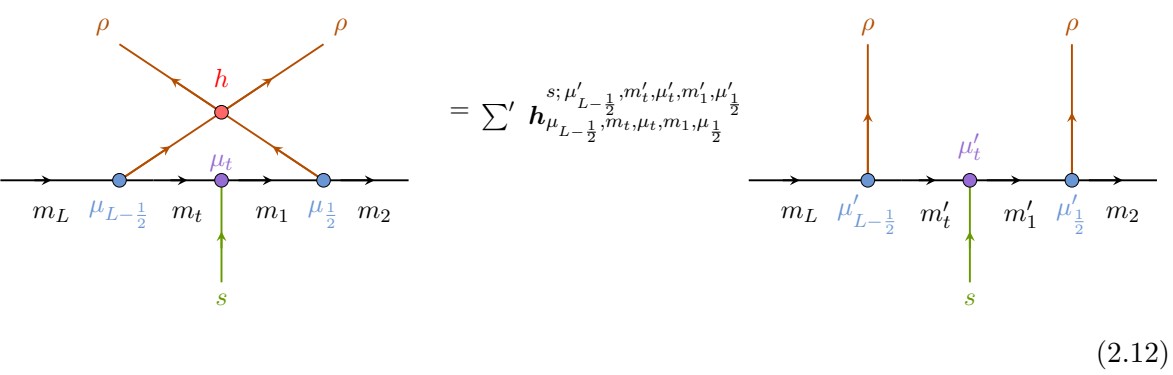

$$(2.12)$$

In general, the $\mathcal{S}$ symmetry can mix together all these Hilbert spaces

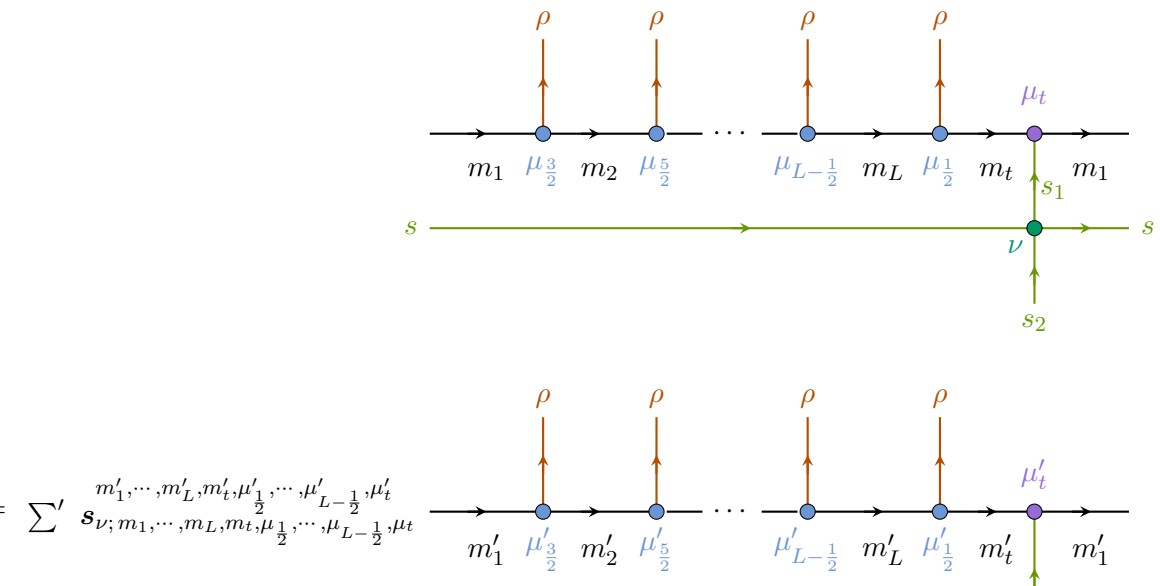

$$(2.13)$$

which is an action of symmetry $s$, parametrized by a basis morphism $\nu \in \mathrm{Hom}(s \otimes s_2, s_1 \otimes s)$, that sends an $s_1$-twisted sector state to an $s_2$-twisted sector state. Such a symmetry action intertwines the actions of Hamiltonians on the $s_1$ and $s_2$ twisted sector Hilbert spaces. Note that if $s_1 = 1$, then we have a symmetry action mapping the untwisted sector into $s_2$-twisted sector, and if instead $s_2 = 1$, then we have a symmetry action mapping the $s_1$-twisted sector into the untwisted sector. Finally, if both $s_1 = s_2 = 1$, then we recover the standard symmetry action (2.7)

**Local Operators Charged under Symmetry.** Let us now discuss local operators acting on the lattice model. We discuss local operators that act on the untwisted sector, but their action can in general map the untwisted sector to both untwisted and twisted sectors. These local operators can be organized according to how the symmetry $\mathcal{S}$ acts on them, or in other words what their charges under $\mathcal{S}$ are. As studied in [18, 21], these charges are described by objects of the Drinfeld center $\mathcal{Z}(\mathcal{S})$ of the fusion category $\mathcal{S}$, which is a modular tensor category (MTC) formed by topological line defects of the SymTFT $\mathfrak{Z}(\mathcal{S})$.

Pick a simple line $\boldsymbol{Q}$ of the SymTFT. A multiplet of local operators acting on the lattice model transforming irreducibly under symmetry $\mathcal{S}$ according to charge $\boldsymbol{Q}$ is specified by a basis morphism

$$\boldsymbol{Q}_\mu^\rho \in \mathrm{Hom}(\rho \otimes Z_{\mathcal{C}}(\boldsymbol{Q}), \rho)\,, \tag{2.14}$$

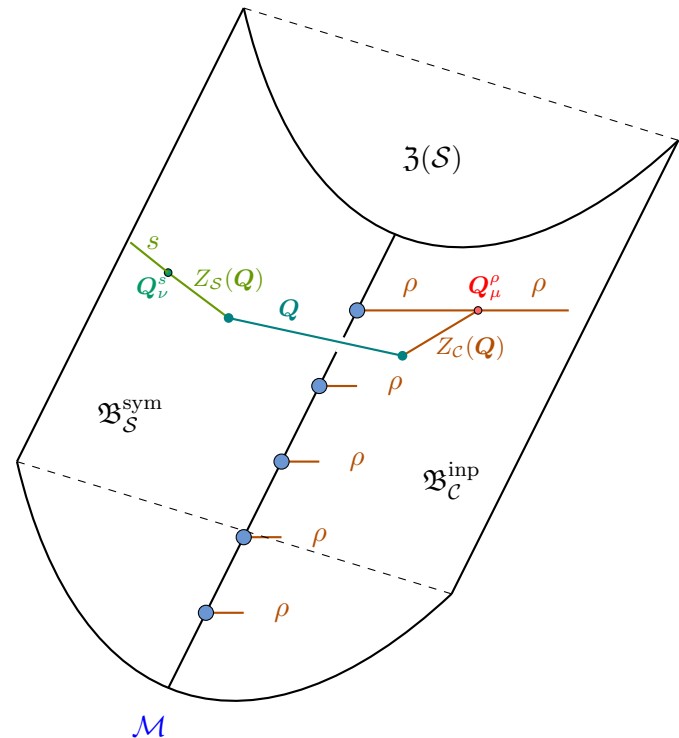

Figure 2: Lattice SymTFT description of local operators. The bulk topological line $\boldsymbol{Q} \in \mathcal{Z}(\mathcal{S})$ (teal) ends on both boundaries $\mathfrak{B}_{\mathcal{S}}^{\mathrm{sym}}$ and $\mathfrak{B}_{\mathcal{C}}^{\mathrm{inp}}$. The former extends along $\mathfrak{B}_{\mathcal{S}}^{\mathrm{sym}}$ and is converted to $s$ by $\boldsymbol{Q}_{\nu}^{s}$, while the latter extends along $\mathfrak{B}_{\mathcal{C}}^{\mathrm{inp}}$ and is absorbed into $\rho$ by $\boldsymbol{Q}_{\mu}^{\rho}$.

where $Z_{\mathcal{C}}(\boldsymbol{Q}) \in \mathcal{C}$ is the line operator of the boundary $\mathfrak{B}_{\mathcal{C}}^{\mathrm{inp}}$ obtained by projecting/stacking the bulk line $\boldsymbol{Q}$ onto it. Mathematically, $Z_{\mathcal{C}}(\boldsymbol{Q})$ is obtained by applying on $\boldsymbol{Q}$ the forgetful functor from the Drinfeld center $\mathcal{Z}(\mathcal{C})$ of $\mathcal{C}$ to $\mathcal{C}$, where we use the fact that there is an equivalence $\mathcal{Z}(\mathcal{C}) \cong \mathcal{Z}(\mathcal{S})$ induced by the module category $\mathcal{M}$.

Each such multiplet, parametrized by $(\boldsymbol{Q}, \boldsymbol{Q}_{\mu}^{\rho})$, contains local operators mapping the untwisted sector to the $s$-twisted sector parametrized by basis morphisms

$$\boldsymbol{Q}_{\nu}^{s} \in \operatorname{Hom}(s, Z_{\mathcal{S}}(\boldsymbol{Q})) \tag{2.15}$$

for any simple $s \in \mathcal{S}$, where $Z_{\mathcal{S}}(\boldsymbol{Q}) \in \mathcal{S}$ is the line operator of the boundary $\mathfrak{B}_{\mathcal{S}}^{\mathrm{sym}}$ obtained by projecting/stacking the bulk line $\boldsymbol{Q}$ onto it. Mathematically, $Z_{\mathcal{S}}(\boldsymbol{Q})$ is obtained by applying on $\boldsymbol{Q}$ the forgetful functor from the Drinfeld center $\mathcal{Z}(\mathcal{S})$ to $\mathcal{S}$. The three-dimensional picture is shown in figure 2.

The $s$-twisted sector state resulting from the action of such an operator is

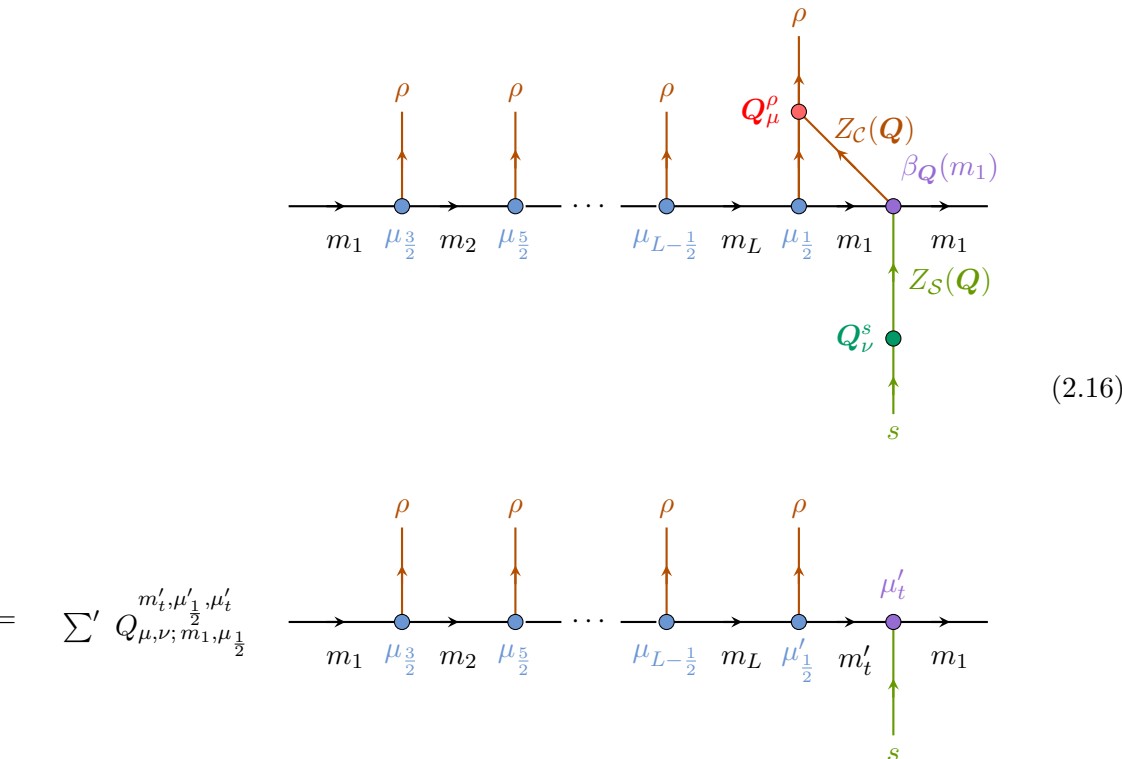

$$(2.16)$$

where

$$\beta_{\boldsymbol{Q}}(m_1): \ m_1 \otimes Z_{\mathcal{S}}(\boldsymbol{Q}) \to Z_{\mathcal{C}}(\boldsymbol{Q}) \otimes m_1 \qquad (2.17)$$

is the half-braiding of $\boldsymbol{Q}$ with $m_1$ obtained from the equivalence of $\mathcal{Z}(\mathcal{S})$ with the Drinfeld center $\mathcal{Z}\left(\mathcal{C}^{\text{multi}}(\mathcal{C},\mathcal{S},\mathcal{M})\right)$ of the multi-fusion category $\mathcal{C}^{\text{multi}}(\mathcal{C},\mathcal{S},\mathcal{M})$ formed by combining the fusion categories $\mathcal{C}$ and $\mathcal{S}$ with the bimodule category $\mathcal{M}$ (along with $\mathcal{M}^{\text{op}}$). Physically, the action of this operator corresponds to stacking the bulk line $\boldsymbol{Q}$ transversely to the interface line $m_1$, which leads to the creation of boundary lines $Z_{\mathcal{C}}(\boldsymbol{Q}), Z_{\mathcal{S}}(\boldsymbol{Q})$. The line $Z_{\mathcal{C}}(\boldsymbol{Q})$ is absorbed into $\rho$ by the junction $\boldsymbol{Q}_\mu^\rho$, and the line $Z_{\mathcal{S}}(\boldsymbol{Q})$ is projected into simple line $s$ by using $\boldsymbol{Q}_\mu^s$. Finally the resulting configuration of lines is converted into the form of a state resulting in coefficients $Q_{\mu,\nu;\,\tilde{m}_1,\mu_{\frac{1}{2}}}^{m_t',\mu_{\frac{1}{2}}',\mu_t'} \in \mathbb{C}$ that depend only on $\{\boldsymbol{Q},\boldsymbol{Q}_\mu^\rho,\boldsymbol{Q}_\nu^s,m_1,\mu_{\frac{1}{2}},m_t',\mu_{\frac{1}{2}}',\mu_t'\}$.

The symmetry action $\tilde{s} \in \mathcal{S}$ on such local operators may be expressed as

$$\tilde{s}: \ \mathcal{O}(\boldsymbol{Q},\boldsymbol{Q}_\mu^\rho,\boldsymbol{Q}_\nu^s) \to \sum_{s',\nu',\alpha'} q_{\tilde{s}}^{s',\nu',\alpha'} \mathcal{O}(\boldsymbol{Q},\boldsymbol{Q}_\mu^\rho,\boldsymbol{Q}_{\nu'}^{s'}), \qquad (2.18)$$

where $\mathcal{O}(\boldsymbol{Q},\boldsymbol{Q}_\nu^s,\boldsymbol{Q}_\mu^\rho)$ is the $\nu$-th operator in multiplet $\mu$ carrying charge $\boldsymbol{Q}$, $\alpha$ parametrizes basis morphisms in $\text{Hom}(\tilde{s} \otimes s \to s' \otimes \tilde{s})$, and $q_{\tilde{s}}^{s',\nu',\alpha'} \in \mathbb{C}$. Such actions were studied in many

cases of fusion category symmetries in [21, 29] which we will take as input in our examples studied in this paper.

More precisely, the symmetry action (2.18) describes a generalized commutation relation between the actions of the local operators $\mathcal{O}(\boldsymbol{Q}, \boldsymbol{Q}^\rho_\mu, \boldsymbol{Q}^s_\nu)$ and the symmetry $\widetilde{s}$ on the untwisted/twisted sector Hilbert spaces

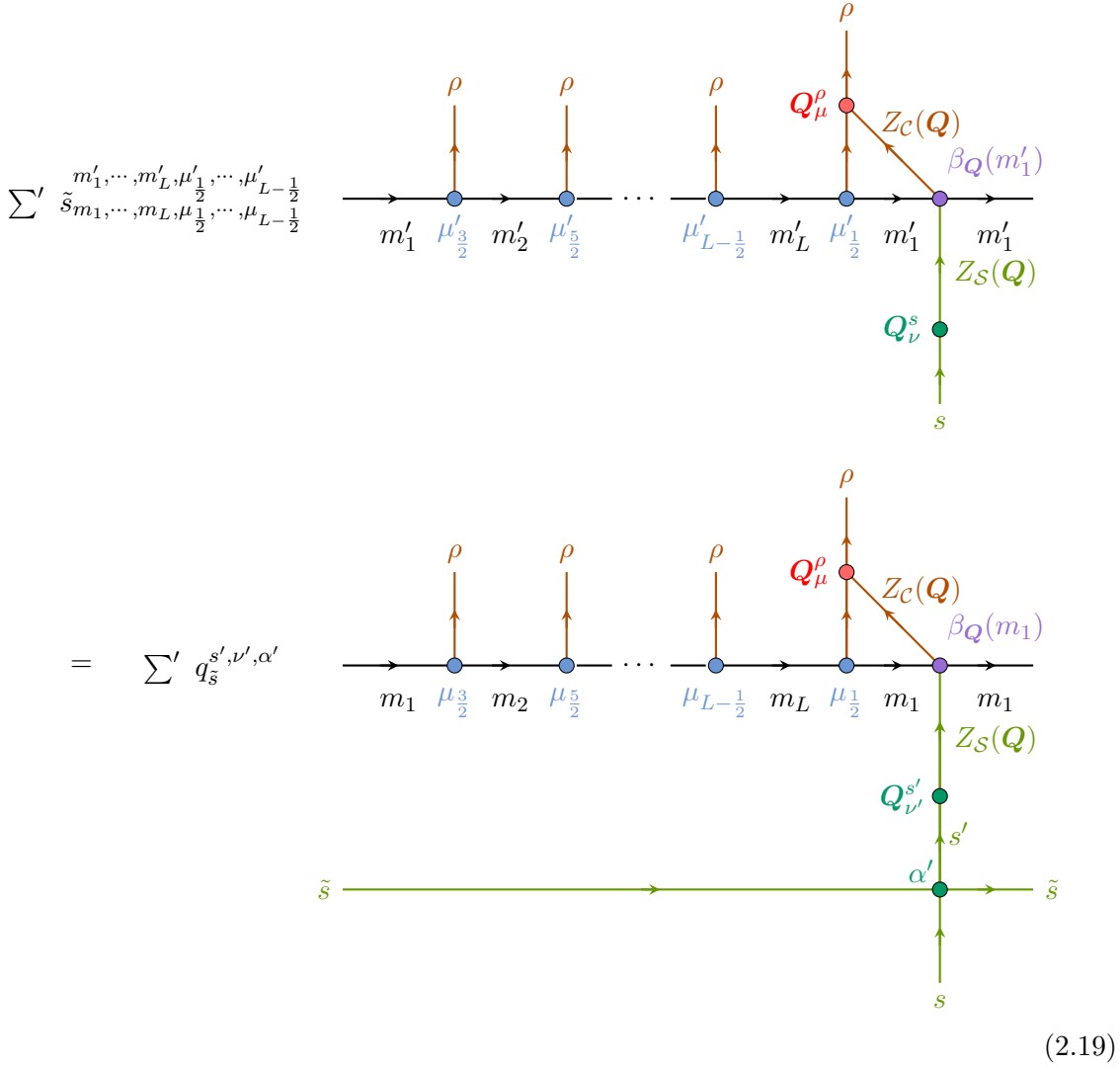

$$(2.19)$$

where in the top half of the equation we have first acted by symmetry $\tilde{s}$ to take the untwisted sector into the untwisted sector and then by the operator $\mathcal{O}(\boldsymbol{Q}, \boldsymbol{Q}^\rho_\mu, \boldsymbol{Q}^s_\nu)$ to take the untwisted sector to the $s$-twisted sector, while on the bottom half of the equation we first act by operators $\mathcal{O}(\boldsymbol{Q}, \boldsymbol{Q}^\rho_\mu, \boldsymbol{Q}^{s'}_{\nu'})$ taking the untwisted sector into the $s'$-twisted sector and then perform an action of symmetry $\tilde{s}$ taking $s'$-twisted sector to $s$-twisted sector.

## 2.2 Abelian Group-like Symmetry

In this section, as a preparation for the study of models with fusion category symmetries, we describe familiar lattice models with finite group Abelian symmetries formulated as anyon chain models (see also [63, 66] and references therein).

The models described here are defined in terms of categorical data introduced in section 2.1, but reduce to prototypical spin models (essentially generalized Ising models) defined on tensor decomposable Hilbert spaces. For the finite group $\mathbb{Z}_n$ for example, each lattice site hosts a spin degree of freedom associated to an $n$-dimensional Hilbert space and the corresponding Hamiltonians can be expressed in terms of $\mathbb{Z}_n$ clock and shift operators that are generalizations of Pauli operators (for which $n = 2$).

**Hilbert space and symmetry action.** To construct a model with $G$ global symmetry, where $G$ is a finite Abelian group, we fix $\mathcal{C} = \mathsf{Vec}_G$. The simple objects in $\mathcal{C}$ are labelled as $g \in G$. To define the Hilbert space of the lattice model, we choose the regular module category $\mathcal{M} = \mathsf{Vec}_G$ and the object $\rho = \oplus g$. With this choice, the Hilbert space of a length $L$ lattice with periodic boundary conditions is spanned by states corresponding to fusion trees

$$|\vec{g}\rangle := |g_1, g_2, \ldots, g_L\rangle \ = \quad$$

(2.20)

At each node, $\rho$ simply provides the appropriate morphism between the adjacent group variables. For example between the sites $j$ and $j + 1$, the element $g_j g_{j+1}^{-1} \in \rho$ is picked out. Since there is a single such morphism for each $g_j, g_{j+1}$, the local Hilbert space at each site is $|G|$-dimensional with a basis spanned by $g \in G$. For $G = \mathbb{Z}_2$, these are the familiar $\sigma = \uparrow, \downarrow$ spin degrees of freedom. The total Hilbert space of the anyon chain model is

$$\mathcal{V}_1 = \mathbb{C}[G]^{\otimes L}, \qquad \mathbb{C}[G] := \mathbb{C}^{|G|}. \tag{2.21}$$

The model comes equipped with a symmetry $\mathcal{S} = \mathcal{C}_{\mathcal{M}}^* = \mathsf{Vec}_G$. The symmetry operators, denoted as $\mathcal{U}_g$, act on the Hilbert space as

(2.22)

i.e.

$$\mathcal{U}_g |g_1, g_2, \ldots, g_L\rangle = |g_1 g, g_2 g, \ldots, g_L g\rangle. \tag{2.23}$$

**SymTFT for Abelian spin chains.** Now, let us situate this spin model within the context of the SymTFT boundary. The SymTFT corresponding to $\mathcal{C} = \mathsf{Vec}_G$ is the Drinfeld center $\mathcal{Z}(\mathsf{Vec}_G)$ or equivalently the $G$ Dijkgraaf-Witten theory. The bulk lines of the SymTFT are dyons that carry a magnetic label $g \in G$ and an electric label $\alpha \in \mathsf{Rep}(G) \cong G$. A general line is denoted as $m_g e_\alpha$. The topological spin of such a line is $\alpha(g) \in U(1)$, while the braiding of a line $m_g e_\alpha$ with another line $m_{g'} e_{\alpha'}$ is $\alpha(g')\alpha'(g)$. Topological boundary conditions for the SymTFT are given by Lagrangian algebras in $\mathcal{Z}(\mathsf{Vec}_G)$. Concretely, Lagrangian algebras are nothing but maximal sets of bulk topological lines that are bosonic (spin 1) and braid-trivially with one another. The topological boundary condition physically means that all lines in the Lagrangian algebra can end (or be condensed) on the corresponding boundary. Topological boundaries for the $G$-SymTFT are classified by a tuple $(H, \beta)$ where $H$ is a subgroup of $G$ and $\beta \in H^2(H, U(1))$. This is precisely also the data that classifies $1 + 1$ dimensional gapped phases with $G$ global symmetry, a fact that we will utilize to construct Hamiltonians for all gapped phases in section 3.2.

For now we focus on the topological boundaries that enter the construction of the anyon chain model. This is the boundary condition $(1, 1)$ i.e., with $H$ being the trivial subgroup of $G$. We choose such a boundary as both the input and symmetry boundary

$$\mathfrak{B}^{\mathrm{inp}} = \mathfrak{B}^{\mathrm{sym}} = \bigoplus_{\alpha \in \mathsf{Rep}(G)} e_\alpha \,. \tag{2.24}$$

The category of lines on each of this topological boundary is $\mathsf{Vec}_G$. The projection of any bulk line in the SymTFT onto a boundary line is provided by the forgetful functor that only remembers the magnetic label of the line

$$\mathcal{Z}(\mathsf{Vec}_G) \;\ni\; m_g e_\alpha \longmapsto g \;\in\; \mathsf{Vec}_G \,. \tag{2.25}$$

Now we may situate a state (2.20) on the boundary of the SymTFT such that the degrees of freedom live on the interface (labelled by $\mathcal{M}$) between $\mathfrak{B}^{\mathrm{sym}}$ (below) and $\mathfrak{B}^{\mathrm{inp}}$ (above), while the SymTFT is coming out of the page. We will now describe other features of these models always keeping in mind this SymTFT construction.

**Symmetry-twisted Hilbert space.** When studying a quantum system with a global symmetry, it is insightful to probe symmetry aspects of the system by coupling to a background gauge field. In the Hamiltonian formulation, this corresponds to studying the model in the presence of an arbitrary configuration of background symmetry defects. On a lattice model, these defects are essentially symmetry twisted boundary conditions since any configuration of

background symmetry defects can be unitarily mapped to a single defect at some chosen site of the lattice. In the anyon model these are depicted as

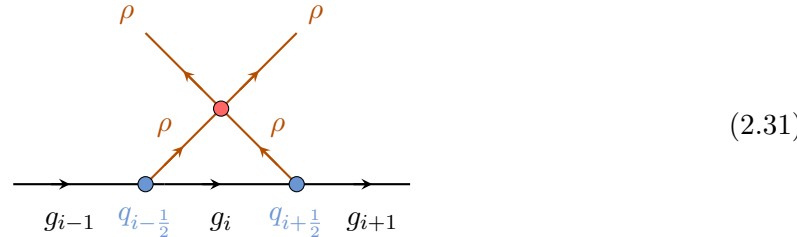

$$\qquad (2.26)$$

where $g \in G$ is a topological line on $\mathfrak{B}^{\mathrm{sym}}$ that ends on $\mathcal{M}$. As can be immediately deduced, each such symmetry twisted sector is isomorphic as a vector space to the untwisted sector. Therefore the total Hilbert space decomposes into a sum of $g$-twisted sectors

$$\mathcal{V} = \bigoplus_g \mathcal{V}_g , \qquad \mathcal{V}_g = \mathbb{C}[G]^{\otimes L} . \qquad (2.27)$$

**On-site spin chain operators.** There are two kinds of operators. The first are denoted $X_g$ for $g \in G$ and act off-diagonally in the group basis

$$X_g |g'\rangle = |gg'\rangle . \qquad (2.28)$$

These are a direct generalization of the Pauli operator $\sigma^x$ that acts off-diagonally between two spin states $|0\rangle$ and $|1\rangle$ that span $\mathbb{C}[\mathbb{Z}_2]$. The second kind of operator $Z_\alpha$ is labelled by a representation $\alpha$ and acts diagonally as

$$Z_\alpha |g\rangle = \alpha(g)|g\rangle . \qquad (2.29)$$

This is a direct generalization of the Pauli operator $\sigma^z$ that acts via the sign representation of $\mathbb{Z}_2$ on the basis states in $\mathbb{C}[\mathbb{Z}_2]$. The identity operator corresponds to the trivial representation. These operators transform under the $G$ action as

$$\mathcal{U}_g X_h \mathcal{U}_g^{-1} = X_h , \qquad \mathcal{U}_g Z_\alpha \mathcal{U}_g^{-1} = \alpha(g)^{-1} Z_\alpha . \qquad (2.30)$$

Therefore, while $Z_\alpha$ act on the local Hilbert space, they do not directly appear in $G$-symmetric Hamiltonians, but appear via combinations such as $Z_{\alpha,j} Z_{\alpha,j'}^\dagger$. Instead, since $Z_\alpha$ are charged under $G$, they do serve as symmetry breaking order parameters.

**$G$-symmetric operators.** In the anyon model, $G$-symmetric operators arise rather naturally as described in section 2.1. These are given by morphisms $h : \rho \otimes \rho \to \rho \otimes \rho$.

$$\qquad (2.31)$$

One may obviously also consider longer range operators implementing an endomorphism of $\rho^{\otimes k}$ for $k > 2$ for example by appropriately concatenating such operators. These operators are $G$ symmetric as their action on the state space commutes with the symmetry action from below. More specifically, the operators (2.31) can be expressed in terms of the following building blocks

$$\mathcal{O}_{(g_L,h,g_R),j} \quad = \quad$$

$$(2.32)$$

which act as

$$\mathcal{O}_{(g_L,h,g_R),j}\big| \ldots , g_{j-1} , g_j , g_{j+1} , \ldots \big\rangle = \delta_{g_L ,g_{j-1}g_j^{-1}}\delta_{g_R ,g_j g_{j+1}^{-1}}\big| \ldots , g_{j-1} , hg_j , g_{j+1} , \ldots \big\rangle . \quad (2.33)$$

Similarly, we may also consider $G$-symmetric operators acting on twisted sector states

$$\mathcal{O}^g_{(g_L,h,g_R),i} \quad = \quad$$

$$(2.34)$$

which act as

$$\mathcal{O}^g_{(g_L,h,g_R),j}\big| \ldots , g_{j-1} , g_j , g_{j+1} , \ldots \big\rangle_g = \delta_{g_L ,g_{j-1}g_j^{-1}}\delta_{g_R ,g_j g g_{j+1}^{-1}}\big| \ldots , g_{j-1} , hg_j , g_{j+1} , \ldots \big\rangle_g . \quad (2.35)$$

These operators can be expressed in terms of the onsite spin operators described above as

$$\mathcal{O}^g_{(g_L,h,g_R),j} = P^{(g_L)}_{j-\frac{1}{2}} X_{h,j} P^{(g_R,g)}_{j+\frac{1}{2}} , \quad (2.36)$$

where $P^{(g_L)}$ and $P^{(g_R,g)}$ are projection operators that impose the delta function constraints in (2.33) and (2.35). Concretely these are expressed as a linear combination of $Z_{\alpha,j-1}Z^{\dagger}_{\alpha,j}$ and $Z_{\alpha,j}Z^{\dagger}_{\alpha,j+1}$ respectively. Notice that the symmetry twist only enters the operators through the projection operator between sites $j$ and $j + 1$. We describe the explicit form of such operators appearing in the fixed-point Hamiltonians for the different gapped phases realized for $G = \mathbb{Z}_4 \times \mathbb{Z}_2$ in appendix A.

**$G$-charges: twisted and untwisted sector operators.** The space of local operators including both twisted and untwisted sector ones can be decomposed into classes labelled by bosonic lines in the SymTFT. These operators become order parameters of gapped phases when the corresponding lines condense on the physical boundary. We will describe the gapped phases and their corresponding order parameters in section 3.2. The local operators corresponding to a buk line $m_g e_\alpha$ takes the very simple form

$$\mathcal{O}_{(g,\alpha),j} = \mathcal{T}_{g,j} Z_{\alpha,j} \,, \tag{2.37}$$

where $Z_{\alpha,j}$ is the charged operator introduced above while $\mathcal{T}_g$ is a twisted sector operator that acts on the states by introducing a $g$ symmetry twist on the $j$-th site.

**Changing the Module Category.** Now consider changing the module category from $\mathcal{M}$ to $\mathcal{M}'$ while keeping the input category $\mathcal{C}$ and other data, i.e. $\rho$ and the Hamiltonian, fixed. This leads to a new model with the symmetry $\mathcal{S}' = \mathcal{C}^*_{\mathcal{M}'}$. $\mathsf{Vec}_G$ module categories are labelled as $\mathcal{M}(H,\beta)$ where $H$ is a subgroup of $G$ and $\beta \in H^2(H, U(1))$. The set of simple objects in $\mathcal{M}(H,\beta)$ is the set of left cosets $G/H \cong K$. We will typically denote such a module category as $\mathsf{Vec}_K$. Let us label a simple object in $\mathcal{M}(H,\beta)$ as $k$, which is a representative element of the $H$-coset $[k]$. The module action of $G$ on $\mathcal{M}(H,\beta)$ is given by

$$\begin{aligned} G \times \mathcal{M}(H,\beta) &\to \mathcal{M}(H,\beta) \\ (g,k) &\mapsto gk \,, \end{aligned} \tag{2.38}$$

where $gk$ is a representative of the $H$-coset $[gk]$. It follows that the simple objects in $\mathsf{Vec}_G$ labelled by $h \in H \subseteq G$ act trivially on all the objects in $\mathcal{M}(H,\beta)$. However, there can be a non-trivial module associator determined by the 2-cocycle $\beta$ as

$$\tag{2.39}$$

The original choice of module category we started with was $\mathcal{M}(1,1) = \mathsf{Vec}_G$. Changing the module category from $\mathcal{M} = \mathcal{M}(1,1)$ to $\mathcal{M}' = \mathcal{M}(H,\beta)$ corresponds to gauging $H \subseteq G$ on the symmetry boundary with a choice of discrete torsion $\beta$.

## 2.3 $\mathsf{Rep}(S_3)$ Symmetry

In this section we describe a lattice model with $\mathsf{Rep}(S_3)$ symmetry formulated as an anyon chain model. Our a model is concretely represented on a tensor product Hilbert space built

on a one-dimensional lattice of alternating qubits and qutrits – see the companion paper [70] for a discussion directly from that perspective.

**Setup and state space.**   As input for the model, we pick the following categorical data

$$\mathcal{C} = \mathsf{Vec}_{S_3}\,, \quad \mathcal{M} = \mathsf{Vec}_{\mathbb{Z}_3}\,, \quad \rho = \bigoplus_{g \in S_3} g\,. \tag{2.40}$$

We present the finite group $S_3$ as

$$S_3 = \langle\, a\,, b \,\mid\, a^3 = 1\,, b^2 = 1\,, bab = a^2 \,\rangle\,. \tag{2.41}$$

Recall, that $\mathsf{Vec}_{S_3}$ is the category of $S_3$-graded vector spaces. Its simple objects are group elements $g \in S_3$ while the monoidal or fusion structure follows from $S_3$ group multiplication. Next, the simple objects in the module category $\mathcal{M} = \mathsf{Vec}_{\mathbb{Z}_3}$ are right cosets

$$S_3/\mathbb{Z}_2^b \simeq \left\{ 1 \sim (1,b)\,, m \sim (a, ab)\,, m^2 \sim (a^2, a^2 b) \right\}\,. \tag{2.42}$$

The module action follows from the group action on the set of cosets. Specifically

$$a^q \times m^p = m^{p+q}\,,$$
$$b \times \{1\,, m\,, m^2\} = \{1\,, m^2\,, m\}$$
$$ab \times \{1\,, m\,, m^2\} = \{m\,, 1\,, m^2\} \tag{2.43}$$
$$a^2 b \times \{1\,, m\,, m^2\} = \{m^2\,, m\,, 1\}\,.$$

The Hilbert space of the model on a chain of length $L$ with periodic boundary conditions is spanned by states corresponding to diagrams of the following type

$$\tag{2.44}$$

It can be seen that for any choice of $m_j$ and $m_{j+1}$, there is a two-dimensional space of morphisms at $\mu_{j+\frac{1}{2}}$ which correspond to picking a $g \in S_3$ such that $m_j = g \times m_{j+1}$. Therefore the total state space decomposes into a tensor product of local qutrit state spaces (i.e., $\mathbb{C}^3$) assigned to the integer sites and qubits (i.e., $\mathbb{C}^2$) assigned to each half integer site. We work with a qutrit basis $|p_j\rangle$ where $p_j = 0, 1, 2$ correspond to module objects $1\,, m\,, m^2$ respectively and a qubit basis $|q_{j+1/2}\rangle$ with $q_{j+1/2} = 0, 1$ using a map $\phi : S_3 \to \mathbb{Z}_2$ such that $\phi(a^p b^q) = q$.

Then a choice of $\mu_{j+\frac{1}{2}}$ corresponds to a group element $g \in S_3$ and picks a state $\phi(g)$ in the qubit space. Doing so, we end up with the basis states.

$$|\vec{p}, \vec{q}\rangle \equiv |\cdots, p_j, q_{j+\frac{1}{2}}, p_{j+1}, q_{j+\frac{3}{2}}, \cdots\rangle. \tag{2.45}$$

To compare the anyon chain model with conventional spin chains, we define local operators acting on the qutrit and qubit Hilbert spaces. Pauli operators $\sigma^\mu_{j+\frac{1}{2}}$ act on the qubit space as

$$\sigma^z_{j+\frac{1}{2}}|q_{j+\frac{1}{2}}\rangle = (-1)^{q_{j+\frac{1}{2}}}|q_{j+\frac{1}{2}}\rangle, \quad \sigma^x_{j+\frac{1}{2}}|q_{j+\frac{1}{2}}\rangle = |q_{j+\frac{1}{2}} + 1 \mod 2\rangle. \tag{2.46}$$

Similarly, we define the operators $X_j, Z_j$ and $\Gamma_j$ that act on the qutrit space at the $j^{\text{th}}$-site as

$$\begin{aligned}
X_j|p_j\rangle &= |p_j + 1 \mod 3\rangle, \\
Z_j|p_j\rangle &= \omega^{p_j}|p_j\rangle, \\
\Gamma_j|p_j\rangle &= \omega^{p_j}|-p_j \mod 3\rangle.
\end{aligned} \tag{2.47}$$

$\mathsf{Rep}(S_3)$ **non-invertible symmetry.** The choice of $\mathcal{M} = \mathsf{Vec}_{\mathbb{Z}_3}$ can be understood as having started from a model with $S_3$ global symmetry (which would correspond to choosing the regular module $\mathcal{M} = \mathsf{Vec}_{S_3}$) and having gauged $\mathbb{Z}_2^b = \{1, b\}$ [66]. Upon such a gauging one naturally obtains a dual $\mathbb{Z}_2$ symmetry in the gauged model whose generator we denote as $P$. Additionally, due to the group relation $bab = a^2$, the $S_3$ symmetry generators corresponding to the elements $a$ and $a^2$ combine into a non-invertible symmetry with quantum dimension 2, which we denote as $E$. These are precisely the irreducible representations of $S_3$: the sign representation $P$, the 2d representation $E$ and the trivial representation. The associated topological lines satisfy the $\mathsf{Rep}(S_3)$ fusion rules, which are dictated by the decomposition of the tensor product of representations into irreps

$$P \otimes P = 1, \qquad P \otimes E = E, \qquad E \otimes E = 1 \oplus P \oplus E. \tag{2.48}$$

**Twisted sector states.** Given a global symmetry, it is also natural to consider symmetry twisted state spaces. These correspond to state spaces where symmetry defects, i.e. lines in $\mathcal{S} = \mathcal{C}^*_\mathcal{M}$, end on the anyon chain from the bottom.

Since $P$ is a $\mathbb{Z}_2$ symmetry, it has a single end on the anyon chain, which we denote as $v_P$. Since $P$ descends from the identity line, upon gauging $\mathbb{Z}_2^b$, it acts trivially on the module degrees of freedom on the integer $j$ sites:

$$|\vec{p}, \vec{q}\rangle_P = \quad \raisebox{-1em}{} \quad . \tag{2.49}$$

In contrast, the $E$ line has quantum dimension= 2 endpoints, which we denote as $v_E^{(1)}$ and $v_E^{(2)}$. These two ends descend from the end-points of the $a$ and $a^2$ line respectively upon gauging $\mathbb{Z}_2^b$. These act on the module degrees of freedom as

$$|\vec{p}, \vec{q}\rangle_{(E,I)} = \quad \text{} \quad .\qquad(2.50)$$

$\mathsf{Rep}(S_3)$ **Action on States.** To derive the $\mathsf{Rep}(S_3)$ action on states we first ask how the symmetry defects are transported along the anyon chain. Being dual to the gauged $\mathbb{Z}_2^b$ symmetry, the end-point of $P$, i.e., $v_P$ is charged under the ends of the $b$ lines from above

$$\text{} \quad = \quad (-1)^{q_{j-\frac{1}{2}}} \times \text{}\qquad(2.51)$$

Transporting the $E$ symmetry lines past $\mathsf{Vec}_{S_3}$ lines ending on the anyon chain from above involves a non-trivial action, is consistent with the transport of $a$ and $a^2$ lines in the pre-gauged setup

$$\text{} \quad = \quad \text{} \quad , \quad \text{} \quad = \quad \text{} \quad , \qquad(2.52)$$

where $\widetilde{1} = 2$ and $\widetilde{2} = 1$. In the case of multiple twists one needs to be able to fuse the symmetry twists. Requiring consistency with transport, i.e. first transporting and then fusing must be the same as first fusing and then transporting imposes the constraint

$$v_P v_E^{(I)} = \sigma_I v_E^{(I)}, \qquad v_E^{(I)} v_P = \overline{\sigma}_I v_E^{(I)},\qquad(2.53)$$

such that $\sigma_1 \sigma_2 = \overline{\sigma}_1 \overline{\sigma}_2 = -1$. We pick the choice $\sigma_2 = \overline{\sigma}_1 = -1$, which implies that

$$\text{} \quad = \quad (-1) \times \text{} \quad = \quad (-1)^{I+1} \times \text{}\qquad(2.54)$$

Similarly, consistency between fusion and transport of the $E$ symmetry twists with various choices of endpoint vectors imposes

$$(2.55)$$

Naturally, a single symmetry defect may also split into two symmetry defects defining a "coproduct" structure on the symmetry defects. Following similar considerations as above one obtains the following splitting rules for the 1 and $P$ lines

$$(2.56)$$

The splitting rules for the $E$ lines are

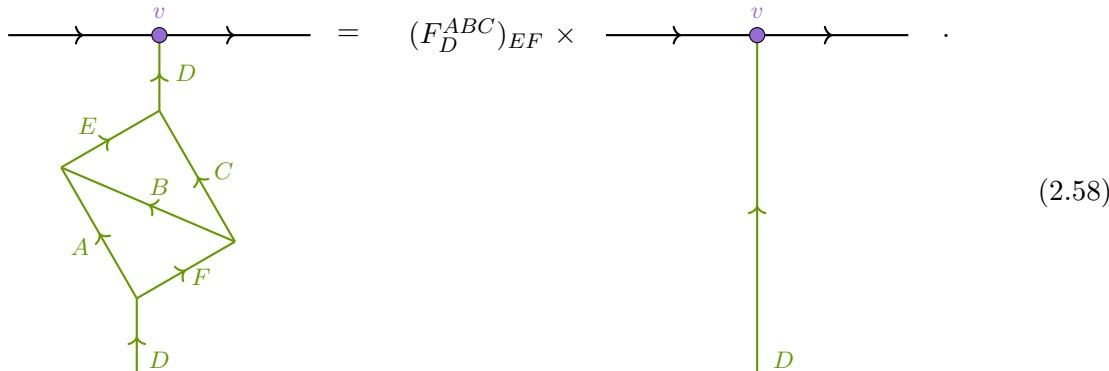

$$\tag{2.57}$$

Using this co-product and product structure, the associators in the $\mathsf{Rep}(S_3)$ fusion category can be computed by evaluating [2]

$$\tag{2.58}$$

With this preparation, we can now derive the symmetry action on states. We first consider the untwisted sector states. We denote the symmetry operators acting on the untwisted states as $\mathcal{U}_\Gamma$ where $\Gamma$ is a simple object in $\mathsf{Rep}(S_3)$

$$\tag{2.59}$$

$$\mathcal{U}_P|\vec{p}, \vec{q}\rangle = \cdots = \cdots$$

$$= (-1)^{\mathrm{hol}(q)} \times \cdots = (-1)^{\mathrm{hol}(q)}|\vec{p}, \vec{q}\rangle \,,$$

where $\mathrm{hol}(q) = \sum_j q_{j+\frac{1}{2}}$ and we have transported one of the ends all around the circle. In terms of the local operators acting on the anyon chain, the $P$ operator is simply

$$\mathcal{U}_P = \prod_j \sigma^z_{j+\frac{1}{2}} \, . \tag{2.60}$$

The $E$ symmetry action can similarly be derived using the coproduct $1 \to E \otimes E$ on the symmetry defects

Then transporting one of the defects around the anyon chain transforms the defect-end point $v_E^{(I)}$ to $v_E^{(\widetilde{I})}$ iff $\mathrm{hol}(q) = 1$. Since, there is no intertwiner from $v_E^{(I)} \otimes v_E^{(I)} \to v_1$, this symmetry operation contains all the states with $\mathrm{hol}(q) = 1$ in its kernel. Being an operation with a non-trivial kernel, this transformation cannot be implemented by a unitary operator. More precisely, the state $|\vec{p}, \vec{q}\rangle$ transforms as follows upon under the $E$ action

$$\mathcal{U}_E |\vec{p}, \vec{q}\rangle = \delta_{\mathrm{hol}(q),0} \Big[ |\vec{p}_1(\vec{q}), \vec{q}\rangle + |\vec{p}_2(\vec{q}), \vec{q}\rangle \Big] , \tag{2.61}$$

where

$$p_s(\vec{q})_j = p_j + s(-1)^{\sum_{i=0}^{j-1} q_{i+\frac{1}{2}}} \, . \tag{2.62}$$

Such an operation is implemented by the operator

$$U_E = \frac{1}{2} \Big( 1 + \prod_j \sigma^z_{j+\frac{1}{2}} \Big) (T_1 + T_2) \, , \tag{2.63}$$

where $T_s$ is an invertible operator that acts diagonally (in $\sigma^z_{j+\frac{1}{2}}$ basis) on the qubit degrees of freedom and transforms the qutrits according to the qubit eigen-state. It has the explicit form

$$T_s = \frac{1}{2} \prod_{j=1}^{L} \sum_{n=1,2} \Big[ \Big( 1 + (-1)^{n+1} \prod_{i=0}^{j-1} \sigma^z_{i+\frac{1}{2}} \Big) X_j^{ns} \Big] \, . \tag{2.64}$$

From the properties, $T_1 T_1 = T_2$, $T_2 T_2 = T_1$ and $T_1 T_2 = T_2 T_1 = 1$, it follows that

$$U_E \times U_E = 1 + U_P + U_E \, . \tag{2.65}$$

A feature of non-invertible symmetries is that they can map between untwisted and twisted sectors, unlike their invertible counterparts as described in section 2.1. For the case of $\mathsf{Rep}(S_3)$,

the untwisted sector states can map to the $P$ and $E$ twisted sector states via

$$\mathcal{U}_E(\Gamma)|\vec{p},\vec{q}\rangle = \quad \text{(2.66)}$$

where $\Gamma = P$ or $E$. For general fusion categories one may also have the freedom of specifying multiple morphisms at the trijunction of symmetry defects, however since there is a unique fusion channel for all fusions of simple objects in $\mathsf{Rep}(S_3)$, we suppress the dependence on morphisms. By following the same steps as outlined above

$$\mathcal{U}_E(P)|\vec{p},\vec{q}\rangle = \delta_{\mathrm{hol}(q),0}\left[|\vec{p}_1(\vec{q}),\vec{q}\rangle_P - |\vec{p}_2(\vec{q}),\vec{q}\rangle_P\right],$$
$$\mathcal{U}_E(E)|\vec{p},\vec{q}\rangle = \delta_{\mathrm{hol}(q),1}\left[|\vec{p}_1(\vec{q}),\vec{q}\rangle_{(E,1)} + |\vec{p}_2(\vec{q}),\vec{q}\rangle_{(E,2)}\right]. \quad \text{(2.67)}$$

Moving onto the $\mathsf{Rep}(S_3)$ symmetry action on the twisted sector states, a state in the $W$ twisted Hilbert space can be mapped to $Z$ twisted space by the action

$$\mathcal{U}_X(Y,Z)|\vec{p},\vec{q}\rangle_{(W,v)} = \quad \text{(2.68)}$$

As a simple example, consider the $P$ action on the $P$-twisted sector states

$$\mathcal{U}_P(1,P)|\vec{p},\vec{q}\rangle_P = \quad = (-1)^{\mathrm{hol}(q)} \times$$

$$= (-1)^{\mathrm{hol}(q)}|\vec{p},\vec{q}\rangle_P. \quad \text{(2.69)}$$

Similarly, the $P$-twisted states may be mapped to untwisted or $E$-twisted sectors under the $E$ symmetry action. The symmetry action from $P$-twisted to untwisted states via the action

of $E$ is computed as

$$\mathcal{U}_E(E\,,1)|\vec{p},\vec{q}\rangle_P = $$

$$= \delta_{\mathrm{hol}(q),0} \times \left\{ \quad + \quad \right\}$$

$$= \delta_{\mathrm{hol}(q),1} \times \left\{ \quad + \quad \right\}$$

$$= \delta_{\mathrm{hol}(q),0}\left\{ |\vec{p}_1(\vec{q}),\vec{q}\rangle - |\vec{p}_2(\vec{q}),\vec{q}\rangle \right\}. \tag{2.70}$$

Similarly,

$$\mathcal{U}_E(E\,,P)|\vec{p},\vec{q}\rangle_P = -\delta_{\mathrm{hol}(q),0}\left\{ |\vec{p}_1(\vec{q}),\vec{q}\rangle_P + |\vec{p}_2(\vec{q}),\vec{q}\rangle_P \right\},$$
$$\mathcal{U}_E(E\,,E)|\vec{p},\vec{q}\rangle_P = -\delta_{\mathrm{hol}(q),1}\left\{ |\vec{p}_1(\vec{q}),\vec{q}\rangle_{(E,1)} + |\vec{p}_2(\vec{q}),\vec{q}\rangle_{(E,2)} \right\}. \tag{2.71}$$

Finally, the $E$-twisted sector states transform as

$$\mathcal{U}_P(E\,,E)|\vec{p},\vec{q}\rangle_{(E,I)} = (-1)^{\mathrm{hol}(q)+1}|\vec{p},\vec{q}\rangle_{(E,I)}$$
$$\mathcal{U}_E(1\,,E)|\vec{p},\vec{q}\rangle_{(E,I)} = \delta_{\mathrm{hol}(q),0}|\vec{p}_I(\vec{q})\,,\vec{q}\rangle_{(E,I)} + \delta_{\mathrm{hol}(q),1}|\vec{p}_{\widetilde{I}}(\vec{q})\,,\vec{q}\rangle_{(E,\widetilde{I})}$$
$$\mathcal{U}_E(P\,,E)|\vec{p},\vec{q}\rangle_{(E,I)} = (-1)^I\left\{ \delta_{\mathrm{hol}(q),0}|\vec{p}_I(\vec{q})\,,\vec{q}\rangle_{(E,I)} + \delta_{\mathrm{hol}(q),1}|\vec{p}_{\widetilde{I}}(\vec{q})\,,\vec{q}\rangle_{(E,\widetilde{I})} \right\}$$
$$\mathcal{U}_E(E\,,1)|\vec{p},\vec{q}\rangle_{(E,I)} = \delta_{\mathrm{hol}(q),1}\left\{ \delta_{I,1}|\vec{p}_2(\vec{q})\,,\vec{q}\rangle + \delta_{I,2}|\vec{p}_1(\vec{q})\,,\vec{q}\rangle \right\}$$
$$\mathcal{U}_E(E\,,P)|\vec{p},\vec{q}\rangle_{(E,I)} = \delta_{\mathrm{hol}(q),1}\left\{ -\delta_{I,1}|\vec{p}_2(\vec{q})\,,\vec{q}\rangle_P + \delta_{I,2}|\vec{p}_1(\vec{q})\,,\vec{q}\rangle_P \right\}$$
$$\mathcal{U}_E(E\,,E)|\vec{p},\vec{q}\rangle_{(E,I)} = \delta_{\mathrm{hol}(q),0}\left\{ \delta_{I,1}|\vec{p}_2(\vec{q})\,,\vec{q}\rangle_{(E,1)} + \delta_{I,2}|\vec{p}_1(\vec{q})\,,\vec{q}\rangle_{(E,2)} \right\}. \tag{2.72}$$

$\mathsf{Rep}(S_3)$ **generalized charges.** Following the general theory described in section 2.1, the $\mathsf{Rep}(S_3)$ generalized charges are in one-to-one correspondence with bosonic lines in the SymTFT for $\mathsf{Rep}(S_3)$ which is $\mathcal{Z}(\mathsf{Vec}_{S_3}) \cong \mathcal{Z}(\mathsf{Rep}(S_3))$.

A simple object in $\mathcal{Z}(\mathsf{Vec}_{S_3})$ is labelled by a tuple

$$([g]\,,R_{[g]})\,, \tag{2.73}$$

where $[g]$ is a conjugacy class in $S_3$ and $R_{[g]}$ is an irreducible representation of the centralizer of (a chosen element in) $[g]$. There are three conjugacy classes in $S_3$

$$[\text{id}] = \{\text{id}\} , \quad [a] = \left\{a , a^2\right\} , \quad [b] = \left\{b , ab , a^2b\right\} . \tag{2.74}$$

The centralizers corresponding to these conjugacy classes are

$$H_{\text{id}} = S_3 , \quad H_a = \mathbb{Z}_3 = \left\{\text{id} , a , a^2\right\} , \quad H_b = \mathbb{Z}_2 = \{\text{id} , b\} . \tag{2.75}$$

Hence the simple objects in $\mathcal{Z}(\text{Vec}_{S_3})$ are labelled

$$
\begin{aligned}
&([\text{id}], 1) , \quad ([\text{id}], 1_-) , \quad ([\text{id}], E) , \\
&([a], 1) , \quad ([a], \omega) , \quad ([a], \omega^2) , \\
&([b], +) , \quad ([b], -) ,
\end{aligned}
\tag{2.76}
$$

where $1, \omega, \omega^2$ denote the three $\mathbb{Z}_3$ representations and $\pm$ denote the trivial and odd (sign) representation of $\mathbb{Z}_2^b$. Our setup involves the SymTFT along with two gapped boundary conditions and an interface between them. The two boundary conditions are given by the Lagrangian algebras in $\mathcal{Z}(\text{Vec}_{S_3})$

$$
\begin{aligned}
\mathfrak{B}_{\mathcal{C}}^{\text{inp}} &= ([\text{id}], 1) \oplus ([\text{id}], 1_-) \oplus 2([a], 1) , \\
\mathfrak{B}_{\mathcal{S}}^{\text{sym}} &= ([\text{id}], 1) \oplus ([a], 1) \oplus ([b], +) .
\end{aligned}
\tag{2.77}
$$

The fusion category of lines on the input and symmetry boundary are $\text{Vec}_{S_3}$ and $\text{Rep}(S_3)$ respectively. On the input boundary, the bulk line $([b], +)$ projects to the decomposable line $b \oplus ab \oplus a^2b$, while $([\text{id}], E)$ projects to $a \oplus a^2$. On the symmetry boundary, the bulk lines $([\text{id}], 1_-)$ and $([\text{id}], E)$ project to $P$ and $E \in \text{Rep}(S_3)$ respectively. The set of possible order parameters are given by the bosonic lines in the SymTFT which are

$$([\text{id}], 1) , \quad ([\text{id}], 1_-) , \quad ([\text{id}], E) , \quad ([a], 1) , \quad ([b], +) . \tag{2.78}$$

Corresponding to each of these lines, one obtains a mulitplet of operators that transform irreducibly under the action of $\text{Rep}(S_3)$. For simplicity, we only describe the action of the order parameters on the untwisted sector states. The identity line $([\text{id}], 1)$ corresponds to the identity operator while the charge line carrying the 1-dimensional representation $1_-$ is a symmetry twist/string operator that acts on states as

$$\mathcal{O}_{P,j} : |\vec{p}, \vec{q}\rangle \longrightarrow |\vec{p}, \vec{q}\rangle_P , \tag{2.79}$$

which follows from the fact that $Z_{\mathcal{C}}(1-) = 1$ and $Z_{\mathcal{S}}(1-) = P$ and $\beta_{1_-}$ is trivial. We have suppressed in our notation that the $P$ twist after the operator action in (2.79) is located at site $j$.

The SymTFT line $([\mathrm{id}], E)$ carrying the $E$-representation gives rise to a doublet of string operators that can be labelled by basis vectors $v_E^{(1)}, v_E^{(2)}$. Their action on states is similarly given by

$$\mathcal{O}_{E,j}^{(I)} : |\vec{p}, \vec{q}\rangle \longrightarrow |\vec{g}, g\rangle_{(E,I)} . \tag{2.80}$$

This follows from the fact that $Z_\mathcal{C}(E) = a \oplus a^2$, $Z_\mathcal{S}(E) = E$ and using the half-braiding from $m^p \otimes E \to a^I \otimes m^p$ picks the vector $v_E^{(I)}$ at the end of the $E$-symmetry twist line.

Next, the SymTFT line $([a], 1)$ has quantum dimension 2 and hence also corresponds to a doublet of operators. It follows from $(2.77)$ that $([a], 1)$ projects to the identity line on the input boundary. It however has two ends, or a two dimensional space of local operators. These local operators transform in the $E$-representation under the $\mathsf{Vec}_{S_3}$ symmetry on the input boundary. Compatibility with the $S_3$ action on the module degrees of freedom impose that we may pick the local operators to be $Z_j$ and $Z_j^2$. On the symmetry boundary, the line $([a], 1)$ projects to $1 \oplus P$. Therefore, we expect each local operator to be part of a multiplet which contains two operators—an operator $\mathcal{O}_{a_I,j}^{(+)}$ which acts within the untwisted sector and an operator $\mathcal{O}_{a_I,j}^{(-)}$ that maps between the untwisted and $P$-twisted sector. There is a compatibility condition that arises from the $\mathsf{Rep}(S_3)$ action on the multiplet (see section 5.2.2 of [29]) that takes the form

$$\mathcal{O}_{a_I,j}^{(+)} \mathcal{U}_E |\vec{p}, \vec{q}\rangle = \left\{ -\frac{1}{2} \mathcal{U}_E \mathcal{O}_{a_I,j}^{(+)} + \left(\omega + \frac{1}{2}\right) \mathcal{U}_E(E, 1) \mathcal{O}_{a_I,j}^{(-)} \right\} |\vec{p}, \vec{q}\rangle , \tag{2.81}$$

where $\omega = \exp\{2\pi i/3\}$. Let us pick $\mathcal{O}_{a_1,j}^{(+)} = Z_j$ and $\mathcal{O}_{a_1,j}^{(-)} |\vec{p}, \vec{q}\rangle = \alpha_1(p_j) |\vec{p}, \vec{q}\rangle$. Then $(2.81)$ imposes that

$$\omega^{p_j+1} = -\frac{1}{2} \omega^{p_j} + \left(\omega + \frac{1}{2}\right) \alpha_1(p_j) ,$$
$$\omega^{p_j+2} = -\frac{1}{2} \omega^{p_j} - \left(\omega + \frac{1}{2}\right) \alpha_1(p_j) , \tag{2.82}$$

which is solved by $\alpha_1(p_j) = \omega^{p_j}$. Therefore we obtain the first multiplet

$$\mathcal{O}_{a_1,j}^{(+)} = Z_j , \qquad \mathcal{O}_{a_1,j}^{(-)} = \mathcal{O}_{P,j} Z_j . \tag{2.83}$$

For the second multiplet, we may pick $\mathcal{O}_{a_2,j}^{(+)} = Z_j^2$ and $\mathcal{O}_{a_2,j}^{(-)} |\vec{p}, \vec{q}\rangle = \alpha_2(p_j) |\vec{p}, \vec{q}\rangle$. Again, Eq. $(2.81)$ imposes that

$$\omega^{2p_j+2} = -\frac{1}{2} \omega^{2p_j} + \left(\omega + \frac{1}{2}\right) \alpha_2(p_j) ,$$
$$\omega^{2p_j+1} = -\frac{1}{2} \omega^{2p_j} - \left(\omega + \frac{1}{2}\right) \alpha_2(p_j) , \tag{2.84}$$

which is solved by $\alpha_2(p_j) = -\omega^{2p_j}$. Hence we obtain the second multiplet corresponding to $([a], 1)$ to be

$$\mathcal{O}_{a_2,j}^{(+)} = Z_j^2 \,, \qquad \mathcal{O}_{a_2,j}^{(-)} = -\mathcal{O}_{P,j} Z_j^2 \,. \tag{2.85}$$

Finally, the SymTFT line $([b], 1)$ has quantum dimension 3 and corresponds to a triplet of operators. It follows from (2.77) that $Z_{\mathcal{C}}(b) = b \oplus ab \oplus a^2 b$ and $Z_{\mathcal{S}}(b) = 1 \oplus E$. All the projected lines on the input boundary implement a non-trivial morphism in $\mathrm{Hom}(\rho \otimes Z_{\mathcal{C}}(b), \rho)$ which is realized on the lattice model as $\sigma_{j+\frac{1}{2}}^x$. Depending on the choice of $Q_\nu^s$ one gets a single untwisted sector operator and two twisted sector operators which are

$$\begin{aligned} \mathcal{O}_{b,j} &= \sigma_{j+\frac{1}{2}}^x \,, \\ \mathcal{O}_{b,j}^{\pm} &= \left( \mathcal{O}_{E,j}^{(1)} \pm \mathcal{O}_{E,j}^{(2)} \right) \sigma_{j+\frac{1}{2}}^x \,. \end{aligned} \tag{2.86}$$

It can be checked that these order parameters satisfy the constraint (see section 5.2.4 in [29])

$$\mathcal{O}_{b,j} \mathcal{U}_E |\vec{p}, \vec{q}\rangle = \mathcal{U}_E(E, 1) \mathcal{O}_{b,j}^+ |\vec{p}, \vec{q}\rangle \,. \tag{2.87}$$

**Hamiltonian operators.** We study the $\mathsf{Rep}(S_3)$ symmetric anyon model on the parameter space spanned by Hamiltonian operators of the following type

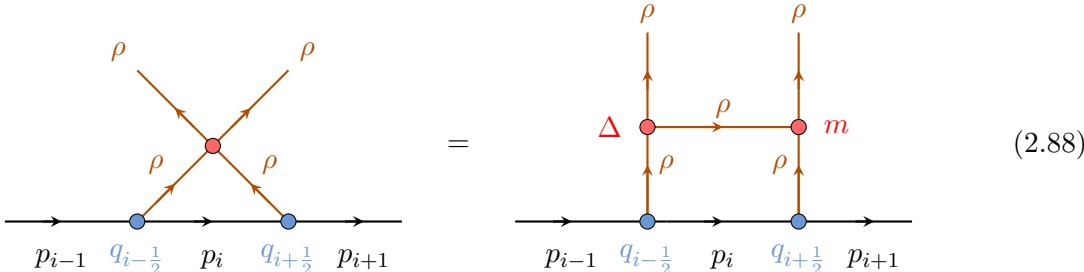

$$\tag{2.88}$$

where $\Delta : \rho \to \rho \otimes \rho$ and $m : \rho \to \rho$ such that $\Delta(g) = \sum_h h \otimes h^{-1} g$ and $m(g, h) = gh$. The building blocks of such operators are

$$\mathcal{O}_{(g_L, h, g_R), i} \quad = \tag{2.89}$$

which can be expressed as a product of three operators

$$\mathcal{O}_{(g_L, h, g_R), i} = P_{i-\frac{1}{2}}^{(g_L)} X_i^{(h)} P_{i+\frac{1}{2}}^{(g_R)} \,. \tag{2.90}$$

The operator $P^{(g)}_{i-\frac{1}{2}}$ is a projector that picks out the morphism corresponding to $g_L$. In terms of the qubit degrees of freedom on the half-integer sites, this projects onto the $q_{i-\frac{1}{2}} = \phi(g_L)$ state. Furthermore, for $g_L = a^p b^q$, this operator constrains the module (qutrit) degrees of freedom such that

$$\left[ p_{i-1} - (-1)^{q_{i-\frac{1}{2}}} p_i \right] \mod 3 = p. \tag{2.91}$$

On the spin chain, this projection operator is implemented as

$$P^{(a^p b^q)}_{i+\frac{1}{2}} = \frac{1}{6} \left[ 1 + (-1)^q \sigma^z_{i+\frac{1}{2}} \right] \left[ \sum_{n=0}^{2} \omega^{-pn} Z_i^n Z_{i+1}^{(2q-1)n} \right]. \tag{2.92}$$

The $X_i^{(h)}$ operator is also a three spin operator which acts on $(q_{i-\frac{1}{2}}, p_i, q_{i+\frac{1}{2}})$ as

$$X_i^{(a^p b^q)} : \quad q_{i \pm \frac{1}{2}} \longmapsto \quad q_{i \pm \frac{1}{2}} + q \quad \mod 2,$$

$$X_i^{(a^p b^q)} : \quad p_i \longmapsto (-1)^q p_i + p \mod 3. \tag{2.93}$$

On the spin chain, this operator is implemented as

$$X_i^{(a^p b^q)} = (X_i)^p \left( \sigma^x_{i-\frac{1}{2}} \Gamma_i \sigma^x_{i+\frac{1}{2}} \right)^q. \tag{2.94}$$

The general Hamiltonians which we consider have the form

$$\mathcal{H}[\lambda] = -\sum_{i} \sum_{g_L, g_R, h} \lambda_{(g_L, h, g_R)} \mathcal{O}_{(g_L, h, g_R), i}. \tag{2.95}$$

In fact, it can be checked that each of the operators $P^{(g)}_{i-\frac{1}{2}}$ and $X_i^{(g)}$ are individually $\mathsf{Rep}(S_3)$ symmetric and we may therefore also study a space of Hamiltonians with these operators instead of their product. Such Hamiltonians are more economical as they involve 3 spin interaction terms instead of 5 spin interaction. In [70], we study the phase diagram of the 3-spin models.

**Twisted sector Hamiltonian operators.** We can also define the Hamiltonians in the presence of symmetry defects as in (2.12)

$$\mathcal{O}^s_{(g_L, h, g_R), i} \quad = \quad$$

$$\tag{2.96}$$

Let us first consider the $P$-twisted Hamiltonian, such that $s = P$ and $\mu_t = v_P$. Acting with $h = a^p b$ now involves an additional minus sign as moving the $b$-line past the end of the $P$-defect picks up a minus sign. The presence of the $P$-defect does not modify the module degrees of freedom and therefore leaves the $P^{(g)}$ projection operator unaltered. To summarize, the Hamiltonian action in the presence of $P$ defect is obtained by making the modifications

$$X_1^{(g)} \longmapsto (-1)^{\phi(g)} X_1^{(h)} . \tag{2.97}$$

Next, we consider the $E$-twisted Hamiltonian action, i.e., we have $s = E$ and $\mu_t = v_E^{(I)}$. The Hamiltonian action has the following modifications. Firstly, in the $X^{(h)}$ action, if $\phi(h) = 1$, i.e., if $h = a^p b$, then moving the $h$ line past the defect implements $v_E^{(I)} \to v_E^{(\widetilde{I})}$. Secondly the constraint implemented by $P_{3/2}^{(g_R)}$ gets modified to

$$\left[ p_1 + I - (-1)^{q\frac{3}{2}} p_2 \right] \mod 3 = p \tag{2.98}$$

where we have used $g_R = a^p b^q$ and $p_t = p_1 + I$. Together these modifications can be summarized as

$$X_1^{(g)} \longmapsto X_1^{(g)} \left[ \sigma_t^x \right]^{\phi(g)} , \qquad Z_1 (Z_2)^\alpha \longmapsto Z_1 \omega^{\sigma_t^z} (Z_2)^\alpha , \tag{2.99}$$

where $\sigma_t^\mu$ are Pauli matrices acting on the impurity vector space at the end-point of the $E$ symmetry defect.

## 3 Gapped Phases

In the previous section, we described a large class of lattice models with fusion category symmetry $\mathcal{S}$. In this section, we identify special points in the parameter space of such lattice models which correspond to commuting projector Hamiltonians that can be fully solved and lead in the IR to gapped phases for $\mathcal{S}$. All the possible gapped phases for $\mathcal{S}$, as captured e.g. by the SymTFT, can be realized by such commuting projector Hamiltonians. We also describe local operators in these lattice models that are charged under $\mathcal{S}$ and condense in these gapped phases, thus serving as order parameters for the gapped phases. This includes operators that map untwisted sector of the model to symmetry twisted sectors, or in other words string order parameters. A hallmark of non-invertible symmetries is the existence of phases exhibiting both local (i.e. non-string) and string order parameters, as they are interchanged by the action of the symmetry.

## 3.1 General Setup

**SymTFT description of gapped phases.** Given a symmetry $\mathcal{S}$, one can ask what are the possible irreducible gapped phases in (1+1)d with symmetry $\mathcal{S}$. From the continuum perspective, such phases can be obtained by studying 2d TQFTs with $\mathcal{S}$ symmetry (and modding out by their continuous deformations) which describe the IR physics of the systems in these gapped phases. Such a 2d TQFT can be constructed by compactifying the 3d SymTFT $\mathfrak{Z}(\mathcal{S})$ on an interval with one end occupied by the topological boundary condition $\mathfrak{B}_{\mathcal{S}}^{\mathrm{sym}}$ (whose topological defects for the symmetry category $\mathcal{S}$), referred to as the symmetry boundary, and the other end occupied by another topological boundary condition $\mathfrak{B}^{\mathrm{phys}}$, referred to as the physical boundary, which may or may not be the same as $\mathfrak{B}_{\mathcal{S}}^{\mathrm{sym}}$. In this way $\mathcal{S}$-symmetric (1+1)d gapped phases correspond to topological boundary conditions $\mathfrak{B}^{\mathrm{phys}}$ of the SymTFT $\mathfrak{Z}(\mathcal{S})$. See [29] for more details.

**Converting SymTFT data into a lattice model for the gapped phase.** In this section, we provide an anyonic chain lattice model realizing the gapped phase associated to any $\mathfrak{B}^{\mathrm{phys}}$. We can begin with any input fusion category $\mathcal{C}$ and module category $\mathcal{M}$, such that $\mathcal{S} = \mathcal{C}_{\mathcal{M}}^{*}$. The boundary condition $\mathfrak{B}^{\mathrm{phys}}$ can be obtained from the input topological boundary condition $\mathfrak{B}_{\mathcal{C}}^{\mathrm{inp}}$ by gauging, either all of or some part of, the fusion category symmetry[2] $\mathcal{C}$ of the boundary $\mathfrak{B}_{\mathcal{C}}^{\mathrm{inp}}$. Such a gauging is specified by a Frobenius algebra $A$ in the fusion category $\mathcal{C}$ [2]. We choose $\rho$ to be the object underlying the algebra $A$

$$\rho = A \tag{3.1}$$

and the Hamiltonian morphism is chosen to be

$$h = \Delta \circ m \,, \tag{3.2}$$

where

$$m : \ \rho \otimes \rho \to \rho \tag{3.3}$$

is the algebra multiplication and

$$\Delta : \ \rho \to \rho \otimes \rho \tag{3.4}$$

---

[2]Note that this should not be confused with the symmetry $\mathcal{S}$ of the lattice model, which is the symmetry of the boundary $\mathfrak{B}_{\mathcal{S}}^{\mathrm{sym}}$.

is the co-multiplication for the Frobenius algebra. This can be represented diagrammatically as

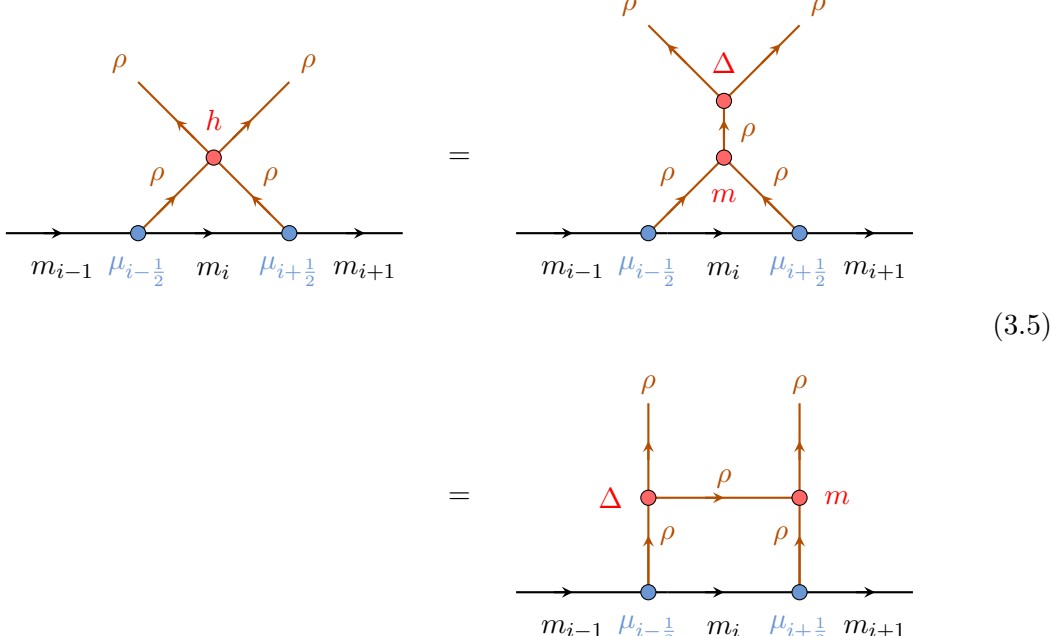

$$(3.5)$$

where at the bottom we have also rearranged it by using identities valid for a Frobenius algebra. This latter form will be useful to derive concrete expressions for $h$ in examples discussed later.

As a consequence of the Frobenius algebra identity

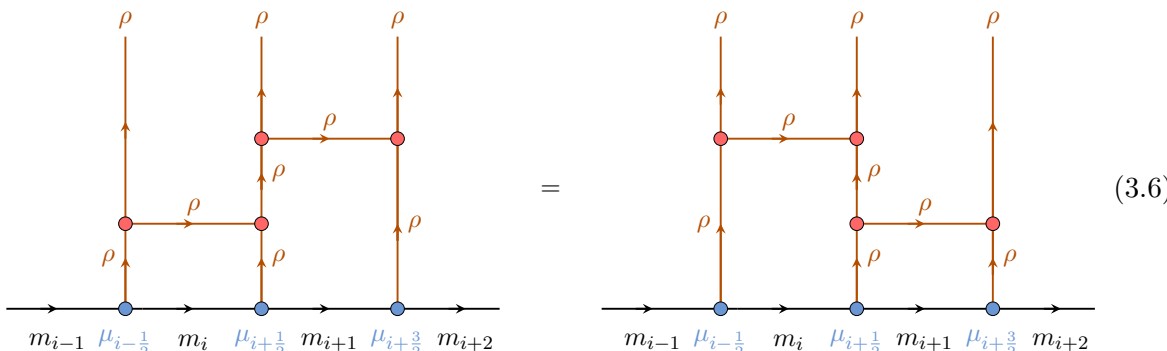

$$(3.6)$$

we learn that the Hamiltonian under consideration is a commuting projector Hamiltonian, and hence we can solve for its ground states easily.

**Ground States of the model.** In fact, the ground states can simply be identified with modules for the Frobenius algebra $\rho$, that we refer to as $\rho$-modules, in the module category $\mathcal{M}$. Such a module is a (not necessarily simple) object $m \in \mathcal{M}$ along with a morphism

$\mu \in \mathrm{Hom}(m, \rho \otimes m)$ satisfying the properties

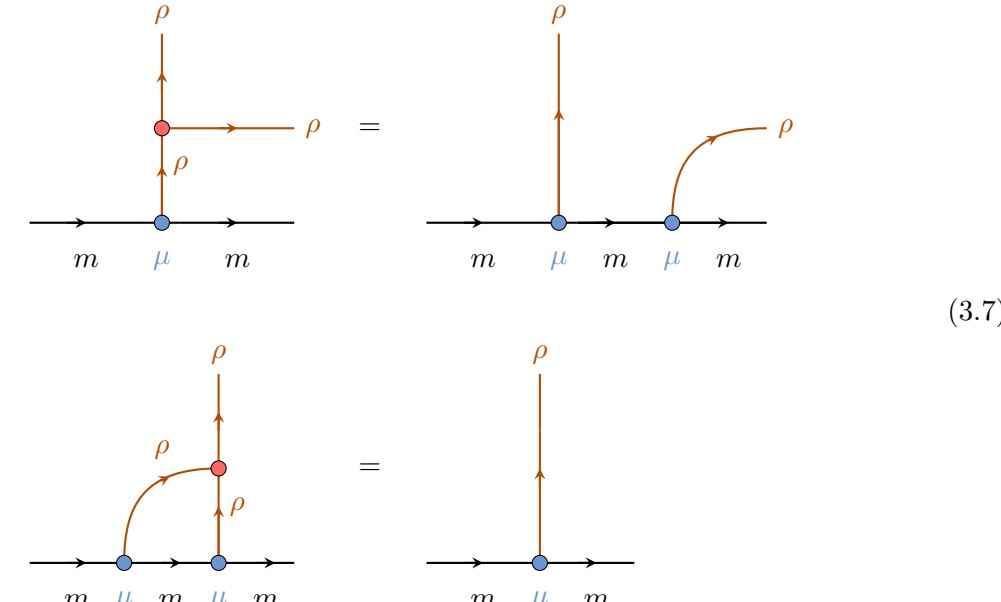

$$(3.7)$$

As a consequence we have

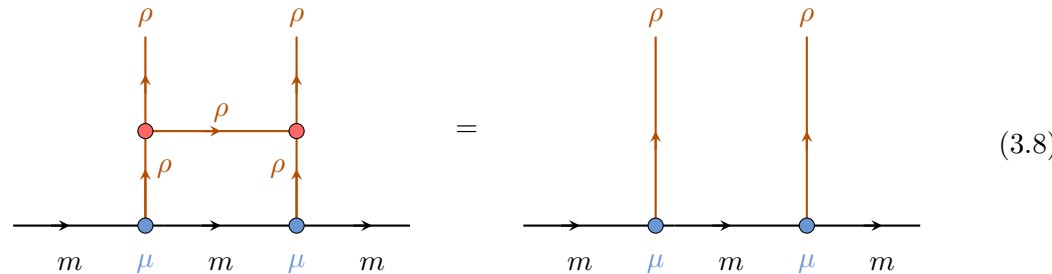

$$(3.8)$$

implying that a state constructed out of a $\rho$-module is a $+1$ eigenstate of all projectors and hence a ground state.

In fact, simple $\rho$-modules (which in general are not comprised of simple objects of $\mathcal{M}$) provide a canonical basis of the space of ground states. The ground states comprising this basis can be identified with **vacuum states**, i.e. these are states such that there do not exist any IR local operators that can map between them. We can argue this by contradiction as follows. If there exists such a local operator then we can act it on a ground state specified by a simple $\rho$-module $(m_1, \mu_1)$. The action leads to a new ground state in which a spatial segment $R$ is occupied by another $\rho$-module $(m_2, \mu_2)$ while the region on the left and right of $R$ is occupied by $(m_1, \mu_1)$. At a boundary between $(m_1, \mu_1)$ and $(m_2, \mu_2)$ regions, there must

be a local operator $\mathcal{O} \in \mathrm{Hom}(m_1, m_2)$

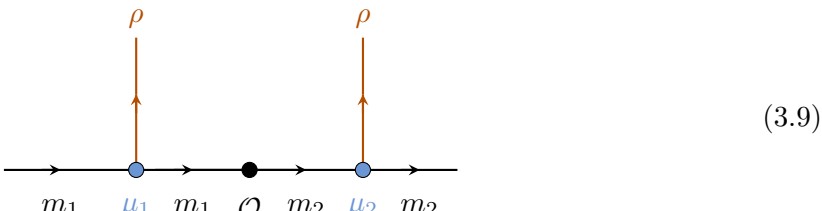

$$(3.9)$$

For this to be a ground state, $\mathcal{O}$ has to satisfy

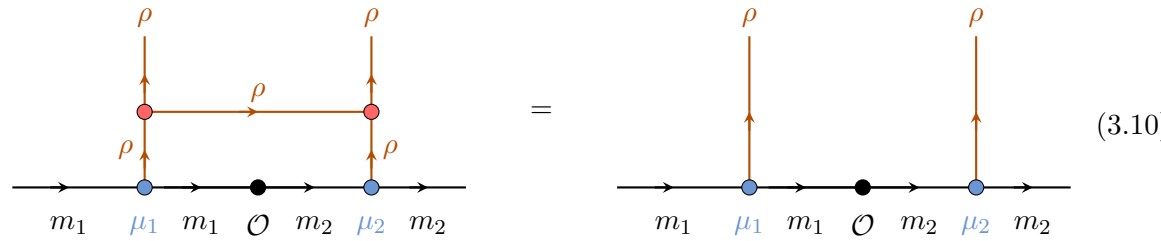

$$(3.10)$$

or equivalently

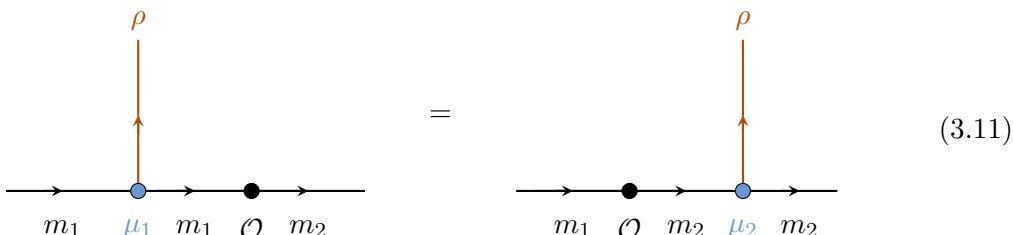

$$(3.11)$$

But this means $\mathcal{O}$ is a $\rho$-module morphism between the simple $\rho$-modules $(m_1, \mu_1)$ and $(m_2, \mu_2)$, which is a contradiction with the fact that these $\rho$-modules are simple.

**A degenerate non-tensor product decomposible case.** The $\rho$-modules form a module category $\mathcal{M}_\rho$ for the symmetry fusion category $\mathcal{S}$, which describes the action of the symmetry on vacuum states. The 2d TQFT describing the ground states of the above lattice model actually arises as another (degenerate) lattice model of the above type where we choose the input to be (we put primes to distinguish them from the $\mathcal{C}, \mathcal{M}, \rho$ being discussed above)

$$(\mathcal{C}', \mathcal{M}') = (\mathcal{S}^*_{\mathcal{M}_\rho}, \mathcal{M}_\rho) \tag{3.12}$$

and the Frobenius algebra to be the trivial one generated by the identity line

$$\rho' = 1 \,. \tag{3.13}$$

In this case every edge is constrained to carry the same simple object of the module category $\mathcal{M}'$ and hence the space of states is simply parametrized by the simple objects of $\mathcal{M}'$. The

Hamiltonian building block $h$ is the identity operator and hence every state is a ground state. Note that the input category $\mathcal{C}' = \mathcal{S}_{\mathcal{M}_\rho}^*$ can be identified as the fusion category formed by $\rho$-bimodules in $\mathcal{C}$, which is the category formed by topological line defects of the topological boundary condition $\mathfrak{B}^{\text{phys}}$ of the SymTFT $\mathfrak{Z}(\mathcal{S})$ featuring at the start of our discussion above equation (3.1).

**Symmetry twisted sectors.** The ground states in twisted sector for $s \in \mathcal{S}$ are parametrized by $(m, \mu, \mu_t)$ where $(m, \mu)$ is a simple $\rho$-module in $\mathcal{M}$, or in other words a simple object in $\mathcal{M}_\rho$, and

$$\mu_t \in \text{Hom}_{\mathcal{M}_\rho}\big((m, \mu) \otimes s, (m, \mu)\big) \tag{3.14}$$

is a basis morphism. If there are no such morphisms for some $(m, \mu)$, then there is no $s$-twisted sector ground state associated to that $\rho$-module. In terms of the original $(\mathcal{C}, \mathcal{M}, \rho)$, $\mu_t$ is a $\rho$-module morphism twisted by $s$, i.e. it is a morphism

$$\mu_t \in \text{Hom}_{\mathcal{M}}(m \otimes s, m) \tag{3.15}$$

satisfying the identity

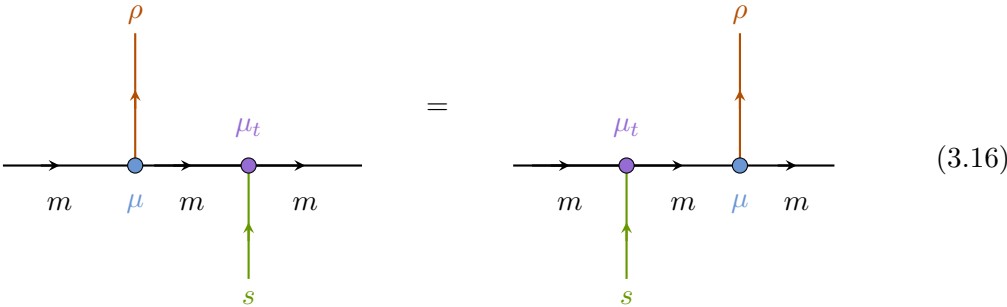

$$\tag{3.16}$$

**Condensed charges and Order parameters.** Some of the charged local operators are condensed in the gapped phase, which are referred to as order parameters for the gapped phase. As discussed in [29], the charges of the order parameters are encoded in a Lagrangian algebra $\mathcal{L}_{\text{phys}}$ in the Drinfeld center $\mathcal{Z}(\mathcal{S})$, which is associated to the physical boundary $\mathfrak{B}^{\text{phys}}$. The Lagrangian algebra takes the form

$$\mathcal{L}_{\text{phys}} = \bigoplus_a n_a \boldsymbol{Q}_a, \qquad n_a \in \mathbb{Z}_{\geq 0} \tag{3.17}$$

where $\boldsymbol{Q}_a$ are simple anyons of the SymTFT $\mathfrak{Z}(\mathcal{S})$, or in other words simple objects of the MTC $\mathcal{Z}(\mathcal{S})$. The relationship between $\mathcal{L}_{\text{phys}}$ and $\mathfrak{B}^{\text{phys}}$ is that the topological line $\boldsymbol{Q}_a$ of $\mathfrak{Z}(\mathcal{S})$ can end along $\mathfrak{B}^{\text{phys}}$ if $n_a \neq 0$, and $n_a$ is the dimension of the vector space of topological local

operators arising at such an end. In fact, $\boldsymbol{Q}_a$ for $n_a \neq 0$ are precisely the charges condensed in the gapped phase associated to $\mathfrak{B}^{\text{phys}}$. The number $n_a$ is the number of linearly independent multiplets of local operators acting on the IR 2d TQFT whose charge under $\mathcal{S}$ is given by $\boldsymbol{Q}_a$.

In the lattice models under consideration, multiplets of local operators carrying charge $\boldsymbol{Q}$ are parametrized by certain morphisms $\boldsymbol{Q}_\mu^\rho$ discussed around (2.14). Among these the multiplets that are condensed in the gapped phase associated to a Frobenius algebra $\rho$ are the ones satisfying the conditions

$$(3.18)$$

as the local operators in such multiplets map ground states to other ground states, rather than excited states. The local operators which map untwisted sector to untwisted sector may be referred to as conventional Landau-type order parameters, while the local operators which map untwisted sector to twisted sectors are referred to as **string order parameters**. A non-invertible symmetry may have both such local operators in the same multiplet, and hence a non-invertible symmetry can mix conventional and string order parameters.

For $\boldsymbol{Q} = \boldsymbol{Q}_a$, the condition (3.18) is solved precisely by $n_a$ linearly independent values of $\boldsymbol{Q}_\mu^\rho$. In fact, this is a gauging construction of topological ends of the bulk topological line operator $\boldsymbol{Q}_a$ along the boundary $\mathfrak{B}^{\text{phys}}$, when viewed as being obtained by $\rho$-gauging of the boundary $\mathfrak{B}_\mathcal{C}^{\text{inp}}$.

## 3.2 Abelian Group-like Symmetry

In this section, we describe the different gapped phases realized in systems with finite Abelian group symmetries.

### 3.2.1 $\mathbb{Z}_2$-symmetric Gapped Phases

As a simple example, we start by discussing the case of $\mathbb{Z}_2 = \{1, P\}$ symmetry. We pick $\mathcal{C} = \mathsf{Vec}_{\mathbb{Z}_2} = \{1, P\} = \mathcal{M} = \mathsf{Vec}_{\mathbb{Z}_2}$, so that $\mathcal{S} = \mathcal{C}_\mathcal{M}^* = \mathsf{Vec}_{\mathbb{Z}_2}$. As discussed in section 2.2, the Hilbert space of such a model decomposes into symmetry twisted sectors as

$$\mathcal{V} = \mathcal{V}_1 \oplus \mathcal{V}_P \,. \tag{3.19}$$

The untwisted sector $\mathcal{V}_1$ is spanned by states $|\vec{g}\rangle = |g_1, g_2, \ldots, g_L\rangle$ where $g_j = 0, 1$. The twisted sector state space is isomorphic and is spanned by basis states $|\vec{g}\rangle_P$. On each lattice site, there are the usual Pauli operators $\sigma^x$ and $\sigma^z$ which act as

$$\sigma_j^x |g_j\rangle = |g_j + 1 \bmod 2\rangle, \qquad \sigma^z |g_j\rangle = (-1)^{g_j} |q\rangle. \tag{3.20}$$

Additionally, there are twisted sector operators $\mathcal{T}$ which map between symmetry twisted sectors as

$$\mathcal{T}_j |\vec{g}\rangle = |\vec{g}\rangle_P, \qquad \mathcal{T}_j |\vec{g}\rangle_P = |\vec{g}\rangle_{P \otimes P} \cong |\vec{g}\rangle_1. \tag{3.21}$$

The SymTFT for $\mathbb{Z}_2$ symmetric systems is the Toric code or $\mathcal{Z}(\mathsf{Vec}_{\mathbb{Z}_2})$, which has topological lines $\{1, e, m, f = em\}$ of which $e$ and $m$ are Bosonic. The local operators corresponding to the Bosonic lines are

$$\mathcal{O}_{e,j} \simeq \sigma_j^z, \qquad \mathcal{O}_{m,j} \simeq \mathcal{T}_j. \tag{3.22}$$

The input and symmetry boundary in the SymTFT are chosen as

$$\mathfrak{B}^{\mathrm{inp}} = \mathfrak{B}^{\mathrm{phys}} = 1 \oplus e. \tag{3.23}$$

The fusion category of lines on this topological boundary is $\mathsf{Vec}_{\mathbb{Z}_2}$ such that the bulk lines $1$ and $e$ project to the identity while the bulk lines $m$ and $f$ project to the $\mathbb{Z}_2$ generator $P$ on the boundary.

We want to construct fixed-point Hamiltonians in each $\mathbb{Z}_2$-symmetric gapped phase. As described above in section 3.1, these are labelled by Frobenius algebras in $\mathsf{Vec}_{\mathbb{Z}_2}$. There are two choices of such algebras that are labelled by a subgroup $H \subseteq \mathbb{Z}_2$. Therefore we have two possibilities

$$A_1 = 1, \qquad A_{\mathbb{Z}_2} = 1 \oplus P. \tag{3.24}$$

We discuss them in turn.

**Trivial $\mathbb{Z}_2$ symmetric phase:** Consider the choice $A_{\mathbb{Z}_2}$, which corresponds to gauging the $\mathbb{Z}_2$ symmetry on the input boundary. Doing so delivers the physics boundary

$$\mathfrak{B}^{\mathrm{phys}} = \mathfrak{B}^{\mathrm{inp}} / A_{\mathbb{Z}_2} = 1 \oplus m. \tag{3.25}$$

The fixed-point Hamiltonian has the form

$$\mathcal{H}_{\mathbb{Z}_2} = -\frac{1}{2} \sum_j \sum_{h, h_L, h_R} \left[ \begin{array}{c} \includegraphics \end{array} \right]_j = -\frac{1}{2} \sum_j \left[ 1 + \sigma_j^x \right], \tag{3.26}$$

where $h, h_L, h_R \in \{1, P\}$. This Hamiltonian has a unique ground state which is the product state

$$|\text{GS}\rangle = \otimes_j |\sigma_j^x = 1\rangle = \frac{1}{2^{L/2}} \sum_{\vec{g}} |\vec{g}\rangle , \qquad (3.27)$$

with energy $E = -L$. Next, let us consider the Hamiltonian in the $P$-twisted sector. Since the presence of symmetry twists only enter the Hamiltonian via the projection operators in (2.36), this fixed-point Hamiltonian is unaltered by an $P$-twist. Consequently, one obtains a unique isomorphic $P$-twisted ground state

$$|\text{GS}\rangle_P = \frac{1}{2^{L/2}} \sum_{\vec{g}} |\vec{g}\rangle_m . \qquad (3.28)$$

Both the untwisted and $P$-twisted ground states are invariant under the $\mathbb{Z}_2$ symmetry action. The order parameters for this gapped phase correspond to the operators labelled by lines in $\mathfrak{B}^{\text{phys}}$. For the present case this is $\mathcal{O}_{m,j} \simeq \mathcal{T}_j$ which swaps the symmetry sector of the ground state

$$\mathcal{T}_j : |\text{GS}\rangle \longleftrightarrow |\text{GS}\rangle_P . \qquad (3.29)$$

$\mathbb{Z}_2$ **SSB Phase:** Now we consider the choice

$$A_1 = 1 , \qquad (3.30)$$

which corresponds to not gauging anything on the input boundary, therefore the input boundary becomes the physical boundary

$$\mathfrak{B}^{\text{phys}} = 1 \oplus e . \qquad (3.31)$$

The fixed-point Hamiltonian becomes

$$\mathcal{H}_1 = -\sum_j \left[ \begin{array}{c} \text{diagram} \end{array} \right]_j = -\sum_j \frac{1 + \sigma_{j-1}^z \sigma_j^z}{2} \cdot \frac{1 + \sigma_j^z \sigma_{j+1}^z}{2} . \qquad (3.32)$$

The expression in terms of the Pauli operators follows form the fact that this Hamiltonian simply enforces that the degrees of freedom at the sites $j-1, j$ and $j+1$ are the same. In other words, this Hamiltonian favors an ordering in the $\mathbb{Z}_2$ variables. Since all the building blocks of this Hamiltonian decompose into mutually commuting projectors, we can instead also study the simpler Hamiltonian

$$\widetilde{H}_1 = -\sum_j \frac{1 + \sigma_j^z \sigma_{j+1}^z}{2} , \qquad (3.33)$$

which is the usual ferromagnetic Hamiltonian (upto a shift). There are two ground states

$$\begin{aligned}
|\text{GS}, 1\rangle &= |\vec{1}\rangle = |1, 1, \ldots 1\rangle \\
|\text{GS}, P\rangle &= |\vec{P}\rangle = |P, P, \ldots P\rangle,
\end{aligned} \tag{3.34}$$

both with energy $E = -L$. The presence of a $P$-twist at a single site, say $j = L$ modifies a single term in the Hamiltonian. The $P$-twisted counterpart to the Hamiltonian (3.33) is

$$\mathcal{H}_1^{(P)} = -\sum_{j \neq L} \frac{1 + \sigma_j^z \sigma_{j+1}^z}{2} - \frac{1 - \sigma_L^z \sigma_1^z}{2} \tag{3.35}$$

It is easy to check that the lowest energies in this $P$-twisted sector are higher in energy as compared to the untwisted ground states. Therefore the true ground state space only contains the untwisted ground states. These are mapped into each other under the action of the symmetry $\mathcal{U}_P$

$$\mathcal{U}_P : |\text{GS}, 1\rangle \longleftrightarrow |\text{GS}, P\rangle. \tag{3.36}$$

The order parameter for this phase is expected to be $\mathcal{O}_{e,j} \simeq \sigma_j^z$ since the physical boundary is (3.31). This is indeed true since the ground state expectation value of this operator is

$$\langle \text{GS}, 1|\sigma_j^z|\text{GS}, 1\rangle = 1 \quad, \quad \langle \text{GS}, P|\sigma_j^z|\text{GS}, P\rangle = -1. \tag{3.37}$$

There are no non-trivial twisted sector ground states.

### 3.2.2  Gapped Phases for General Abelian $G$

We now slightly abstract from the previous example and discuss the case of general Abelian group $G$. Frobenius algebras in $\mathsf{Vec}_G$ are classified by tuples $(H, \beta)$ where $H$ is a subgroup of $G$ and $\beta \in H^2(H, U(1))$ [71]. We label the corresponding algebra as $A_{(H,\beta)}$. At the level of objects, $A_{(H,\beta)} = \oplus_{h \in H} h$, while the product structure $m : A \otimes A \to A$ and coproduct structure $\Delta : A \to A \otimes A$ in the algebra are determined by $\beta$ as

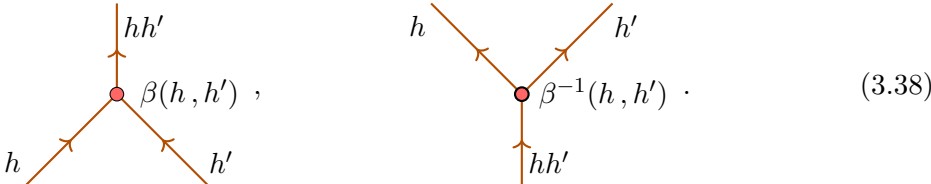

$$\tag{3.38}$$

Physically $(H, \beta)$ labels a gapped phase where the global symmetry is spontaneously broken to $H$ and each symmetry broken ground state realizes an $H$-symmetry protected topological (SPT) phase labelled by $\beta \in H^2(H, U(1))$ [72]. Let us describe the ground states of these fixed-point Hamiltonians.

We first consider a fixed-point Hamiltonian in the phase labelled by $(H, 1)$ i.e., corresponding to trivial 2-cocycle $\beta$. The Hamiltonian operator acts on a basis state as

$$= \frac{1}{|H|} \sum_{h \in H} \delta_{g_{j-1}^{-1} g_j, H} \delta_{g_j^{-1} g_{j+1}, H} \times \quad \cdots \quad hg_j \quad \cdots \tag{3.39}$$

where $\delta_{g,H}$ is 1 if $g \in H$ and 0 otherwise. It follows that there are $|G/H|$ ground states in the untwisted Hilbert space $\mathcal{V}_1$, each with energy $E = -L$, which have the form

$$|\text{GS}_{(H,1)}\,;[k]\rangle = \frac{1}{|H|^{L/2}} \prod_j \sum_{h_j \in H} |h_1 k, \ldots, h_L k\rangle, \tag{3.40}$$

with $[k]$ an $H$-coset and $k$ a representative element in $[k]$. Each ground state $|\text{GS}_{(H,1)}\,;[k]\rangle$ is invariant under $H \subseteq G$ while for $g \notin H$, the symmetry acts as

$$\mathcal{U}_g |\text{GS}_{(H,1)}\,;[k]\rangle = |\text{GS}_{(H,1)}\,;[gk]\rangle. \tag{3.41}$$

For non-trivial $\beta$, we consider a 2-site lattice to perform an explicit computation of the ground state. The Hilbert space is spanned by basis states $|g_1, g_2\rangle$ where

$$|g_1, g_2\rangle = \overset{g_1}{\underset{g_2}{\bigcirc}}. \tag{3.42}$$

We have not labelled the vertical lines since they are uniquely determined by choices of $g_1$ and $g_2$. Operators in $\mathcal{H}_{A_{(H,\beta)}}$ act as

$$\tag{3.43}$$

It follows that again there are $|G/H|$ ground states labelled by $H$-cosets $[k]$. The precise form of the ground states depends on $\beta$ as

$$|\text{GS}_{(H,\beta)}\,;[k]\rangle = \frac{1}{|H|} \sum_{h_1,h_2 \in H} \frac{\beta(h_1^{-1}, h_1)\beta(h_1 h_2^{-1}, h_2)}{\beta(h_2, h_1^{-1})} |h_1 k, h_2 k\rangle. \tag{3.44}$$

By using the cocycle condition

$$\delta\beta(h_1\,,h_2\,,h_3) \equiv \frac{\beta(h_1\,,h_2)\beta(h_1h_2\,,h_3)}{\beta(h_1\,,h_2h_3)\beta(h_2\,,h_3)} = 1\,, \tag{3.45}$$

it can be shown that in the untwisted sector, i.e., with periodic boundary conditions, the symmetry properties of the ground states are independent of $\beta$ such that

$$U_g|\mathrm{GS}_{(H,\beta)}\,;[k]\rangle = |\mathrm{GS}_{(H,\beta)}\,;[gk]\rangle\,. \tag{3.46}$$

The defining property of SPT ground states is that their ground state acquires a non-trivial charge when defined with symmetry twisted boundary conditions. Therefore, to detect the SPT, the ground states with symmetry twisted boundary conditions are required. The symmetry twisted Hilbert space is spanned by the basis $|g_1\,,g_2\rangle_g$ where

$$|g_1\,,g_2\rangle_g = \quad\text{}\;. \tag{3.47}$$

The Hamiltonian operators act on the twisted sector Hilbert space as

$$\tag{3.48}$$

The factor of $\delta_{g,H}$ implies that the ground states are in the $H$-twisted sectors and the lowest lying states in the $g \notin H$ twisted sectors have a finite energy gap of order 1. It follows from (3.48) that the twisted sector ground states take the form

$$|\mathrm{GS}_{(H,\beta)}\,;[k]\rangle_h = \frac{1}{|H|}\sum_{h_{1,2}} \frac{\beta(h_1^{-1}\,,h_1)\beta(h_1h_2^{-1}h\,,h_2)}{\beta(h_1\,,h)\beta(h_2\,,h_1^{-1})}|h_1k\,,h_2k\rangle_h\,. \tag{3.49}$$

By using the cocycle condition, we obtain the SPT property

$$U_{h_0}|\mathrm{GS}_{(H,\beta)}\,;[k]\rangle_h = \frac{\beta(h_0\,,h)}{\beta(h\,,h_0)}|\mathrm{GS}_{(H,\beta)}\,;[k]\rangle_h\,. \tag{3.50}$$

**Order parameters for gapped phases:** The order parameters for different gapped phases are constructed from operators on the lattice that are labelled by objects in the Drinfeld center $\mathcal{Z}(\mathsf{Vec}_G)$. For the present case of finite Abelian $G$, these labels are a tuple $(g, \alpha) \in G \times \mathsf{Rep}(G)$. Such a local operator acting on site $j$ of the lattice takes the form (2.37)

$$\mathcal{O}_{(g,\alpha),j} = \mathcal{T}_{g,j} Z_{\alpha,j} \, . \tag{3.51}$$

Specifically, a gapped phase labelled as $(H, \beta)$ has a non-vanishing expectation value of the set of order parameters

$$\left\{ \mathcal{O}_{(h,\beta_h \cdot \alpha_g),j} \mid h \in H \, , \ \beta_h(h') = \beta(h,h')/\beta(h',h) \, , \ \alpha_g|_H = 1 \right\} \tag{3.52}$$

where $\beta_h \in H^1(H, U(1))$ is obtained from $\beta \in H^2(H, U(1))$ via a slant product by $h$ and $\alpha_g \in \mathsf{Rep}(G/H) \subseteq \mathsf{Rep}(H)$ such that $\alpha_g(h) = 1$ for all $h \in H$. These order parameters reduce to usual symmetry breaking order parameters when we restrict to $h = 1$. When $h \neq 1$, $\alpha_g = 1$ and $\beta$ is trivial, these operators simply map between ground states in different twisted sectors. One can also consider the operator

$$\mathcal{O}_{(h,1),j} \, , \tag{3.53}$$

which is a symmetry twist operator extending at the site $j$ on the lattice. Finally, when $\beta$ in non-trivial, $\mathcal{O}_{(h,\beta_h,j}$ is the operator that maps between ground states in different twisted sectors. The operator

$$\mathcal{O}^\dagger_{(h,\beta_h),i} \mathcal{O}_{(h,\beta_h),j} \, , \tag{3.54}$$

is a string order parameter of the SPT [73–75], which is a finite symmetry string corresponding to $h \in H$ appended with charges $\beta_h$ and $\beta_h^{-1}$ at the two ends located at site $j$ and $i$ respectively.

## 3.3 $\mathsf{Rep}(S_3)$-symmetric Gapped Phases

In this section we study the $\mathsf{Rep}(S_3)$ gapped phases as realized in the anyon chain model described in section 2.3. On general grounds [29], we know that there are four gapped phases, which can be characterized as the trivial phase, the $\mathbb{Z}_2$ SSB, $\mathsf{Rep}(S_3)$ SSB and $\mathsf{Rep}(S_3)/\mathbb{Z}_2$ SSB. Here we will construct the UV lattice models and determine also the order parameters for each vacuum and the action of the non-invertible symmetry.

**Hamiltonian for gapped phases.** For each gapped phase, we describe a fixed-point or commuting projector Hamiltonian and study the characteristic properties of the gapped phase such as the symmetry action on the multiplet of ground states and the structure of order parameters. Recall that the input boundary and symmetry boundary in the SymTFT associated

with the anyon chain model are

$$\mathfrak{B}_{\mathcal{C}}^{\text{inp}} = ([\text{id}], 1) \oplus ([\text{id}], 1_-) \oplus 2([a], 1) \,,$$

$$\mathfrak{B}_{\mathcal{S}}^{\text{sym}} = ([\text{id}], 1) \oplus ([a], 1) \oplus ([b], +) \,. \tag{3.55}$$

Correspondingly, the fusion category of lines on the input boundary is $\mathcal{C} = \mathsf{Vec}_{S_3}$ while the lines on the symmetry boundary are $\mathcal{S} = \mathsf{Rep}(S_3)$. In the SymTFT different infra-red gapped phases are in correspondence with topological boundary conditions on the physical boundary $\mathfrak{B}^{\text{phys}}$. Different topological $\mathfrak{B}^{\text{phys}}$ are obtained from $\mathfrak{B}^{\text{inp}}$ by gauging some or a part of $\mathcal{C}$. Each such gauging in turn corresponds to picking a Frobenius algebra $A$ in $\mathcal{C}$ and restricting the Hamiltonian to $A$.

For the present case, Frobenius algebras in $\mathcal{C} = \mathsf{Vec}_{S_3}$ are labelled by subgroups $H$ of $S_3$. We denote the Frobenius algebra corresponding to $H \subseteq S_3$ as $A_H$ such that

$$A_H = \bigoplus_{h \in H} h \,. \tag{3.56}$$

The fixed-point Hamiltonians are simply

$$\mathcal{H}_H = -\frac{1}{|H|} \sum_i \sum_{h_L, h, h_R \in H} \mathcal{O}_{(h_L, h, h_R), i} \,, \tag{3.57}$$

where

$$\mathcal{O}_{(h_L, h, h_R), j} \quad = \quad$$

$$\tag{3.58}$$

Let us now describe the different gapped phases by choosing different subgroups $H$.

$\mathsf{Rep}(S_3)$ **Trivial phase.** The trivial phase sometimes also referred to as the paramagnetic or disordered phase has a single untwisted sector ground state that is $\mathsf{Rep}(S_3)$ invariant. This phase corresponds to the case with physical boundary

$$\mathfrak{B}^{\text{phys}} = ([\text{id}], 1) \oplus ([\text{id}], 1) \oplus 2([\text{id}], E) \,, \tag{3.59}$$

which can be obtained from $\mathfrak{B}^{\text{inp}}$ by gauging the $\mathbb{Z}_3$ symmetry generated by $a$. Therefore we choose

$$A_{\mathbb{Z}_3} = 1 \oplus a \oplus a^2 \,. \tag{3.60}$$

Correspondingly, the Hamiltonian implements that the ground state lies in the subspace with all the qubit degrees of fixed to be $\phi(h) = 0$. However the qutrit or module degrees of freedom are summed over due to the action of $X_h$. More precisely, using (2.92) and (2.94), the Hamiltonian operator at site $i$ for this fixed-point Hamiltonian becomes

$$\sum_{h_L,h,h_R \in H} \mathcal{O}_{(h_L,h,h_R),i} = \frac{1 + \sigma^z_{i-\frac{1}{2}}}{2} \cdot \frac{1 + X_i + X_i^2}{3} \cdot \frac{1 + \sigma^z_{i-\frac{1}{2}}}{2}. \tag{3.61}$$

Correspondingly the untwisted sector ground-state is

$$|\text{GS}\rangle_1 = \bigotimes_j \left| X_j = 1 \,, \sigma^z_{j+\frac{1}{2}} = 1 \right\rangle = \frac{1}{3^{L/2}} \sum_{\vec{p}} |\vec{p}, \vec{0}\rangle. \tag{3.62}$$

Next, consider the $P$-twisted sector. Recall that the presence of a $P$-twist defect, only alters the operators $X^{(a^p b)}$ which do not appear in the Hamiltonian. Since the trivial phase fixed-point Hamiltonian does not include operators with $h = a^p b$, the untwisted and $P$-twisted Hamiltonians and their ground states are isomorphic.

$$|\text{GS}\rangle_P = \bigotimes_j \left| X_j = 1 \,, \sigma^z_{j+\frac{1}{2}} = 1 \right\rangle_P = \frac{1}{3^{L/2}} \sum_{\vec{p}} |\vec{p}, \vec{0}\rangle_P. \tag{3.63}$$

Using (2.99), it follows that the $E$-twisted Hamiltonians also remain form invariant for this choice of $H$. Therefore we obtain two $E$-twisted ground states, one for each choice of vector $v_E^{(I)}$ at the end of the $E$-line

$$|\text{GS}\rangle_{(E,I)} = \bigotimes_j \left| X_j = 1 \,, \sigma^z_{j+\frac{1}{2}} = 1 \right\rangle_{(E,I)} = \frac{1}{3^{L/2}} \sum_{\vec{p}} |\vec{p}, \vec{0}\rangle_{(E,I)}. \tag{3.64}$$

Consequently, there are $\dim(\Gamma)$ ground states for each $\Gamma \in \mathsf{Rep}(S_3)$ twisted sector. The $\mathsf{Rep}(S_3)$ action on this multiplet can be straightforwardly computed using the procedure described in section 2.3. On the untwisted and $P$-twisted sector it takes the form (using (2.69) and (2.71))

$$\mathcal{U}_\Gamma |\text{GS}\rangle_1 = \dim(\Gamma)|\text{GS}\rangle_1 \,,$$
$$\mathcal{U}_P(1, P)|\text{GS}\rangle_P = |\text{GS}\rangle_P \,, \tag{3.65}$$
$$\mathcal{U}_E(E, P)|\text{GS}\rangle_P = -2|\text{GS}\rangle_P \,,$$

while on the $E$-twisted sector ground states (using (2.72))

$$\mathcal{U}_P(E, E)|\text{GS}\rangle_{(E,I)} = -|\text{GS}\rangle_{(E,I)} \,,$$
$$\mathcal{U}_E(P, E)|\text{GS}\rangle_{(E,I)} = -|\text{GS}\rangle_{(E,I)} \,, \tag{3.66}$$
$$\mathcal{U}_E(X, E)|\text{GS}\rangle_{(E,I)} = +|\text{GS}\rangle_{(E,I)} \,, \quad X = 1 \,, E \,.$$

Note that no two distinct twisted sector ground states map into each other under $\mathsf{Rep}(S_3)$ action. The order parameters for the trivial phase correspond to the lines in the Lagrangian

algebra corresponding to $\mathfrak{B}^{\text{phys}}$ which are $\mathcal{O}_{P,i}$ and $\mathcal{O}_{E,i}^{(I)}$. Using (2.79), these transform the untwisted sector ground state as

$$\mathcal{O}_{P,i}|\text{GS}\rangle_1 = |\text{GS}\rangle_P\,,$$
$$\mathcal{O}_{E,i}^{(I)}|\text{GS}\rangle_1 = |\text{GS}\rangle_{(E,I)}\,.$$

(3.67)

Meanwhile the order parameters corresponding to the SymTFT lines $([a],1)$ and $([b],+)$ do not act within the multiplet of ground states i.e.,

$$_{(\Gamma_1,v_1)}\langle\text{GS}|\mathcal{O}_{a,j}^\pm|\text{GS}\rangle_{(\Gamma_2,v_2)} = 0\,,$$
$$_{(\Gamma_1,v_1)}\langle\text{GS}|\mathcal{O}_{b,j}|\text{GS}\rangle_{(\Gamma_2,v_2)} = 0\,,$$
$$_{(\Gamma_1,v_1)}\langle\text{GS}|\mathcal{O}_{b,j}^\pm|\text{GS}\rangle_{(\Gamma_2,v_2)} = 0\,.$$

(3.68)

$\mathbb{Z}_2$ **SSB phase.** Next we consider the gapped phase obtained by gauging the full $S_3$ symmetry on the physical boundary. Therefore we choose

$$A_{S_3} = \bigoplus_{h\in S_3} h\,.$$

(3.69)

Upon such a gauging, the physical boundary becomes

$$\mathfrak{B}^{\text{phys}} = ([\text{id}],1) \oplus ([b],+) \oplus ([\text{id}],E)\,.$$

(3.70)

Since, we allow all morphisms on the half-integer sites, a priori there are no constraints on the qubit degrees of freedom. The fixed-point Hamiltonian comprises of operators

$$H_{S_3} = -\frac{1}{|S_3|}\sum_j \sum_{h_L,h,h_R\in S_3} \mathcal{O}_{(h_L,h,h_R),j}$$
$$= -\sum_j \frac{1}{6}\left(1+X_j+X_j^2\right)\left(1+\sigma_{j-\frac{1}{2}}^x \Gamma_j \sigma_{j+\frac{1}{2}}^x\right)\,.$$

(3.71)

The terms in the two brackets mutually commute, and can be simultaneously diagonalized. We first project to the $X_j = 1$ subspace, i.e., the $+1$ eigenspace of the first bracket. Then noting that $\Gamma_j$ acts as the identity on this space, we obtain two ground states corresponding to $\sigma_{i+\frac{1}{2}}^x = \pm1$.

$$|\text{GS},\pm\rangle_1 = \bigotimes_j \left|X_j=1,\sigma_{j+\frac{1}{2}}^x=\pm1\right\rangle = \frac{1}{6^{L/2}}\sum_{\vec{p},\vec{q}}(\pm1)^{\text{hol}(q)}|\vec{p},\vec{q}\rangle\,.$$

(3.72)

Next consider the $P$ twisted Hamiltonian which is given by

$$H_{S_3}^{(P)} = -\frac{1}{|S_3|}\sum_{h_L,h,h_R\in S_3}\left[\mathcal{O}_{(h_L,h,h_R),1}^P + \sum_{j\neq1}\mathcal{O}_{(h_L,h,h_R),j}\right]$$
$$= -\frac{1}{6}\left[\left(1+X_1+X_1^2\right)\left(1-\sigma_{\frac{1}{2}}^x\Gamma_1\sigma_{\frac{3}{2}}^x\right) - \sum_{j\neq1}\left(1+X_j+X_j^2\right)\left(1+\sigma_{j-\frac{1}{2}}^x\Gamma_j\sigma_{j+\frac{1}{2}}^x\right)\right]\,.$$

(3.73)

We require the $P$-twisted ground state $|\mathrm{GS}\rangle_P$ to satisfy

$$\mathcal{O}^P|\mathrm{GS}\rangle_P \equiv \mathcal{O}^P_{(h_L,h,h_R),1}\prod_{j\neq 1}\mathcal{O}_{(h_L,h,h_R),j}|\mathrm{GS}\rangle_P = |\mathrm{GS}\rangle_P\,. \tag{3.74}$$

For any basis state $|\vec{p},\vec{q}\rangle_P$, the state $\mathcal{O}^P|\vec{p},\vec{q}\rangle_P$ only contains basis states $|\vec{p}\,',\vec{q}\,'\rangle$ with $\mathrm{hol}(q) = \mathrm{hol}(q')$. In other words $\mathrm{hol}(q)$ is preserved under the Hamiltonian action. There are $6^L/2$ basis states with a fixed $\mathrm{hol}(q)$ while $\mathcal{O}^P$ is a sum of $6^L$ operators. In fact, $|\vec{p}\,',\vec{q}\,'\rangle$ appears twice with opposite signs in $\mathcal{O}^P|\vec{p},\vec{q}\rangle_P$ and consequently

$$\mathcal{O}^P|\vec{p},\vec{q}\rangle_P = 0\,, \tag{3.75}$$

Therefore there are no $P$-twisted sector ground states in this gapped phase phase. Equivalently, the lowest energy eigenstates in the $P$-twisted sector are higher in energy as compared with the untwisted sector ground states and therefore do not participate in the infra red physics.

The $E$ twisted Hamiltonian is given by

$$
\begin{aligned}
H^{(E)}_{S_3} &= -\frac{1}{|S_3|}\sum_{h_L,h,h_R\in S_3}\left[\mathcal{O}^E_{(h_L,h,h_R),1}+\sum_{j\neq 1}\mathcal{O}_{(h_L,h,h_R),j}\right] \\
&= -\frac{1}{6}\left[\left(1+X_1+X_1^2\right)\left(1+\sigma^x_t\sigma^x_{\frac{1}{2}}\Gamma_1\sigma^x_{\frac{3}{2}}\right)-\sum_{j\neq 1}\left(1+X_j+X_j^2\right)\left(1+\sigma^x_{j-\frac{1}{2}}\Gamma_j\sigma^x_{j+\frac{1}{2}}\right)\right].
\end{aligned}
\tag{3.76}
$$

We require the $E$-twisted ground state $|\mathrm{GS}\rangle_E$ to satisfy

$$\mathcal{O}^E_{(h_L,h,h_R),1}|\mathrm{GS}\rangle_E = \mathcal{O}_{(h_L,h,h_R),j\neq 1}|\mathrm{GS}\rangle_E = |\mathrm{GS}\rangle_E\,. \tag{3.77}$$

There are two states that satisfy this requirement which are the two $E$ twisted sector ground states

$$|\mathrm{GS},q_0\rangle_E = \frac{1}{6^{L/2}}\sum_I\sum_{\vec{p},\vec{q}}\delta_{\mathrm{hol}(q),q_0}|\vec{p},\vec{q}\rangle_{(E,I)}\,. \tag{3.78}$$

To summarize there are four ground states

$$\left\{|\mathrm{GS},+\rangle_1\,,|\mathrm{GS},-\rangle_1\,,|\mathrm{GS},q_0=0\rangle_E\,,\ |\mathrm{GS},q_0=1\rangle_E\right\}. \tag{3.79}$$

On the untwisted states, the $\mathsf{Rep}(S_3)$ symmetry lines act

$$\mathcal{U}_P|\mathrm{GS},\pm\rangle_1 = |\mathrm{GS},\mp\rangle_1\,,\qquad \mathcal{U}_E|\mathrm{GS},\pm\rangle_1 = |\mathrm{GS},+\rangle_1+|\mathrm{GS},-\rangle_1\,, \tag{3.80}$$

which satisfy the $\mathsf{Rep}(S_3)$ fusion rules. Since the $\mathcal{U}_P$ symmetry operator which generates the $\mathbb{Z}_2\in\mathsf{Rep}(S_3)$ exchanges the two ground states, we refer to this phase as the $\mathbb{Z}_2$ SSB phase.

Next, consider the $\mathsf{Rep}(S_3)$ action that maps between the twisted and untwisted sector ground states. Using (2.67) and (2.72),

$$\mathcal{U}_E(E)|\mathrm{GS},\pm\rangle_1 = \pm|\mathrm{GS},1\rangle_E \,,$$
$$\mathcal{U}_E(E,1)|\mathrm{GS},0\rangle_E = 0 \,, \tag{3.81}$$
$$\mathcal{U}_E(E,1)|\mathrm{GS},1\rangle_E = |\mathrm{GS},+\rangle_1 - |\mathrm{GS},-\rangle_1 \,.$$

Let us now discuss the order parameters. We expect $\mathcal{O}_{E,j}^{(I)}$, $\mathcal{O}_{b,j}$ and $\mathcal{O}_{b,j}^{\pm}$ to play the role of order parameters as these correspond to lines condensed on the physical boundary in (3.70). The ground states have the following transformation properties under these order parameters

$$\mathcal{O}_{b,j}|\mathrm{GS},\pm\rangle_1 = \pm|\mathrm{GS},\pm\rangle_1 \,,$$
$$\mathcal{O}_{b,j}|\mathrm{GS},s\rangle_E = |\mathrm{GS},s\rangle_E \,,$$
$$\mathcal{O}_{b,j}^{+}|\mathrm{GS},\pm\rangle_1 = \pm\Big[|\mathrm{GS},0\rangle_E + |\mathrm{GS},1\rangle_E\Big] \,, \tag{3.82}$$
$$(\mathcal{O}_{E,j}^{(1)} + \mathcal{O}_{E,j}^{(2)})|\mathrm{GS},\pm\rangle_1 = |\mathrm{GS},0\rangle_E \pm |\mathrm{GS},1\rangle_E \,.$$

The other order parameters, i.e., those corresponding to the SymTFT lines $([a],1)$ and $([\mathrm{id}],1_-)$ are 0 when projected to the ground state multiplet.

**$\mathsf{Rep}(S_3)/\mathbb{Z}_2$ SSB.** Now we move onto the gapped phase obtained by setting $\mathfrak{B}^{\mathrm{phys}} = \mathfrak{B}^{\mathrm{inp}}$. Hence we choose

$$A_{\mathbb{Z}_1} = 1 \,. \tag{3.83}$$

The corresponding commuting projector Hamiltonian acting on the untwisted Hilbert space is

$$H_{\mathbb{Z}_1} = -\sum_j \mathcal{O}_{(1,1,1),j} = -\sum_j P_{j-\frac{1}{2}}^{(1)} P_{j+\frac{1}{2}}^{(1)} \,, \tag{3.84}$$

where $P_{j+1/2}^{(1)}$ is defined in (2.92). There are clearly three untwisted sector ground states

$$|\mathrm{GS},n\rangle_1 = \bigotimes_j \left| Z_j = e^{\frac{2\pi i n}{3}} , \sigma_{j+\frac{1}{2}}^z = 1 \right\rangle = \left| \vec{n}, \vec{0} \right\rangle \,. \tag{3.85}$$

All three vacua are left invariant under the action of $P$. However under the $E$ action, they transform as

$$\mathcal{U}_E|\mathrm{GS},n\rangle_1 = |\mathrm{GS},n+1 \bmod 3\rangle_1 + |\mathrm{GS},n+2 \bmod 3\rangle_1 \,. \tag{3.86}$$

Since $P$ acts trivially while $E$ permutes the the three vacua, this gapped phase was referred to as the $\mathsf{Rep}(S_3)/\mathbb{Z}_2$ SSB phase in [29].

Next consider the symmetry twisted sectors. Since the $P$ symmetry twist only alters the operators $X_j^{(a^p b)}$ which do not appear in the Hamiltonian (3.84), this Hamiltonian remains

form invariant in the presence of a $P$ symmetry twist. The corresponding ground states are also isomorphic to their untwisted sector counter parts and we denote them as

$$|\text{GS}, n\rangle_P = \left|\vec{n}, \vec{0}\right\rangle_P. \tag{3.87}$$

The presence of an $E$ twist modifies the Hamiltonian following (2.99) such that $P_{\frac{3}{2}}^{(1)\,3}$ enforces the condition (see (2.98))

$$p_1 + 1 = p_2 \bmod 3, \tag{3.88}$$

where the end-point of the $E$ symmetry defect is taken to be $v_E^{(I)}$. Since it is not possible to simultaneously satisfy both (3.88) and $p_j = p_{j+1} \bmod 3$ for all $j \neq 1$ together, there are no $E$-twisted ground states which have the same energy as the untwisted and $P$-twisted sector ground states. For this reason, the $E$-twisted states do not appear in the infra-red theory.

The $\mathsf{Rep}(S_3)$ symmetry action map between the untwisted and $P$ twisted sector ground states (using (2.67) and (2.70)) as

$$\mathcal{U}_E(P)|\text{GS}, n\rangle_1 = |\text{GS}, n+1 \bmod 3\rangle_P - |\text{GS}, n+2 \bmod 3\rangle_P,$$
$$\mathcal{U}_E(E, 1)|\text{GS}, n\rangle_P = |\text{GS}, n+1 \bmod 3\rangle_1 - |\text{GS}, n+2 \bmod 3\rangle_1. \tag{3.89}$$

Similarly, using (2.69) and (2.71), the action of $P$ and $E$ symmetry lines map within the $P$-twisted ground states as

$$\mathcal{U}_P(1, P)|\text{GS}, n\rangle_P = |\text{GS}, n\rangle_P,$$
$$\mathcal{U}_E(E, P)|\text{GS}, n\rangle_P = -|\text{GS}, n+1 \bmod 3\rangle_P - |\text{GS}, n+2 \bmod 3\rangle_P. \tag{3.90}$$

The order parameters for this phase are the multiplets $\mathcal{O}_{a_I,j}^{(\pm)}$ (for $I = 1, 2$) and $\mathcal{O}_{P,j}$ which correspond to the SymTFT lines $([a], 1)$ and $([\text{id}], 1_-)$ respectively. Their action on the multiplet of ground states is

$$\mathcal{O}_{P,j}|\text{GS}, n\rangle_1 = |\text{GS}, n\rangle_P,$$
$$\mathcal{O}_{a_I,j}^{(+)}|\text{GS}, n\rangle_1 = \omega^{In}|\text{GS}, n\rangle_1,$$
$$\mathcal{O}_{a_I,j}^{(+)}|\text{GS}, n\rangle_P = \omega^{In}|\text{GS}, n\rangle_P, \tag{3.91}$$
$$\mathcal{O}_{a_I,j}^{(-)}|\text{GS}, n\rangle_1 = (-1)^{I+1}\omega^{In}|\text{GS}, n\rangle_P.$$

The ground state projection of the order parameters corresponding to the SymTFT lines $([\text{id}], E)$ and $([b], +)$ vanishes.

---

[3]Recall the symmetry defect is located at site $j = 1$.

Rep($S_3$) **SSB.** We now study the gapped phase for which $\mathfrak{B}^{\mathrm{phys}} = \mathfrak{B}^{\mathrm{sym}}$. For this, we choose

$$A_{\mathbb{Z}_2} = 1 \oplus b \,. \tag{3.92}$$

The corresponding Hamiltonian is

$$H_{\mathbb{Z}_2} = -\frac{1}{2} \sum_j \sum_{h_L, h, h_R} \mathcal{O}_{(h_L, h, h_R), j} = -\frac{1}{2} \sum_{h_L} P_{j-\frac{1}{2}}^{(h_L)} \sum_h X_j^{(h)} \sum_{h_R} P_{j+\frac{1}{2}}^{(h_R)} \,. \tag{3.93}$$

where $h_L, h, h_R \in \mathbb{Z}_2^b$ and

$$\begin{aligned}
\sum_h P_{j+\frac{1}{2}}^{(h)} &= -\frac{1}{6} \sum_{\alpha = \pm 1} \left[ 1 + \alpha \sigma_{j+\frac{1}{2}}^z \right] \left[ \sum_{n=0}^{2} Z_j^n Z_{j+1}^{-\alpha n} \right] \\
\sum_h X_j^{(h)} &= 1 + \sigma_{j-\frac{1}{2}}^x \Gamma_j \sigma_{j+\frac{1}{2}}^x \,.
\end{aligned} \tag{3.94}$$

The operators $\sum_h P_{j_1+\frac{1}{2}}^{(h)}$ and $\sum_h X_{j_2}^{(h)}$ mutually commute for all $j_1, j_2$. Therefore the ground state space is the $+1$ eigenspace of each of these operators. This space decomposes into a direct sum $V_1 \oplus V_2$ where

$$\begin{aligned}
V_1 &= \mathrm{Span}_{\mathbb{C}} \left\{ |\vec{0}, \vec{q}\rangle \right\}, \\
V_2 &= \mathrm{Span}_{\mathbb{C}} \left\{ |\vec{p}, \vec{q}\rangle \;\middle|\; p_j \neq 0 \,, \; p_j + p_{j+1} \bmod 2 = q_{j+\frac{1}{2}} \right\}.
\end{aligned} \tag{3.95}$$

The ground states can be obtained by writing the operators appearing in the Hamiltonian projected to each of these spaces. On $V_1$, since $p_j = 0$ for all $j$, $Z_j$ and $\Gamma_j$ act as the identity. Therefore we obtain

$$\left. \sum_h P_{j+\frac{1}{2}}^{(h)} \right|_{V_1} = 1 \,, \qquad \left. \sum_h X_j^{(h)} \right|_{V_1} = 1 + \sigma_{j-\frac{1}{2}}^x \sigma_{j+\frac{1}{2}}^x \,. \tag{3.96}$$

The effective Hamiltonian acting on $V_1$ then simplifies to

$$\left. \mathcal{H}_{\mathbb{Z}_2} \right|_{V_1} = -\frac{1}{2} \sum_j \left\{ 1 + \sigma_{j-\frac{1}{2}}^x \sigma_{j+\frac{1}{2}}^x \right\}, \tag{3.97}$$

which has two ground states

$$|\mathrm{GS}, 1\rangle_1 = \frac{1}{2^{L/2}} \sum_{\vec{q}} |\vec{0}, \vec{q}\rangle \,, \qquad |\mathrm{GS}, 2\rangle_1 = \frac{1}{2^{L/2}} \sum_{\vec{q}} (-1)^{\mathrm{hol}(q)} |\vec{0}, \vec{q}\rangle \,. \tag{3.98}$$

The space $V_2$ is also $2^L$ dimensional as (i) the qubits on the half-integer sites are completely constrained by their neighboring degrees of freedom and (ii) each qutrit degree of freedom is constrained to its two-dimensional subspace spanned by $p_j = 1, 2$. We define effective Pauli

operators $\widetilde{\sigma}_j^\mu$ acting on the constrained qutrits such that the states $p_j = 1, 2$ are $\widetilde{\sigma}_j^z$ eigenstates with eigenvalues $+1$ and $-1$ respectively. In terms of these

$$\sum_h P_{j+\frac{1}{2}}^{(h)}\bigg|_{V_2} = 1\,, \qquad \sum_h X_j^{(h)}\bigg|_{V_2} = 1 + \widetilde{\sigma}_j^x\,. \tag{3.99}$$

The effective Hamiltonian acting on $V_2$ simplifies to

$$\mathcal{H}_{\mathbb{Z}_2}\bigg|_{V_2} = -\frac{1}{2}\sum_j \left\{1 + \widetilde{\sigma}_j^x\right\}\,, \tag{3.100}$$

which has a single ground state

$$|\text{GS}, 3\rangle_1 = \frac{1}{2^{L/2}}{\sum_{\vec{p},\vec{q}}}' |\vec{p}, \vec{q}\rangle\,, \tag{3.101}$$

where $\sum'$ denotes a restricted sum over basis states in $V_2$. Let us describe the $\mathsf{Rep}(S_3)$ action on the untwisted sector ground states. Note that $\text{hol}(q) = 0$ for any state in $V_2$, therefore $U_P$ acts trivially on $V_2$. Meanwhile the first and second ground state are exchanged by $U_P$

$$U_P|\text{GS}, 1\rangle_1 = |\text{GS}, 2\rangle_1\,, \quad U_P|\text{GS}, 2\rangle_1 = |\text{GS}, 1\rangle_1\,, \quad U_P|\text{GS}, 3\rangle_1 = |\text{GS}, 3\rangle_1\,. \tag{3.102}$$

From the perspective for $\mathbb{Z}_2^P \in \mathsf{Rep}(S_3)$, the untwisted subspace splits into a direct sum of a $\mathbb{Z}_2^P$ SSB in $V_1$ and $\mathbb{Z}_2^P$ trivial phase in $V_2$. The ground states transform under $E$ action as

$$\mathcal{U}_E|\text{GS}, 1\rangle_1 = \mathcal{U}_E|\text{GS}, 2\rangle_1 = |\text{GS}, 3\rangle_1\,,$$
$$\mathcal{U}_E|\text{GS}, 3\rangle_1 = |\text{GS}, 1\rangle_1 + |\text{GS}, 2\rangle_1 + |\text{GS}, 3\rangle_1\,. \tag{3.103}$$

Next we describe the $P$-twisted sector ground states. The operator $\sum_h P_{j+\frac{1}{2}}^{(h)}$ remains unaltered in the presence of the $P$ symmetry twist while $X_j^{(b)}$ in $\sum_h X_{j_2}^{(h)}$ gets modified by a sign at the location of the symmetry twist. The $P$-twisted Hamiltonian continues to act block diagonally on $V_1 \oplus V_2$. We first look for $P$-twisted ground states in $V_1$. Since there is no state $|\Psi\rangle \in V_1$ that satisfies

$$-\sigma_{\frac{1}{2}}^x \sigma_{\frac{3}{2}}^x |\Psi\rangle_P = \left[\sigma_{j-\frac{1}{2}}^x \sigma_{j+\frac{1}{2}}^x\right]\bigg|_{j \neq 1} |\Psi\rangle_P = |\Psi\rangle_P\,, \tag{3.104}$$

there is no $P$-twisted ground state in $V_1$. In contrast, in $V_2$, there is $P$-twisted ground state which satisfies

$$\widetilde{\sigma}_j^x\bigg|_{j \neq 1} |\text{GS}\rangle_P = -\widetilde{\sigma}_1^x |\text{GS}\rangle_P = |\text{GS}\rangle_P\,, \tag{3.105}$$

and has the form

$$|\text{GS}\rangle_P = \frac{1}{2^{L/2}}{\sum_{\vec{p},\vec{q}}}' (-1)^{p_1+1} |\vec{p}, \vec{q}\rangle\,. \tag{3.106}$$

Let us now consider the $E$-twisted sectors. Specifically we insert a single twist at the site $j = 1$. The states with $p_j = 0$ for any $j$ do not contribute to the ground state physics as one cannot satisfy the projectors $\sum_h P^{(h)}_{j'+\frac{1}{2}}$ for each $j'$ for such a state. The ground states lie instead in the subspace spanned by basis states $|\vec{p}, \vec{q}\rangle_{(E,I)}$ with $p_j \neq 0$ and satisfying

$$q_{j+\frac{1}{2}} = p_{j+1} - p_j \bmod 2\,, \quad j \neq 1\,,$$
$$q_{\frac{3}{2}} = p_2 - (p_1 + I) \bmod 2\,. \tag{3.107}$$

The $E$-twisted ground state is an equal weight superposition of all such basis states

$$|GS\rangle_E = \frac{1}{2^{L/2}} {\sum_{\vec{p},\vec{q}}}'' \sum_I |\vec{p}, \vec{q}\rangle_{(E,I)}\,, \tag{3.108}$$

where $\sum''$ denotes a sum over basis states that satisfy (3.107). To summarize, there are three untwisted sector ground states and a single ground state each in the $P$ and $E$ twisted sectors.

Using (2.67), the untwisted sector ground states map to the twisted sector ground states as

$$\mathcal{U}_E(P)|GS, 1\rangle_1 = |GS, 1\rangle_1\,,$$
$$\mathcal{U}_E(P)|GS, 2\rangle_1 = |GS\rangle_P\,,$$
$$\mathcal{U}_E(P)|GS, 3\rangle_1 = -|GS\rangle_P\,,$$
$$\mathcal{U}_E(E)|GS, 1\rangle_1 = |GS\rangle_E\,, \tag{3.109}$$
$$\mathcal{U}_E(E)|GS, 2\rangle_1 = -|GS\rangle_E\,,$$
$$\mathcal{U}_E(E)|GS, 3\rangle_1 = 0\,.$$

The $\mathsf{Rep}(S_3)$ action on the $P$-twisted sector ground state is

$$\mathcal{U}_P(1, P)|GS\rangle_P = |GS\rangle_P\,,$$
$$\mathcal{U}_E(E, 1)|GS\rangle_P = |GS, 3\rangle_1\,,$$
$$\mathcal{U}_E(E, P)|GS\rangle_P = -|GS, 3\rangle_P\,, \tag{3.110}$$
$$\mathcal{U}_E(E, E)|GS\rangle_P = 0\,,$$

The $\mathsf{Rep}(S_3)$ action on the $E$-twisted ground state is

$$\mathcal{U}_P(E, E)|GS\rangle_E = |GS\rangle_E\,,$$
$$\mathcal{U}_E(1, E)|GS\rangle_E = |GS\rangle_E\,,$$
$$\mathcal{U}_E(P, E)|GS\rangle_E = -|GS\rangle_E\,, \tag{3.111}$$
$$\mathcal{U}_P(E, 1)|GS\rangle_E = |GS, 1\rangle_1 - |GS, 2\rangle_1\,,$$
$$\mathcal{U}_P(E, X)|GS\rangle_E = 0\,, \quad X = P, E\,.$$

Now we describe the $\mathsf{Rep}(S_3)$ order parameters for the $\mathsf{Rep}(S_3)$ SSB phase. Since in the SymTFT picture,

$$\mathfrak{B}^{\mathrm{phys}} = \mathfrak{B}^{\mathrm{sym}} = ([\mathrm{id}], 1) \oplus ([a], 1) \oplus ([b], +), \tag{3.112}$$

we expect the order parameters corresponding to the condensed lines on the physical boundary to act within the multiplet of ground states and hence act as order parameters for this phase. The order parameters corresponding to $([a], 1)$ act as

$$
\begin{aligned}
{}_1\langle \mathrm{GS}, n | \mathcal{O}^{(+)}_{a_+,j} | \mathrm{GS}, n \rangle_1 &= \begin{cases} 1, & n = 1, 2, \\ -\frac{1}{2}, & n = 3, \end{cases} \\
{}_X\langle \mathrm{GS} | \mathcal{O}^{(+)}_{a_+,j} | \mathrm{GS} \rangle_X &= -\frac{1}{2}, \quad X = P, E, \\
{}_P\langle \mathrm{GS}, n | \mathcal{O}^{(-)}_{a_+,j} | \mathrm{GS}, n \rangle_1 &= \begin{cases} 0, & n = 1, 2, \\ (\omega - \omega^2), & n = 3, \end{cases}
\end{aligned}
\tag{3.113}
$$

where $\mathcal{O}^{(\pm)}_{a_+,j} = (\mathcal{O}^{(\pm)}_{a_1,j} + \mathcal{O}^{(\pm)}_{a_2,j})/2$. The order parameters corresponding to $([b], +)$ act as

$$
\begin{aligned}
{}_1\langle \mathrm{GS}, n | \mathcal{O}_{b,j} | \mathrm{GS}, n \rangle_1 &= \begin{cases} (-1)^n, & n = 1, 2, \\ 0, & n = 3, \end{cases} \\
{}_E\langle \mathrm{GS} | \mathcal{O}^{(+)}_{b,j} | \mathrm{GS}, 3 \rangle_X &= 1, \quad X = P, E, \\
{}_{P\otimes E}\langle \mathrm{GS} | \mathcal{O}^{(-)}_{b,j} | \mathrm{GS}, 3 \rangle_X &= 1, \quad X = P, E,
\end{aligned}
\tag{3.114}
$$

where the state $| \mathrm{GS} \rangle_{P\otimes E}$ corresponds to inserting a product of $P$ and $E$ symmetry defects at the first site. Concretely it has the form

$$| \mathrm{GS} \rangle_{P\otimes E} = \frac{1}{2^{L/2}} \sum_{\vec{p}, \vec{q}}{}'' (-1)^{p_1+1} \sum_I | \vec{p}, \vec{q} \rangle_{P\otimes(E,I)}. \tag{3.115}$$

This concludes the analysis of gapped phases for $\mathsf{Rep}(S_3)$ and provides a concrete lattice realization of the continuum results in [29].

# 4 Gapless Phases and Phase Transitions

In the previous section, we discussed lattice models for gapped phases with a fusion category symmetry $\mathcal{S}$. In this section, we discuss lattice models for gapless phases with $\mathcal{S}$ symmetry. Such phases were discussed in the continuum using the SymTFT in [33]. Such a lattice model may admit deformations to two gapped phases with $\mathcal{S}$ symmetry, in which case it can also be thought of as realizing a transition between the two gapped phases.

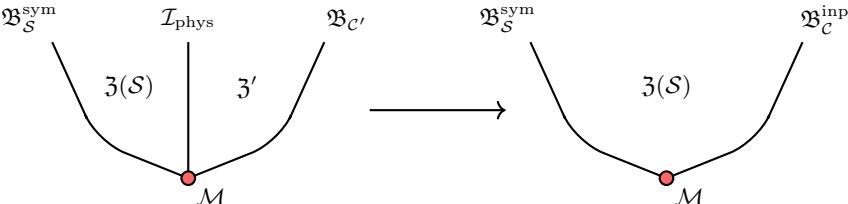

Figure 3: SymTFT picture for gapless phases, aka the club-sandwich. The interface $\mathcal{I}_{\text{phys}}$ defined by the condensable algebra reduces the topological order $\mathfrak{Z}(\mathcal{S})$ to $\mathfrak{Z}'$. The symmetry boundary carries the symmetry $\mathcal{S}$. The physical boundary is given by $\mathfrak{B}_{\mathcal{C}'}$. We compactify the interval occupied by $\mathfrak{Z}'$, which results in the right hand side picture: a topological boundary $\mathfrak{B}_{\mathcal{C}}^{\text{inp}}$ (carrying $\mathcal{C}$ topological defects) for $\mathfrak{Z}(\mathcal{S}) = \mathfrak{Z}(\mathcal{C})$, and a module category at the intersection of the symmetry boundary $\mathfrak{B}_{\mathcal{S}}^{\text{sym}}$ and $\mathfrak{B}_{\mathcal{C}}^{\text{inp}}$.

## 4.1 General Setup

**Condensed and (De-)Confined Charges in the Gapless Phase from the SymTFT.**
Recall that the gapped phases are characterized by Lagrangian algebras $\mathcal{L}_{\text{phys}}$ formed by anyons of the SymTFT $\mathfrak{Z}(\mathcal{S})$. The gapless phases, on the other hand, are characterized by condensable algebras $\mathcal{A}_{\text{phys}}$ of anyons of $\mathfrak{Z}(\mathcal{S})$ that are not Lagrangian, i.e. not maximal. The condensable algebra can be expressed as

$$\mathcal{A}_{\text{phys}} = \bigoplus n_a \boldsymbol{Q}_a, \qquad n_a \in \mathbb{Z}_{\geq 0}, \tag{4.1}$$

where $\boldsymbol{Q}_a$ are simple anyons.

The anyons with $n_a \neq 0$ are the charges of local operators that are condensed in the gapless phase. It should be noted that the condensed charges are mutually local, i.e. if $\boldsymbol{Q}_a$ and $\boldsymbol{Q}_b$ appear with non-zero coefficients in (4.1) then the braiding of these anyons is trivial.

Any local operator with a charge $\boldsymbol{Q}$ not mutually local with some condensed charge $\boldsymbol{Q}_a$, i.e. the braiding between $\boldsymbol{Q}$ and $\boldsymbol{Q}_a$ is non-trivial, must confine by a generalization of the Meissner effect. On the other hand, a non-condensed charge $\boldsymbol{Q}$ mutually local with all condensed charges $\boldsymbol{Q}_a$ can remain deconfined. Such non-condensed deconfined charges describe the charges of gapless excitations arising in the IR of the corresponding gapless phase with symmetry $\mathcal{S}$.

The condensable algebra $\mathcal{A}_{\text{phys}}$ also describes a topological interface $\mathcal{I}_{\text{phys}}$ from the SymTFT $\mathfrak{Z}(\mathcal{S})$ to another 3d TQFT $\mathfrak{Z}'$. In this setup, an anyon $\boldsymbol{Q}_a$ with $n_a \neq 0$ can end at the interface $\mathcal{I}_{\text{phys}}$, and $n_a$ is the dimension formed by topological local operators arising at the end of $\boldsymbol{Q}_a$ along $\mathcal{I}_{\text{phys}}$.

**Input Boundary for Lattice Model.** Let's choose a topological boundary condition $\mathfrak{B}_{\mathcal{C}'}$ of $\mathfrak{Z}'$, such that topological defects of $\mathfrak{B}_{\mathcal{C}'}$ form a fusion category $\mathcal{C}'$. Then, compactifying

the interval occupied by $\mathfrak{Z}'$ with $\mathcal{I}_{\mathrm{phys}}$ and $\mathfrak{B}_{\mathcal{C}'}$ being the two ends, we obtain a topological boundary condition $\mathfrak{B}_{\mathcal{C}}^{\mathrm{inp}}$ of the SymTFT $\mathfrak{Z}(\mathcal{S})$

$$\mathfrak{B}_{\mathcal{C}}^{\mathrm{inp}} = \mathcal{I}_{\mathrm{phys}} \otimes_{\mathfrak{Z}'} \mathfrak{B}_{\mathcal{C}'}\,, \tag{4.2}$$

whose topological defects form a fusion category $\mathcal{C}$ such that its Drinfeld center is the same as that for the symmetry $\mathcal{S}$, $\mathcal{Z}(\mathcal{C}) = \mathcal{Z}(\mathcal{S})$. This is shown in figure 3.

We will use $\mathfrak{B}_{\mathcal{C}}^{\mathrm{inp}}$ as the input boundary condition for constructing a lattice model for the gapless phase corresponding to $\mathcal{A}_{\mathrm{phys}}$. This fixes the module category $\mathcal{M}$ to be given by topological interfaces between the boundaries $\mathfrak{B}_{\mathcal{C}}^{\mathrm{inp}}$ and $\mathfrak{B}_{\mathcal{S}}^{\mathrm{sym}}$. This highlights an important point. A gapless phase for $\mathcal{S}$ associated to a choice $(\mathfrak{B}_{\mathcal{S}}^{\mathrm{sym}}, \mathcal{I}_{\mathrm{phys}})$ can be constructed via our method only for specific input values of $(\mathcal{C}, \mathcal{M})$. These possible input values are characterized by irreducible topological boundary conditions $\mathfrak{B}_{\mathcal{C}'}$ of $\mathfrak{Z}'$ for which the associated $(\mathcal{C}, \mathcal{M})$ are obtained as described above. This is in contrast with the story for gapped phases, which can all be constructed by our method using any possible input value of $(\mathcal{C}, \mathcal{M})$.

**Other input data for the model and symmetry action.**  Let us now describe a choice of $(\rho, h)$ using which we can construct a lattice model lying in this gapless phase. We assume that we have the knowledge of a lattice model $(\mathcal{C}', \mathcal{M}', \rho', h')$ for some indecomposable $\mathcal{C}'$-module category $\mathcal{M}'$, which is

- gapless, and

- carries gapless excitations (or local operators taking the IR theory to itself) transforming in all possible charges under the symmetry

$$\mathcal{S}' = (\mathcal{C}')_{\mathcal{M}'}^* \tag{4.3}$$

of the model. Recall that such charges are parametrized by anyons of the 3d TQFT $\mathfrak{Z}'$ discussed above, which can be identified as the SymTFT $\mathfrak{Z}(\mathcal{S}')$.

Using this model, we can construct a larger model $(\mathcal{C}', \mathcal{M}, \rho', h')$ where $\mathcal{M}$ is the indecomposable $\mathcal{C}$-module category discussed above. Let us explain how this is done. First of all, observe that the topological interface $\mathcal{I}_{\mathrm{phys}}$ provides a pivotal tensor functor

$$\phi: \ \mathcal{C}' \to \mathcal{C}\,, \tag{4.4}$$

which physically describes the image of each topological line operator $L$ living along $\mathfrak{B}_{\mathcal{C}}'$ after the compactification (C.80). The image $\phi(L)$ is a topological line operator living along $\mathfrak{B}_{\mathcal{C}}^{\mathrm{inp}}$.

Using this functor we can regard $\mathcal{M}$ as a (possibly decomposable) $\mathcal{C}'$-module category

$$\mathcal{M} = \bigoplus_{i=1}^{n} \mathcal{M}'_i, \tag{4.5}$$

where each $\mathcal{M}'_i$ is an indecomposable $\mathcal{C}'$-module category.

Second, note that the model $(\mathcal{C}', \mathcal{M}', \rho', h')$ can be converted into a model $(\mathcal{C}', \mathcal{M}'_i, \rho', h')$, for each $i$, by gauging some part of the symmetry $\mathcal{S}'$ of $(\mathcal{C}', \mathcal{M}', \rho', h')$. This model has symmetry

$$\mathcal{S}'_i = (\mathcal{C}')^*_{\mathcal{M}'_i}. \tag{4.6}$$

The model $(\mathcal{C}', \mathcal{M}, \rho', h')$ can then be expressed as

$$(\mathcal{C}', \mathcal{M}, \rho', h') = \bigoplus_{i=1}^{n} (\mathcal{C}', \mathcal{M}'_i, \rho', h'), \tag{4.7}$$

which means that we have $n$ decoupled universes, with the universe $i$ carrying the lattice model $(\mathcal{C}', \mathcal{M}'_i, \rho', h')$. Note that the Hilbert space for $(\mathcal{C}', \mathcal{M}, \rho', h')$ is a direct sum of the Hilbert spaces for $(\mathcal{C}', \mathcal{M}'_i, \rho', h')$ and the Hamiltonian block diagonalizes, with each block acting only within a single universe.

The symmetry of the model $(\mathcal{C}', \mathcal{M}, \rho', h')$ is a multi-fusion category if $n > 1$ and a fusion category if $n = 1$, and can be expressed as

$$\tilde{\mathcal{S}} = \mathcal{C}'^*_{\mathcal{M}}. \tag{4.8}$$

This tensor category $\tilde{\mathcal{S}}$ comprises of $n$ fusion category sectors described respectively by $\mathcal{S}'_i$. Physically, the tensor category $\tilde{\mathcal{S}}$ describes topological line defects living on the topological boundary

$$\tilde{\mathfrak{B}} = \mathfrak{B}^{\mathrm{sym}}_{\mathcal{S}} \otimes_{\mathfrak{Z}(\mathcal{S})} \mathcal{I}_{\mathrm{phys}}, \tag{4.9}$$

of $\mathfrak{Z}'$ by compactifying the interval occupied by $\mathfrak{Z}(\mathcal{S})$ whose two ends are $\mathfrak{B}^{\mathrm{sym}}_{\mathcal{S}}$ and $\mathcal{I}_{\mathrm{phys}}$. The $\mathcal{C}'$-module category $\mathcal{M}$ describes topological line defects living at the interface between boundaries $\tilde{\mathfrak{B}}$ and $\mathfrak{B}_{\mathcal{C}'}$ of $\mathfrak{Z}'$.

We expect that there exists a sub-manifold in the parameter space of possible $\mathcal{S}$-symmetric Hamiltonians, in the vicinity of the point occupied by the model (4.10), where this gapless phase persists, and the various universes are coupled together by gapped excitations acting as domains walls between the different universes. We leave an exploration of the phase diagram around the special models (4.10) for future work. Morally, one may think of the special gapless models (4.10) realizing gapless phases as analogs of the commuting projector Hamiltonian models realizing gapped phases.

**Lattice realization of condensed charges.** The model lying in the gapless phase that we are after is isomorphic to $(\mathcal{C}', \mathcal{M}, \rho', h')$ and can be expressed as

$$\big(\mathcal{C}, \mathcal{M}, \phi(\rho'), \phi(h')\big). \tag{4.10}$$

The symmetry $\mathcal{S}$ is realized on the system by a pivotal tensor functor

$$\sigma : \ \mathcal{S} \to \tilde{\mathcal{S}}, \tag{4.11}$$

describing the image of topological lines living on $\mathfrak{B}_{\mathcal{S}}^{\mathrm{sym}}$ under the compactification (4.9). Such functors were studied in detail for examples appearing in this work in [32].

The model $\big(\mathcal{C}, \mathcal{M}, \rho = \phi(\rho'), h = \phi(h')\big)$ has local operators transforming non-trivially under $\mathcal{S}$ that are condensed. These are a special class of operators of the form (2.16) that can be decomposed as

$$\tag{4.12}$$

and satisfy the condition

$$\tag{4.13}$$

If we pick the charge $\boldsymbol{Q} = \boldsymbol{Q}_a$, then there are $n_a$ number of linearly independent choices of $\tilde{\boldsymbol{Q}}_\mu^\rho$ satisfying this equation. These $\tilde{\boldsymbol{Q}}_\mu^\rho$ descend from the topological ends of the bulk anyon $\boldsymbol{Q}_a$ along $\mathcal{I}_{\mathrm{phys}}$.

**Phase Transitions and Order Parameters.** The gapless models (4.10) discussed above serve as phase transitions between $\mathcal{S}$-symmetric gapped phases, if the starting gapless model $(\mathcal{C}', \mathcal{M}', \rho', h')$ serves as a phase transition between two $\mathcal{S}'$-symmetric gapped phases. Let us assume there is a small deformation $\epsilon$ of $h'$ such that the two lattice models

$$(\mathcal{C}', \mathcal{M}', \rho', h' \pm \epsilon), \tag{4.14}$$

are gapped and lie respectively in $\mathcal{S}'$-symmetric gapped phases characterized by topological boundaries $\mathfrak{B}_+^{\mathrm{phys}'}$ and $\mathfrak{B}_-^{\mathrm{phys}'}$ of $\mathfrak{Z}'$. Then the models

$$\left(\mathcal{C}, \mathcal{M}, \phi(\rho'), \phi(h') \pm \phi(\epsilon)\right), \tag{4.15}$$

which are deformations of (4.10) realize $\mathcal{S}$-symmetric gapped phases characterized by topological boundaries

$$\mathfrak{B}_\pm^{\mathrm{phys}} = \mathcal{I}_{\mathrm{phys}} \otimes_{\mathfrak{Z}'} \mathfrak{B}_\pm^{\mathrm{phys}'} \tag{4.16}$$

of the SymTFT $\mathfrak{Z}(\mathcal{S})$.

We can also describe order parameters for the resulting $\mathcal{S}$-symmetric phase transition in terms of order parameters for the input $\mathcal{S}'$-symmetric phase transition. Let $(\boldsymbol{Q}', \boldsymbol{Q}'^{\rho'}_\mu)$ be a multiplet of local operators carrying charge $\boldsymbol{Q}'$ under $\mathcal{S}'$ and acting on the model $(\mathcal{C}', \mathcal{M}', \rho', h')$, which condenses in one of the gapped models $(\mathcal{C}', \mathcal{M}', \rho', h' \pm \epsilon)$, while remaining uncondensed in the other. Such a local operator is an order parameter for the $\mathcal{S}'$-symmetric phase transition $(\mathcal{C}', \mathcal{M}', \rho', h')$ between the gapped phases $\mathfrak{B}_+^{\mathrm{phys}'}$ and $\mathfrak{B}_-^{\mathrm{phys}'}$. This multiplet gives rise to a multiplet

$$\left(\boldsymbol{Q}, \phi(\boldsymbol{Q}'^{\rho'}_\mu)\right) \tag{4.17}$$

of local operators carrying charge $\boldsymbol{Q}$ under $\mathcal{S}$ and acting on the model (4.10), which condenses in one of the gapped models (4.15), while remaining uncondensed in the other. Here $\boldsymbol{Q}$ is a simple anyon in the image $\mathcal{Z}_\phi(\boldsymbol{Q}') \in \mathcal{Z}(\mathcal{S})$ of the anyon $\boldsymbol{Q}' \in \mathcal{Z}'$ under the pivotal braided tensor functor

$$\mathcal{Z}_\phi : \ \mathcal{Z}' \to \mathcal{Z}(\mathcal{S}) \tag{4.18}$$

determined by the functor $\phi$. As explained in [32], this functor is easily determined by the form of the non-Lagranigan condensable algebra $\mathcal{A}_{\mathrm{phys}}$ associated to the interface $\mathcal{I}_{\mathrm{phys}}$.

## 4.2 Example: $\mathbb{Z}_4$ SSB to $\mathbb{Z}_2$ SSB Phase Transition

We consider the construction of the gapless SSB (gSSB) phase for $\mathbb{Z}_4$ at the second-order phase transition between the $\mathbb{Z}_4$ SSB phase and the $\mathbb{Z}_2$ SSB phase. To realize this, we follow the club sandwich setup of [32]. This uses a SymTFT construction which we depict as

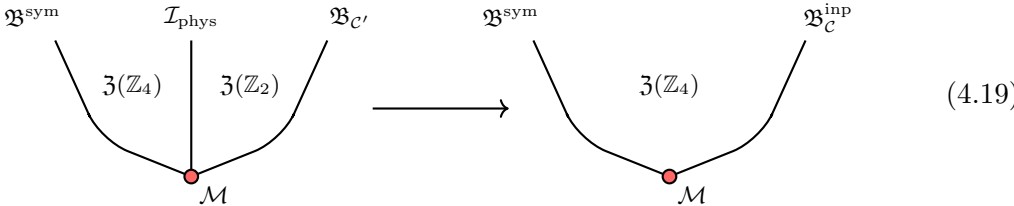

$$\tag{4.19}$$

We start by considering the SymTFT for $\mathbb{Z}_4$, which is the $\mathbb{Z}_4$ Dijkgraaf-Witten theory $\mathfrak{Z}(\mathbb{Z}_4)$, with the choice of symmetry boundary specified by

$$\mathfrak{B}^{\mathrm{sym}} = 1 \oplus e \oplus e^2 \oplus e^3 \,, \tag{4.20}$$

meaning that we condense all the purely electric charges as in (2.24). The SymTFT has a codimension-1 domain wall defined via the condensable non-Lagrangian algebra

$$\mathcal{A}_{\mathrm{phys}} = 1 \oplus e^2 \,, \tag{4.21}$$

which produces an interface $\mathcal{I}_{\mathrm{phys}}$ to a topological order $\mathfrak{Z}' = \mathfrak{Z}(\mathbb{Z}_2)$ which is the Toric Code. We denote its topological lines as

$$\{1, e', m', f'\} \in \mathcal{Z}(\mathbb{Z}_2) \,. \tag{4.22}$$

An anyon of $\mathcal{Z}(\mathbb{Z}_4)$ is converted to an anyon of $\mathcal{Z}(\mathbb{Z}_2)$ when passing through the interface $\mathcal{I}_{\mathrm{phys}}$. This determines a map $\mathcal{Z}(\mathbb{Z}_2) \to \mathcal{Z}(\mathbb{Z}_4)$ given by

$$1 \to 1 \oplus e^2 \,, \quad e' \to e \oplus e^3 \,, \quad m' \to m^2 \oplus e^2 m^2 \,, \quad f' \to em^2 \oplus e^3 m^2 \,. \tag{4.23}$$

We choose for $\mathfrak{Z}(\mathbb{Z}_2)$ the topological boundary condition

$$\mathfrak{B}_{\mathcal{C}'} = 1 \oplus e' \,, \tag{4.24}$$

on which there is a $\mathbb{Z}_2$ symmetry $\mathcal{C}' = \{1, P\}$. The $P$ line on $\mathfrak{B}_{\mathcal{C}'}$ is obtained by the projection of the bulk lines $m', f'$, while the bulk line $e'$ project to the identity in $\mathcal{C}'$ as we are condensing it. We can now construct a lattice model on this input boundary $\mathfrak{B}_{\mathcal{C}'}$, with choice $\rho'$ given by $1 \oplus P$. In particular, we consider a Hamiltonian realizing the $\mathbb{Z}_2$ transverse field Ising (TFI) model on $\mathfrak{B}_{\mathcal{C}'}$. Such a Hamiltonian corresponds to a choice of operators written in terms of $1, P$, which can be determined explicitly using the approach in section 2.2 to be

$$\mathcal{H}^{(\mathbb{Z}_2)}_{\mathrm{TFI}} = -\sum_j \left[ \begin{array}{c} \text{\small 1} \end{array} + \frac{\lambda}{2} \sum_{g,h,k} \begin{array}{c} \text{\small $g$ \quad $h$ \quad $k$} \end{array} \right]_j \,, \tag{4.25}$$

where $g, h, k \in 1, P$. This model realizes a $\mathbb{Z}_2$ symmetric trivial phase (Triv), a $\mathbb{Z}_2$ SSB phase and a $\mathbb{Z}_2$ symmetric Ising CFT at $\lambda = 1$, giving a transition between the two phases.

There are two choices of indecomposable $\mathbb{Z}_2$ module categories that can be used to define a state space that (4.25) acts on, namely the regular module $\mathcal{M} = \mathsf{Vec}_{\mathbb{Z}_2}$ and $\mathcal{M} = \mathsf{Vec}$. Here we focus on the first choice, for which the Hilbert space decomposes into $\mathbb{C}[\mathbb{Z}_2]$ state spaces

assigned to each integer site, with Pauli operators $\sigma_j^\mu$ acting on them. The Hamiltonian (4.25) then takes the familiar form

$$\mathcal{H}_{\mathrm{TFI}}^{(\mathbb{Z}_2)} = -\frac{1}{2}\sum_j \left[ (1 + \sigma_j^z \sigma_{j+1}^z) + \lambda(1 + \sigma_j^x) \right]. \tag{4.26}$$

Now consider compactifying the interval between $\mathcal{I}_{\mathrm{phys}}$ and $\mathfrak{B}_{\mathcal{C}'}$ containing $\mathfrak{Z}(\mathbb{Z}_2)$, as depicted in (4.19). Collapsing together $\mathcal{I}_{\mathrm{phys}}$ and $\mathfrak{B}_{\mathcal{C}'}$ produces a topological boundary condition $\mathfrak{B}_{\mathcal{C}}^{\mathrm{inp}}$ of the SymTFT $\mathfrak{Z}(\mathbb{Z}_4)$, which using the map (4.23) is determined to be again

$$\mathfrak{B}_{\mathcal{C}}^{\mathrm{inp}} = 1 \oplus e \oplus e^2 \oplus e^3 \,. \tag{4.27}$$

This in particular fixes the module category $\mathcal{M}$ between $\mathfrak{B}^{\mathrm{sym}}$ and $\mathfrak{B}_{\mathcal{C}}^{\mathrm{inp}}$ to be the regular module $\mathsf{Vec}_{\mathbb{Z}_4}$. As a $\mathcal{C}' = \mathbb{Z}_2$ module category, $\mathcal{M}$ decomposes as

$$\mathcal{M} = \mathsf{Vec}_{\mathbb{Z}_2} \oplus \mathsf{Vec}_{\mathbb{Z}_2} \,. \tag{4.28}$$

Correspondingly, the state space of the model splits into the direct sum of two spaces. Alternatively, we can observe this decomposition by noticing that, after compactifying $\mathfrak{Z}(\mathbb{Z}_2)$, the $\rho'$ of the $\mathbb{Z}_2$ model is converted to $\rho = 1 \oplus U^2$ for the $\mathbb{Z}_4$ model, where by $U$, $U^4 = 1$, we denote the $\mathcal{C} = \mathbb{Z}_4$ symmetry generator on $\mathfrak{B}_{\mathcal{C}}^{\mathrm{inp}}$. This is due to the map $m' \to m^2 \oplus e^2 m^2$ and the fact that $U$ is obtained as the projection of the bulk $\mathfrak{Z}(\mathbb{Z}_4)$ anyon $m$. Restricting to $\rho = 1 \oplus U^2$ leads to a direct sum decomposition into state spaces $V_1$ and $V_2$ spanned by

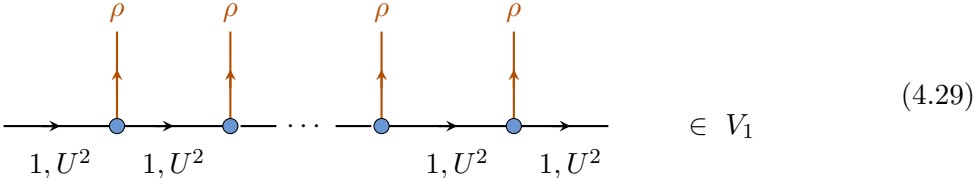

$$\tag{4.29}$$

and

$$\tag{4.30}$$

Both $V_1$ and $V_2$ correspond to the state spaces of $\mathbb{Z}_2$ symmetric models with the choice of regular module category. They are therefore tensor product spaces of local qubits $|q_j\rangle$ assigned to integer sites, where $q_j = 0, 1$ depending on whether $m_j$ is $1$ or $U^2$ respectively for $V_1$, and analogously $U$ or $U^3$ for $V_2$. Notice indeed that the degrees of freedom on the half-integer sites are completely constrained by those on integer sites. We denote a basis state as $|\vec{q}\rangle$.

Let us now study how the Hamiltonian (4.25) is realized after compactifying $\mathfrak{Z}(\mathbb{Z}_2)$ on a model with $\mathcal{M} = \mathsf{Vec}_{\mathbb{Z}_4}$ and $\rho = 1 \oplus U^2$. From the discussion above, we expect to obtain two decoupled sectors, both realizing (4.26). This is easy to see as the Hamiltonian acts block-diagonally on $V_1 \oplus V_2$ precisely in this fashion. The model at $\lambda = 1$ describes a CFT at the phase transition between the $\mathbb{Z}_4$ SSB and the $\mathbb{Z}_2$ SSB gapped phases, which decomposes as

$$\mathsf{Ising}_1 \oplus \mathsf{Ising}_2 \tag{4.31}$$

into two dynamically decoupled sectors, each realizing the Ising transition. The two sectors are connected by the action of the full $\mathbb{Z}_4$ symmetry, which maps between them as depicted here schematically

$$U^2 \circlearrowright \mathsf{Ising1} \oplus \mathsf{Ising}_2 \circlearrowleft U^2 \quad \substack{U \\ U} \tag{4.32}$$

Now we can consider the relevant deformations of this gapless model, which are obtained from the relevant deformations of the $\mathbb{Z}_2$ symmetric Ising CFT after compacitfying the $\mathfrak{Z}(\mathbb{Z}_2)$ interval. In particular, we consider the Kramers-Wannier odd relevant deformation (related to the $\epsilon$ operator in the Ising CFT or $\mathcal{O} \simeq (\sigma^z \sigma^z - \sigma^x)$ on the lattice) which, depending on its sign, drives the model to either the $\mathbb{Z}_2$ spontaneously broken or the $\mathbb{Z}_2$ trivial gapped phases. The fixed-point Hamiltonians of these two phases correspond to the Frobenius algebras $1$ and $1 \oplus P$ in $\mathsf{Vec}_{\mathbb{Z}_2}$ respectively. Correspondingly, there are two Frobenius algebras after compactifying $\mathfrak{Z}(\mathbb{Z}_2)$, which are $\mathcal{A}_1 = 1$ and $\mathcal{A}_{\mathbb{Z}_2} = 1 \oplus U^2$. The Hamiltonian $H_1$, with algebra $\mathcal{A}_1$, acting on $V_1 \oplus V_2$ has four ground states

$$|\mathrm{GS}, U^i\rangle = |\overrightarrow{U^i}\rangle = |U^i, U^i, \dots, U^i\rangle, \quad i = 0, 1, 2, 3, \tag{4.33}$$

giving the $\mathbb{Z}_4$ SSB phase. The Hamiltonian $H_{\mathbb{Z}_2}$, with algebra $\mathcal{A}_{\mathbb{Z}_2}$, acting on $V_1 \oplus V_2$ has two ground states

$$|\mathrm{GS}, 1\rangle = \frac{1}{2^{L/2}} \sum_{\vec{g}} |\vec{g}\rangle, \quad g_j = \{1, U^2\}$$

$$|\mathrm{GS}, U\rangle = \frac{1}{2^{L/2}} \sum_{\vec{g}} |\vec{g}\rangle, \quad g_j = \{U, U^3\}. \tag{4.34}$$

giving the $\mathbb{Z}_2$ SSB phase.

$$\begin{array}{ccc} \mathbb{Z}_4 \text{ SSB} & \mathsf{Ising}_1 \oplus \mathsf{Ising}_2 & \mathbb{Z}_2 \text{ SSB} \\ \hline \text{(SSB} \oplus \text{SSB)} & \lambda = 1 & \text{(Triv} \oplus \text{Triv)} \quad \lambda \end{array} \tag{4.35}$$

This model realizes the $\mathbb{Z}_4$ SSB phase and the $\mathbb{Z}_2$ SSB phase for $\lambda < 1$ and $\lambda > 1$ respectively.

Let us now dicuss the action of the $\mathbb{Z}_4$ symmetry generators. First of all, it is easy to see that $U^2$ acts within each of the two decoupled state spaces as a standard $\mathbb{Z}_2$ flip symmetry, so that we have

$$U^2 = P_{11} + P_{22} \,, \tag{4.36}$$

where $P_{jj}$ denotes the $\mathbb{Z}_2$ symmetry generator on $V_j$, for $j = 1, 2$. The action of $U$ is more interesting as it maps between $V_1$ and $V_2$. In particular, we can identify

$$U = 1_{12} + P_{21} \,. \tag{4.37}$$

The presence of the $\mathbb{Z}_2$ element $P_{21} = 1_{21} \times P_{11}$ is due precisely to the fact that acting with $\mathcal{U}_U$ twice sends a state $|\vec{q}\rangle \in V_1$ to $|-\vec{q}\rangle \in V_1$. This fully reproduces the diagram in (4.32) and provides the desired functor from $\mathbb{Z}_4$ to the multi-fusion category describing the symmetry of the two copies of Ising.

We can also discuss the order parameters that condense across this phase transition, for which we focus on the untwisted sector. The order parameters for the $\mathbb{Z}_4$ SSB phase are realized by the local operators

$$\mathcal{O}_{e,j} = (\sigma_j^z)_1 - i(\sigma_j^z)_2 \,, \quad \mathcal{O}_{e^2,j} = (1_j)_1 - (1_j)_2 \,, \quad \mathcal{O}_{e^3,j} = (\sigma_j^z)_1 + i(\sigma_j^z)_2 \,, \tag{4.38}$$

where $(\sigma_j^z)_i$ denotes the usual Pauli operator $\sigma^z$ at site $j$ of the state space $V_i$, $i = 1, 2$, while $(1_j)_i$ denotes the identity operator at site $j$ of the state space $V_i$, $i = 1, 2$. Using (4.37), one can indeed check that these operators acquire the expected non-zero vev when the two Ising models flow to the $\mathbb{Z}_2$ SSB phase. The order parameter for the $\mathbb{Z}_2$ SSB is realized by the local operator

$$\mathcal{O}_{e^2,j} = (1_j)_1 - (1_j)_2 \,, \tag{4.39}$$

which again using (4.37) acquires the expected non-zero vev when the two Ising models flow to the $\mathbb{Z}_2$ trivial phase.

## 4.3 Rep($S_3$) Phase Transitions

In this section, we describe Hamiltonians realizing second-order phase transitions between Rep($S_3$) protected gapped phases in the anyon chain model.

### 4.3.1 Rep($S_3$) SSB to Rep($S_3$)/$\mathbb{Z}_2$ SSB Phase Transition

We start with the lattice realization of the intrinsically gapless SSB (igSSB) phase at the second-order phase transition between the Rep($S_3$) SSB and Rep($S_3$)/$\mathbb{Z}_2$ SSB gapped phases.

In order to deduce the lattice realization of the transition, we use the club sandwich setup of the SymTFT [32] adapted to the lattice. This can be depicted as

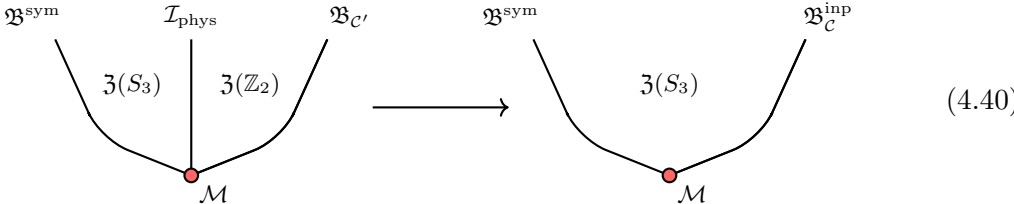

$$(4.40)$$

Consider the SymTFT for $\mathsf{Rep}(S_3)$ symmetry, that is $\mathfrak{Z}(S_3) = \mathfrak{Z}(\mathsf{Rep}(S_3))$, with the symmetry boundary

$$\mathfrak{B}^{\mathrm{sym}} = ([\mathrm{id}], 1) \oplus ([a], 1) \oplus ([b], +). \tag{4.41}$$

The SymTFT has a co-dimension-1 domain wall defined via the condensable non-Lagrangian algebra

$$\mathcal{A}_{\mathrm{phys}} = ([\mathrm{id}], 1) \oplus ([a], 1). \tag{4.42}$$

This condensation produces an interface $\mathcal{I}_{\mathrm{phys}}$ to a topological order $\mathfrak{Z}' = \mathfrak{Z}(\mathbb{Z}_2)$ which is the Toric Code. The topological lines in $\mathfrak{Z}(\mathbb{Z}_2)$ are the objects $\{1, e, m, f\} \in \mathcal{Z}(\mathbb{Z}_2)$. The interface $\mathcal{I}_{\mathrm{phys}}$ produces a map $\mathfrak{Z}(\mathbb{Z}_2) \to \mathfrak{Z}(S_3)$ under which

$$1 \to ([\mathrm{id}], 1) \oplus ([a], 1), \quad e \to ([\mathrm{id}], 1_-) \oplus ([a], 1), \quad m \to ([b], +), \quad f \to ([b], -). \tag{4.43}$$

We choose the topological boundary of $\mathfrak{Z}(\mathbb{Z}_2)$ to be

$$\mathfrak{B}_{\mathcal{C}'} = 1 \oplus e, \tag{4.44}$$

on which the symmetry is $\mathcal{C}' = \mathbb{Z}_2 = \{1, P\}$. The $P$ line on $\mathfrak{B}_{\mathcal{C}'}$ is obtained by the projections of the bulk lines $m, f$ while the bulk $e$ line projects to the identity in $\mathcal{C}'$. As in the previous example, we consider as lattice system constructed on the boundary $\mathfrak{B}_{\mathcal{C}'}$, with input $\rho' = 1 \oplus P$, the TFI model (4.25). Again, there are two possible choices of module categories: $\mathcal{M} = \mathsf{Vec}_{\mathbb{Z}_2}$ and $\mathcal{M} = \mathsf{Vec}$. The first choice is the regular module, for which the Hilbert space decomposes into $\mathbb{C}[\mathbb{Z}_2]$ state spaces assigned to each integer site with Pauli operators $\widetilde{\sigma}_j^{\mu}$ acting on them. For this choice, (4.25) is realized as

$$\mathcal{H}_{\mathrm{TFI}}^{(\mathbb{Z}_2)}(\mathcal{M} = \mathsf{Vec}_{\mathbb{Z}_2}) = -\frac{1}{2} \sum_j \left[ (1 + \widetilde{\sigma}_j^z \widetilde{\sigma}_{j+1}^z) + \lambda(1 + \widetilde{\sigma}_j^x) \right]. \tag{4.45}$$

The second choice of module category is $\mathcal{M} = \mathsf{Vec}$, for which the state space decomposes into a tensor product of qubits on the half-integer sites. We denote the Pauli operators acting on these as $\sigma_j^{\mu}$. For this choice, (4.25) is realized as

$$\mathcal{H}_{\mathrm{TFI}}^{(\mathbb{Z}_2)}(\mathcal{M} = \mathsf{Vec}) = -\frac{1}{2} \sum_j \left[ (1 + \sigma_{j+\frac{1}{2}}^z) + \lambda(1 + \sigma_{j-\frac{1}{2}}^x \sigma_{j+\frac{1}{2}}^x) \right]. \tag{4.46}$$

Notice that (4.45) and (4.46) are related by a Kramers-Wannier duality as expected.

Now consider compactifying the interval containing $\mathfrak{Z}(\mathbb{Z}_2)$ between $\mathcal{I}_{\text{phys}}$ and $\mathfrak{B}_{\mathcal{C}'}$ as depicted in (4.40). Doing so, we obtain the following topological boundary condition $\mathfrak{B}_{\mathcal{C}}^{\text{inp}}$ of the SymTFT $\mathfrak{Z}(S_3)$ using the map (4.43), which gives

$$\mathfrak{B}_{\mathcal{C}}^{\text{inp}} = ([\text{id}], 1) \oplus ([\text{id}], 1_-) \oplus 2([a], 1) . \tag{4.47}$$

Notice this was the input boundary for the anyon model described in section 2.3. The module category $\mathcal{M}$ for the $\text{Rep}(S_3)$ symmetric anyon model is determined by $\mathfrak{B}^{\text{sym}}$ and $\mathfrak{B}_{\mathcal{C}}^{\text{inp}}$ to be $\text{Vec}_{\mathbb{Z}_3}$. As a $\mathcal{C}' = \text{Vec}_{\mathbb{Z}_2}$ module category, $\mathcal{M}$ decomposes as

$$\mathcal{M} = \text{Vec} \oplus \text{Vec}_{\mathbb{Z}_2} . \tag{4.48}$$

Correspondingly, the state space of the model thus obtained splits into a direct sum of spaces. An alternate way to see this is by noting that after compactifying $\mathfrak{Z}(\mathbb{Z}_2)$ one obtains $\rho = 1 \oplus b$ in the $\text{Rep}(S_3)$ symmetric model. Restricting to $\rho = 1 \oplus b$ leads to a direct sum decomposition into state spaces $V_1$ and $V_2$ spanned by

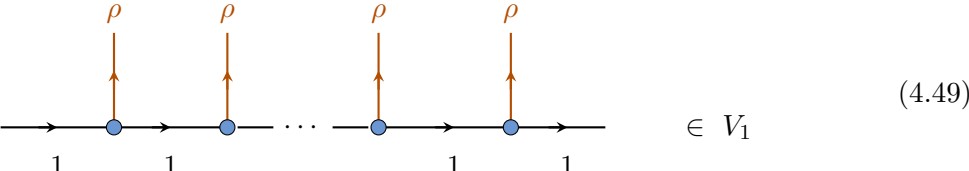

$$\tag{4.49}$$

and

$$\tag{4.50}$$

The decomposition is due to the following fusion products in $\mathcal{C} \otimes \mathcal{M} \to \mathcal{M}$

$$b \times 1 = 1 \quad , \quad b \times m = m^2 \quad , \quad b \times m^2 = m . \tag{4.51}$$

The sectors $V_1$ and $V_2$ are related by a $\mathbb{Z}_2$ gauging or Kramers-Wannier duality. $V_1$ corresponds to the $\text{Vec}$ module category. It is a tensor product space of qubits $|q_{j+1/2}\rangle$ with $p_j = 0$ for all $j$. We denote a basis state as $|\vec{q}\rangle$. $V_2$ corresponds to the $\text{Vec}_{\mathbb{Z}_2}$ module category and also decomposes as a tensor product of local qubits $|p_j\rangle$, with $p_j = 1, 2$ (a restriction of the qutrit degrees of freedom), assigned to integer sites, depending on whether $m_j$ is $m$ or $m^2$. In $V_2$,

the degrees of freedom on the half-integer sites are completely constrained by those on the integer sites via

$$p_j + p_{j+1} = q_{j+\frac{1}{2}} \mod 2 \,. \tag{4.52}$$

We will denote a basis state in $V_2$ as $|\vec{p}\rangle$.

Note that the $\mathbb{Z}_2 \subset \mathsf{Rep}(S_3)$ generated by $\mathcal{U}_P$ acts non-trivially within $V_1$ and as the identity in $V_2$. This is a consequence of the $\mathbb{Z}_2$ gauging that relates $V_1$ and $V_2$. Instead, a different (dual) $\mathbb{Z}_2$ symmetry acts identically on $V_1$ and within $V_2$ as

$$\widetilde{\mathcal{U}} : |\vec{p}\rangle \longmapsto |-\vec{p}\rangle \,. \tag{4.53}$$

By $|-\vec{p}\rangle$ we denote a state that is obtained from $|\vec{p}\rangle$ by sending each $p_j$ to $-p_j \mod 3$.

Let us now study how the model (4.25) is realized after compactifying $\mathfrak{Z}(\mathbb{Z}_2)$. On general grounds, we expect to recover two decoupled sectors realizing (4.45) and (4.46) respectively. Using (2.92) and (2.94), the Hamiltonian (4.25) realized on $\mathcal{M} = \mathsf{Vec}_{\mathbb{Z}_3}$ with $\rho = 1 \oplus b$ becomes

$$\mathcal{H}_{\mathrm{TFI}}^{(\mathsf{Rep}(S_3))} = -\sum_j \left[ \frac{1 + \sigma^z_{j+\frac{1}{2}}}{2} \cdot \frac{1 + Z_j Z^\dagger_{j+1} + Z^\dagger_j Z_{j+1}}{3} + \frac{\lambda}{2} P^{(\mathbb{Z}_2^b)}_{j-\frac{1}{2}} \cdot (1 + \sigma^x_{j-\frac{1}{2}} \Gamma_j \sigma^x_{j-\frac{1}{2}}) \cdot P^{(\mathbb{Z}_2^b)}_{j+\frac{1}{2}} \right], \tag{4.54}$$

where $P^{(\mathbb{Z}_2^b)}_{j+\frac{1}{2}} = P^{(1)}_{j+\frac{1}{2}} + P^{(b)}_{j+\frac{1}{2}}$. It is easy to check that this Hamiltonian acts block-diagonally on $V_1 \oplus V_2$. On each of these blocks we can deduce an effective projected Hamiltonian.

Since $p_j = 0$ for all $j$ on $V_1$, it follows that

$$Z_j \Big|_{V_1} = \Gamma_j \Big|_{V_1} = P^{(\mathbb{Z}_2^b)}_{j+\frac{1}{2}} \Big|_{V_1} = 1 \,. \tag{4.55}$$

Therefore, the Hamiltonian simplifies to

$$\mathcal{H}_{\mathrm{TFI}}^{(\mathsf{Rep}(S_3))} \Big|_{V_1} = -\frac{1}{2} \sum_j \left[ (1 + \sigma^z_j) + \lambda (1 + \sigma^x_j \sigma^x_{j+1}) \right] = \mathcal{H}_{\mathrm{TFI}}^{(\mathbb{Z}_2)}(\mathcal{M} = \mathsf{Vec}) \,. \tag{4.56}$$

On $V_2$, we define Pauli operators $\widetilde{\sigma}^\mu_j$ as in (3.100) acting on the reduced qutrit space spanned by $p_j \neq 0$. In terms of these operators

$$\left( \frac{1 + \sigma^z_{j+\frac{1}{2}}}{2} \cdot \frac{1 + Z_j Z^\dagger_{j+1} + Z^\dagger_j Z_{j+1}}{3} \right) \Bigg|_{V_2} = \frac{1 + \widetilde{\sigma}^z_j \widetilde{\sigma}^z_{j+1}}{2} \,,$$

$$\left( P^{(\mathbb{Z}_2^b)}_{j+\frac{1}{2}} \right) \Bigg|_{V_2} = 1 \,, \tag{4.57}$$

$$\left( \sigma^x_{j-\frac{1}{2}} \Gamma_j \sigma^x_{j+\frac{1}{2}} \right) \Bigg|_{V_2} = \widetilde{\sigma}^x_j = \mathcal{H}_{\mathrm{TFI}}^{(\mathbb{Z}_2)}(\mathcal{M} = \mathsf{Vec}_{\mathbb{Z}_2}) \,.$$

Hence, the effective Hamiltonian on $V_2$ simplifies to

$$\mathcal{H}_{\text{TFI}}^{(\text{Rep}(S_3))}\Big|_{V_2} = -\frac{1}{2}\sum_j \left[(1 + \widetilde{\sigma}_j^z\widetilde{\sigma}_{j+1}^z) + \lambda(1 + \widetilde{\sigma}_j^x)\right]. \tag{4.58}$$

As expected, we recover that (4.56) and (4.58) are related by a Kramers-Wannier duality and are (4.25) realized on the Vec and $\text{Vec}_{\mathbb{Z}_2}$ module cetegories for $\text{Vec}_{Z_2}$ respectively. The model at $\lambda = 1$ describes a CFT at the phase transition between the $\text{Rep}(S_3)$ SSB and $\text{Rep}(S_3)/\mathbb{Z}_2$ SSB gapped phases which decomposes as

$$\text{Ising}_1 \oplus \text{Ising}_2\,, \tag{4.59}$$

into two dynamically decoupled sectors that each realize the Ising transition. The two sectors are constrained by $\text{Rep}(S_3)$ symmetry which acts between them, schematically as

$$E \circlearrowleft \underbrace{\text{Ising}_1 \oplus \text{Ising}_2}_{E} \circlearrowright P \tag{4.60}$$

The relevant deformations of this gapless model are obtained, as previously discussed, from relevant deformations of Ising. In particular, we consider on the Kramers-Wannier odd relevant deformation, which drives the model to the $\mathbb{Z}_2$ SSB or $\mathbb{Z}_2$ trivial gapped phases. The fixed-point Hamiltonians of these two phases correspond to the Frobenius algebras $1$ and $1\oplus P$ in $\text{Vec}_{\mathbb{Z}_2}$ respectively. Correspondingly, there are two Frobenius algebras after compactifying $\mathfrak{Z}(\mathbb{Z}_2)$, which are $\mathcal{A}_1 = 1$ and $\mathcal{A}_{\mathbb{Z}_2} = 1\oplus b$. These correspond to the two possible Hamiltonians $H_1$ and $H_{\mathbb{Z}_2}$ discussed in section 3.3, which realize the $\text{Rep}(S_3)/\mathbb{Z}_2$ SSB and $\text{Rep}(S_3)$ SSB phases respectively.

$$\begin{array}{ccc} \underset{\text{(Triv} \oplus \text{SSB)}}{\text{Rep}(S_3)/\mathbb{Z}_2 \text{ SSB}} & \underset{\lambda = 1}{\text{Ising}_1 \oplus \text{Ising}_2} & \underset{\text{(SSB} \oplus \text{Triv)}}{\text{Rep}(S_3) \text{ SSB}} \\ \xrightarrow{\hspace{10cm}} & & \lambda \end{array} \tag{4.61}$$

The model realizes the $\text{Rep}(S_3)/\mathbb{Z}_2$ SSB and the $\text{Rep}(S_3)$ SSB for $\lambda < 1$ and $\lambda > 1$ respectively. From the perspective of the $\mathbb{Z}_2$ symmetries $\mathcal{U}_P$ and $\widetilde{\mathcal{U}}$ acting within $V_1$ and $V_2$, $\mathcal{H}_1$ and $\mathcal{H}_{\mathbb{Z}_2}$ are the $\text{Triv} \oplus \text{SSB}$ and $\text{SSB} \oplus \text{Triv}$ gapped phases. The Hamiltonian $\mathcal{H}_1$ has 3 ground states given in (3.85), 1 of which is in $V_1$ while 2 are in $V_2$. Similarly the Hamiltonian $\mathcal{H}_{\mathbb{Z}_2}$ also has 3 ground states however 2 of them are in $V_1$ given in (3.101) while 1 is in $V_2$ given in (3.98). Despite having the same number of vacua, these two gapped phases can be distinguished by their pattern of $\text{Rep}(S_3)$ symmetry breaking as detailed in section 3.3.

Let us now discuss the action of the $\mathsf{Rep}(S_3)$ symmetry generators. $P$ acts trivially on $V_2$ and as a $\mathbb{Z}_2$ operator measuring the total spin parity on $V_1$. Therefore we have the identification

$$P = P_{11} + 1_{22} \,, \tag{4.62}$$

where $P_{11}$ denotes the $\mathbb{Z}_2$ symmetry generator on $V_1$ while $1_{22}$ denotes the identity in $V_2$. The action of $E$ is more interesting as it maps between $V_1$ and $V_2$. Let us consider the action of $E$ on $V_1$ first. We know from (2.61) that all the states with $\sum_j q_{j+\frac{1}{2}} = 1$ mod 2 are in the kernel of the E action. The states $|\vec{q}\rangle$ with $\sum_j q_{j+\frac{1}{2}} = 0$ mod 2 are mapped to states in $V_1$ as follows

$$\mathcal{U}_E|\vec{q}\rangle = |\vec{p}_1(\vec{q})\rangle + |\vec{p}_2(\vec{q})\rangle \,, \tag{4.63}$$

where any two qubits $p_j$ and $p_{j+1}$ are constrained as

$$p_j + p_{j+1} = q_{j+\frac{1}{2}} \text{ mod } 2 \,. \tag{4.64}$$

We denote this action by $S_{12}$ as it maps $V_1$ to $V_2$. Acting with $E$ on $V_2$ gives

$$\mathcal{U}_E|\vec{p}\rangle = |\vec{q}(\vec{p})\rangle + |-\vec{p}\rangle \,. \tag{4.65}$$

Here the qubits $q_{j+\frac{1}{2}}$ in $|\vec{q}(\vec{p})\rangle$ are again constrained by the initial $\vec{p}$ qubits via (4.64). Therefore, $E$ acts on the $V_2$ as $S_{21} + \widetilde{U}_{22}$. In summary, we obtain the identification

$$E = S_{12} + S_{21} + \widetilde{U}_{22} \,. \tag{4.66}$$

This fully reproduces the diagram in (4.60) and provides the desired functor from $\mathsf{Rep}(S_3)$ to the multi-fusion category describing the symmetry of the two copies of Ising.

Let us finally discuss some of the local order parameters that condense across this phase transition. The order parameters for the $\mathsf{Rep}(S_3)/\mathbb{Z}_2$ SSB phase descend from the multiplets $\mathcal{O}_{a_J,j}^{\pm}$, for $J = 1, 2$. These include in particular the local operator $\mathcal{O}_{a,j} = (\sigma_j^z)_2$, which acquires a vev when $\mathsf{Ising}_1$ flows to the trivial phase and $\mathsf{Ising}_2$ flows to the $\mathbb{Z}_2$ SSB phase. The untwisted order parameters for the $\mathsf{Rep}(S_3)$ SSB phase descend from the multiplets $\mathcal{O}_{a,j}^{\pm}$ and $\mathcal{O}_{b,j}$. Among these, we have $\mathcal{O}_b = (\sigma_j^z)_1$, which acquires a vev when $\mathsf{Ising}_1$ flows to the $\mathbb{Z}_2$ SSB phase and $\mathsf{Ising}_2$ flows to the trivial phase.

### 4.3.2   $\mathsf{Rep}(S_3)$ Trivial to $\mathsf{Rep}(S_3)/\mathbb{Z}_2$ SSB Phase Transition

We now describe a model realizing the gSPT phase for $\mathsf{Rep}(S_3)$ corresponding to the transition from the $\mathsf{Rep}(S_3)$ trivial phase to the $\mathsf{Rep}(S_3)/\mathbb{Z}_2$ SSB phase. In the club sandwich setup, this is realized by starting from $\mathfrak{Z}(S_3)$ and condensing the non-Lagrangian algebra $\mathcal{A}_{\text{phys}} = ([\text{id}], 1) \oplus$

$([\mathrm{id}], 1_-)$. This produces an interface $\mathcal{I}_{\mathrm{phys}}$ to the reduced topological order $\mathfrak{Z}' = \mathfrak{Z}(\mathbb{Z}_3)$ which is the $\mathbb{Z}_3$ Dijkgraaf-Witten theory.

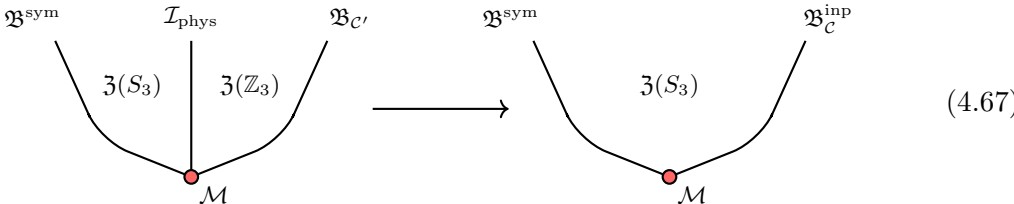

$$(4.67)$$

The topological lines in $\mathfrak{Z}' = \mathfrak{Z}(\mathbb{Z}_3)$ form the Abelian group $\mathbb{Z}_3 \times \mathbb{Z}_3 = \langle e, m \rangle$ under fusion. The interface $\mathcal{I}_{\mathrm{phys}}$ provides a map from the lines in $\mathfrak{Z}(\mathbb{Z}_3)$ to the lines in $\mathfrak{Z}(S_3)$ under which

$$1 \longmapsto ([\mathrm{id}], 1) \oplus ([\mathrm{id}], 1_-), \quad e \longmapsto ([a], 1), \quad m \longmapsto ([\mathrm{id}], E). \qquad (4.68)$$

We pick the topological boundary for $\mathfrak{Z}(\mathbb{Z}_3)$ to be

$$\mathfrak{B}_{\mathcal{C}'} = 1 \oplus e \oplus e^2, \qquad (4.69)$$

on which the symmetry is $\mathcal{C}' = \mathbb{Z}_3 = \{1, P, P^2\}$. The $P$ line on $\mathfrak{B}_{\mathcal{C}'}$ is obtained by the projection of the bulk line $m$, while the bulk $e$ line projects to the identity in $\mathcal{C}'$. The projections of the remaining lines are deduced by requiring consistency between fusion rules. Now we may consider a lattice system constructed on the input boundary $\mathfrak{B}_{\mathcal{C}'}$ with input $\rho' = 1 \oplus P \oplus P^2$. Specifically, let us consider a Hamiltonian that realizes the $\mathbb{Z}_3$ quantum clock model on the boundary $\mathfrak{B}_{\mathcal{C}'}$. Such a Hamiltonian corresponds to a choice of operators written in terms of $1, P, P^2$

$$\mathcal{H}_{\mathbb{Z}_3 \text{ clock}}^{(\mathbb{Z}_2)} = -\sum_j \left[ \Big|_1 + \frac{\lambda}{3} \sum_{g,h,k} {}_g\!\!\overset{h}{\underset{\phantom{k}}{\longrightarrow}}{}_k \right]_j, \qquad (4.70)$$

where in $g, h, k \in 1, P, P^2$. This model realizes a $\mathbb{Z}_3$ symmetric trivial phase (Triv), a $\mathbb{Z}_3$ SSB phase and a $\mathbb{Z}_3$ transition between the two phases in the universality class of the critical three-state Potts model at $\lambda = 1$.

Now consider compactifying the interval containing $\mathfrak{Z}(\mathbb{Z}_3)$ between $\mathcal{I}_{\mathrm{phys}}$ and $\mathfrak{B}_{\mathcal{C}'}$ as depicted in (4.67). Doing so, we obtain the following topological boundary condition $\mathfrak{B}_{\mathcal{C}}^{\mathrm{inp}}$ of the SymTFT $\mathfrak{Z}(S_3)$ using (4.68)

$$\mathfrak{B}_{\mathcal{C}}^{\mathrm{inp}} = ([\mathrm{id}], 1) \oplus ([\mathrm{id}], 1_-) \oplus 2([a], 1), \qquad (4.71)$$

which was the input boundary for the anyon model described in section 2.3. The category of lines on $\mathfrak{B}_{\mathcal{C}}^{\mathrm{inp}}$ is $\mathsf{Vec}_{S_3}$, such that the SymTFT line $([b], +)$ projects to $b \oplus ab \oplus a^2 b$, while the

charged line $([\mathrm{id}], E)$ projects to $a \oplus a^2$. Therefore, after compactifying, $\rho = 1 \oplus a \oplus a^2$. The module category $\mathcal{M}$ for the $\mathsf{Rep}(S_3)$ symmetric anyon model is determined by $\mathfrak{B}^{\mathrm{sym}}$ and $\mathfrak{B}^{\mathrm{inp}}_{\mathcal{C}}$ to be $\mathsf{Vec}_{\mathbb{Z}_3}$, which is also indecomposable as a $\mathcal{C}' = \mathsf{Vec}_{\mathbb{Z}_3}$ module category. To summarize, the club sandwich after compactifying $\mathfrak{Z}(\mathbb{Z}_3)$ produces an anyon model with input

$$\mathcal{C} = \mathsf{Vec}_{S_3} \quad , \quad \mathcal{M} = \mathsf{Vec}_{\mathbb{Z}_3} \quad , \quad \rho = 1 \oplus a \oplus a^2 \,. \tag{4.72}$$

The Hamiltonian for this model is given by (4.70). In terms of the spin operators described in section 2.3, this takes the form

$$\mathcal{H}^{(\mathsf{Rep}(S_3))}_{\mathbb{Z}_3\text{-clock}} = -\frac{1}{6} \sum_j \left[ (1 + \sigma^z_{j+\frac{1}{2}})(1 + Z_j Z^\dagger_{j+1} + Z^\dagger_j Z_{j+1}) + \frac{\lambda}{2}(1 + \sigma^z_{j-\frac{1}{2}})(1 + X_j + X^2_j)(1 + \sigma^z_{j+\frac{1}{2}}) \right]$$

$$\approx -\frac{1}{3} \sum_j \left[ (1 + Z_j Z^\dagger_{j+1} + Z^\dagger_j Z_{j+1}) + \lambda(1 + X_j + X^2_j) \right] . \tag{4.73}$$

In the second line, with a slight abuse of notation, we write down the effective low energy model in the $\sigma^z_{j+\frac{1}{2}} = 1$ subspace. This is the well known $\mathbb{Z}_3$ clock model realized as an $\mathsf{Rep}(S_3)$ symmetric model. The phase diagram parametrized by $\lambda$ is

$$\tag{4.74}$$

Finally, let us discuss the action of the $\mathsf{Rep}(S_3)$ symmetry generators on this model. Firstly $P$ acts trivially on the low energy subspace since $\sigma^z = 1$. Furthermore, there is an emergent $\mathbb{Z}_3$ symmetry generated by $\eta = \prod_j X_j$ within this subspace. The $E$ symmetry acts as the sum of the $\mathbb{Z}_3$ generators

$$\mathcal{U}_P = 1 \quad , \quad \mathcal{U}_E = \eta + \eta^2. \tag{4.75}$$

This provides the desired functor from $\mathsf{Rep}(S_3)$ to the symmetry $\mathbb{Z}_3$ of the reduced model.

The order parameters for this phase transition, focusing on the untwisted sector ones, are realized by the local operators

$$\mathcal{O}^+_{a_1,j} = Z_j , \quad \mathcal{O}^+_{a_2,j} = Z^\dagger_j , \tag{4.76}$$

which acquire a vev when the critical 3-state Potts flows to the $\mathbb{Z}_3$ SSB phase.

## 4.4 Example: $\mathsf{Rep}(D_8)$ gSPTs as Transitions between $\mathsf{Rep}(D_8)$ SPTs

Another particularly simple non-invertible symmetry in (1+1)d is the representation category of the dihedral group $D_8 = \mathbb{Z}_4 \rtimes \mathbb{Z}_2$. The phases with $\mathsf{Rep}(D_8)$ were discussed from the continuum in [33], including SPTs, SSB and gapless phases, which include the first non-invertible intrinsically gapless SPT phase. Earlier analysis of the gapped SPT phases appeared in [4]. Recently, a lattice model realizing these SPTs on the cluster state appeared in [52].

The gapped phases can be readily constructed using the anyon chain as prescribed in our section 3. Our focus will be on phase-transitions between the gapped phases. Motivated by [52], we consider the phase transitions between the SPT phases.

We use the red, green, blue (RGB) notation for the elements of $\mathsf{Rep}(D_8)$ and the corresponding SymTFT anyons, as in [76] (see [33] for the full dictionary to the more standard notation in terms of conjugacy classes and representations of stabilizers). There exist three $\mathsf{Rep}(D_8)$-symmetric SPT phases, which in [33] are characterized in terms of the three Lagrangian algebras

$$\mathcal{A}_{\mathrm{SPT}_{RG}} := \mathcal{A}_{27} = 1 \oplus e_G \oplus e_R \oplus e_{RG} \oplus 2m_B$$

$$\mathcal{A}_{\mathrm{SPT}_{RB}} := \mathcal{A}_{30} = 1 \oplus e_B \oplus e_R \oplus e_{RB} \oplus 2m_G \qquad (4.77)$$

$$\mathcal{A}_{\mathrm{SPT}_{GB}} := \mathcal{A}_{32} = 1 \oplus e_B \oplus e_G \oplus e_{GB} \oplus 2m_R \,.$$

The transitions between these SPTs are gapless $\mathsf{Rep}(D_8)$-symmetric phases given in terms of non-maximal condensable algebras[4]

$$\mathcal{A}_{\mathrm{gSPT}_R} := \mathcal{A}_{\mathrm{SPT}_{R,G}} \cap \mathcal{A}_{\mathrm{SPT}_{R,B}} = \mathcal{A}_4 = 1 \oplus e_R$$

$$\mathcal{A}_{\mathrm{gSPT}_G} := \mathcal{A}_{\mathrm{SPT}_{RG}} \cap \mathcal{A}_{\mathrm{SPT}_{GB}} = \mathcal{A}_5 = 1 \oplus e_G \qquad (4.78)$$

$$\mathcal{A}_{\mathrm{gSPT}_B} := \mathcal{A}_{\mathrm{SPT}_{RB}} \cap \mathcal{A}_{\mathrm{SPT}_{GB}} = \mathcal{A}_6 = 1 \oplus e_B \,.$$

As discussed in [33], these are gapless SPTs for $\mathsf{Rep}(D_8)$.

From this point on we consider only $\mathcal{A}_{gSPT_R}$. The other two cases are similar. In [33], the condensable algebra $\mathcal{A}_{gSPT_R}$ was shown to correspond to an interface $\mathcal{I}_R$ between SymTFTs $\mathfrak{Z}(D_8)$ and $\mathfrak{Z}(\mathbb{Z}_2 \times \mathbb{Z}_2)$, giving the following club-sandwich setup

$$(4.79)$$

---

[4]the notation $\mathcal{A}_i$ again refers to the conventions in [33].

where $\mathfrak{B}^{\text{sym}}_{\mathsf{Rep}(D_8)}$ corresponds to the Lagrangian algebra

$$\mathcal{L}^{\text{sym}}_{\mathsf{Rep}(D_8)} = 1 \oplus e_{RGB} \oplus m_{RB} \oplus m_{GB} \oplus m_{RB} \,, \tag{4.80}$$

Inserting a boundary condition $\mathfrak{B}^{\text{phys}}$ of $\mathfrak{Z}(\mathbb{Z}_2 \times \mathbb{Z}_2)$ gives a $\mathsf{Rep}(D_8)$-symmetric theory $\mathfrak{T}$. Any such theory $\mathfrak{T}$ has the property that the charge $e_R$ is condensed in it.

The gapped phases $\mathcal{A}_{\text{SPT}_{R,G}}$ and $\mathcal{A}_{\text{SPT}_{R,B}}$ arise by choosing $\mathfrak{B}^{\text{phys}}$ to be topological boundary conditions associated to Lagrangian algebras $1 \oplus e_1 \oplus e_2 \oplus e_1 e_2$ and $1 \oplus m_1 \oplus m_2 \oplus m_1 m_2$, where $e_i$ and $m_i$ are topological line defects of the SymTFT $\mathfrak{Z}' = \mathfrak{Z}(\mathbb{Z}_2 \times \mathbb{Z}_2)$.

We are looking for a conformal boundary condition $\mathfrak{B}^{\text{phys}}$ that acts as a transition between the topological boundaries $1 \oplus e_1 \oplus e_2 \oplus e_1 e_2$ and $1 \oplus m_1 \oplus m_2 \oplus m_1 m_2$. Such a boundary is provided through the sandwich construction

$$\tag{4.81}$$

where we stack two Ising CFTs together, referred to as Ising $\times$ Ising, which is then opened up into a $\mathbb{Z}_2 \times \mathbb{Z}_2$ using the $\mathbb{Z}_2$ spin-flip symmetries of the two Ising factors. The required boundary condition is the physical boundary $\mathfrak{B}^{\text{phys}}_{\text{Ising} \times \text{Ising}}$ for this sandwich construction, where we choose the symmetry boundary $\mathfrak{B}^{\text{sym}}_{\mathbb{Z}_2 \times \mathbb{Z}_2}$ to be the one corresponding to the Lagrangian algebra $1 \oplus e_1 \oplus e_2 \oplus e_1 e_2$. Since Ising $\times$ Ising acts as a transition between $\mathbb{Z}_2 \times \mathbb{Z}_2$ fully SSB phase with 4 vacua and the trivial $\mathbb{Z}_2 \times \mathbb{Z}_2$ symmetric phase with a single vacuum, the boundary $\mathfrak{B}^{\text{phys}}_{\text{Ising} \times \text{Ising}}$ acts as a transition between $1 \oplus e_1 \oplus e_2 \oplus e_1 e_2$ and $1 \oplus m_1 \oplus m_2 \oplus m_1 m_2$.

We can construct the Ising $\times$ Ising lattice model on the boundary $\mathfrak{B}_{\mathcal{C}'} = 1 \oplus e_1 \oplus e_2 \oplus e_1 e_2$ and hence we have

$$\mathcal{C}' = \mathsf{Vec}_{\mathbb{Z}_2 \times \mathbb{Z}_2} = \{1, P_1, P_2, P_1 P_2\} \tag{4.82}$$

formed by lines living on $\mathfrak{B}_{\mathcal{C}'}$. We choose

$$\rho' = 1 \oplus P_1 \oplus P_2 \oplus P_1 P_2 \tag{4.83}$$

with $h'$ being just the stack product for the local Hamiltonians corresponding to the two Ising models, which has been discussed earlier in the text. The IR limit of the model is the conformal Ising $\times$ Ising theory.

Colliding $\mathfrak{B}_{\mathcal{C}'}$ with $\mathcal{I}_R$ we learn that the input boundary is

$$\mathfrak{B}^{\text{inp}}_{\mathcal{C}} = 1 \oplus e_R \oplus e_G \oplus e_{RG} \oplus 2m_B \,. \tag{4.84}$$

For such a boundary we have the input fusion category

$$\mathcal{C} = \mathsf{Vec}_{D_8} \tag{4.85}$$

carrying topological defects generating $D_8$ group. The functor from $\mathcal{C}'$ to $\mathcal{C}$ can be computed to be

$$
\begin{aligned}
P_1 &\mapsto x \\
P_2 &\mapsto a^2 \,,
\end{aligned}
\tag{4.86}
$$

where we have expressed $D_8$ as $\mathbb{Z}_4 \rtimes \mathbb{Z}_2$ with $a$ being the generator of $\mathbb{Z}_4$ and $x$ being the generator of $\mathbb{Z}_2$. Using this we find that $\rho$ is

$$\rho' \mapsto \rho = 1 \oplus x \oplus a^2 \oplus a^2 x \tag{4.87}$$

and $h$ can also be obtained by applying the above functor to $h'$.

The module category is easily seen to be

$$\mathcal{M} = \mathcal{M}' = \mathsf{Vec} \tag{4.88}$$

as the intersection between the Lagrangian algebras for $\mathfrak{B}_{\mathcal{C}}^{\mathrm{inp}}$ and $\mathfrak{B}_{\mathsf{Rep}(D_8)}^{\mathrm{sym}}$ is trivial.

Colliding $\mathfrak{B}_{\mathsf{Rep}(D_8)}^{\mathrm{sym}}$ and $\mathcal{I}_R$, we obtain a topological boundary of $\mathfrak{Z}(\mathbb{Z}_2 \times \mathbb{Z}_2)$

$$\tilde{\mathfrak{B}} = 1 \oplus e_1 m_2 \oplus e_2 m_1 \oplus e_1 e_2 m_1 m_2 \tag{4.89}$$

which is obtained from $\mathfrak{B}_{\mathbb{Z}_2 \times \mathbb{Z}_2}^{\mathrm{sym}}$ by gauging the $\mathbb{Z}_2 \times \mathbb{Z}_2$ symmetry of $\mathfrak{B}_{\mathbb{Z}_2 \times \mathbb{Z}_2}^{\mathrm{sym}}$ with a non-trivial discrete torsion in $H^2(\mathbb{Z}_2 \times \mathbb{Z}_2, U(1)) = \mathbb{Z}_2$. This means that the underlying lattice model describing the $\mathsf{Rep}(D_8)$ transition is obtained by gauging the $\mathbb{Z}_2 \times \mathbb{Z}_2$ spin-flip symmetry of Ising $\times$ Ising lattice model with discrete torsion.

The topological lines living on $\tilde{\mathfrak{B}}$ form $\tilde{\mathcal{S}} = \mathbb{Z}_2 \times \mathbb{Z}_2$ that is dual to the $\mathbb{Z}_2 \times \mathbb{Z}_2$ living on $\mathfrak{B}_{\mathbb{Z}_2 \times \mathbb{Z}_2}^{\mathrm{sym}}$, and hence we denote them by hats on top

$$\tilde{\mathcal{S}} = \{1, \widehat{P}_1, \widehat{P}_2, \widehat{P}_1 \widehat{P}_2\} \tag{4.90}$$

The functor from the $\mathsf{Rep}(D_8)$ symmetry to this dual $\mathbb{Z}_2 \times \mathbb{Z}_2$ symmetry is

$$
\begin{aligned}
R &\mapsto 1 \\
G &\mapsto \widehat{P}_1 \widehat{P}_2 \\
B &\mapsto \widehat{P}_1 \oplus \widehat{P}_2 \,.
\end{aligned}
\tag{4.91}
$$

This converts the gauged Ising $\times$ Ising lattice model into a $\mathsf{Rep}(D_8)$ symmetric model. This model transitions between the two $\mathsf{Rep}(D_8)$ SPT phases under discussion.

This completes the description of these transitions as anyonic chain models, which can be converted into a spin chain model. We will return to this aspect along with a discussion of other types of transitions for $\mathsf{Rep}(D_8)$ in a future work.

**Acknowledgements.**

We thank Ömer Aksoy, Andrea Antinucci, Arkya Chatterjee, Christian Copetti, Luisa Eck, Paul Fendley, Sanjay Moudgalya, Shu-Heng Shao, Xiao-Gang Wen for discussions. We thank Ömer Aksoy, Arkya Chatterjee, and Xiao-Gang Wen for coordinating submission of their related work [77] with ours. LB thanks Niels Bohr International Academy for hospitality, where a part of this work was completed. LB is funded as a Royal Society University Research Fellow through grant URF\R1\231467. The work of SSN is supported by the UKRI Frontier Research Grant, underwriting the ERC Advanced Grant "Generalized Symmetries in Quantum Field Theory and Quantum Gravity" and the Simons Foundation Collaboration on "Special Holonomy in Geometry, Analysis, and Physics", Award ID: 724073, Schafer-Nameki. The work of AT is funded by Villum Fonden Grant no. VIL60714.

# A Example: Abelian symmetry $\mathcal{S} = \mathbb{Z}_4 \times \mathbb{Z}_2$

In this appendix we study the anyon chain model with $\mathbb{Z}_4 \times \mathbb{Z}_2$ symmetry. This symmetry group is simple enough, yet captures all the aspects of finite Abelian group symmetries discussed in the main text. We denote the group as

$$\mathbb{Z}_4 \times \mathbb{Z}_2 = \langle a, b \,|\, a^4 = b^2 = 1, ab = ba \rangle. \tag{A.1}$$

An element $g = a^p \, b^q \in \mathbb{Z}_4 \times \mathbb{Z}_2$ will be denoted as $(p, q)$ where $p = 0, 1, 2, 3$ and $q = 0, 1$. To define the lattice model, we pick

$$\mathcal{C} = \mathcal{M} = \mathsf{Vec}_{\mathbb{Z}_4 \times \mathbb{Z}_2}, \qquad \rho = \bigoplus_g g. \tag{A.2}$$

A basis state in the symmetry untwisted Hilbert space has the form

$$\tag{A.3}$$

where $m_j = (p_j, q_j)$ are simple objects in $\mathcal{M}$. Since the morphisms from $\mathcal{C} \times \mathcal{M} \to \mathcal{M}$ are uniquely specified for simple objects in $\mathcal{C}$ and $\mathcal{M}$, there are no degrees of freedom on the half integer sites. The untwisted Hilbert space $\mathcal{V}_1$ admits a tensor decomposition into on-site Hilbert spaces as

$$\mathcal{V}_1 \cong \mathbb{C}[\mathbb{Z}_4 \times \mathbb{Z}_2]^{\otimes L} = \mathrm{Span}_{\mathbb{C}}\left\{ |\vec{g} = (\vec{p}, \vec{q})\rangle \equiv |(p_1, q_1), (p_2, q_2), \cdots, (p_L, q_L)\rangle \right\}. \tag{A.4}$$

Additionally there are symmetry twisted sectors whose basis states are

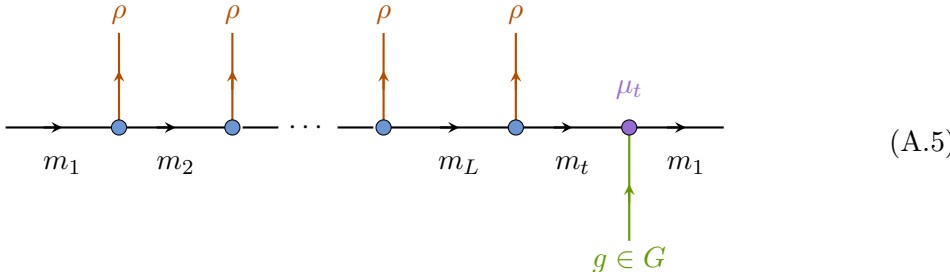

$$\text{(A.5)}$$

Each twisted Hilbert space is isomorphic. The total Hilbert space decomposes into a direct sum of symmetry twisted sectors as

$$\mathcal{V} = \bigoplus_g \mathcal{V}_g \,, \qquad \mathcal{V}_g \cong \mathbb{C}[\mathbb{Z}_4 \times \mathbb{Z}_2]^{\otimes L} \,. \tag{A.6}$$

To compare the anyon chain model with generalized Ising spin chains, it is illustrative to define the on-site operators $\left\{ X_j \,, Z_j \,, \sigma_j^x \,, \sigma^z \right\}$ which act within each symmetry twisted block and have the form

$$X_j = \begin{pmatrix} 0 & 1 & 0 & 0 \\ 0 & 0 & 1 & 0 \\ 0 & 0 & 0 & 1 \\ 1 & 0 & 0 & 0 \end{pmatrix} \otimes \text{Id}_2 \,, \quad Z_j = \begin{pmatrix} 1 & 0 & 0 & 0 \\ 0 & \text{i} & 0 & 0 \\ 0 & 0 & -1 & 1 \\ 1 & 0 & 0 & -\text{i} \end{pmatrix} \otimes \text{Id}_2 \,. \tag{A.7}$$

These satisfy the $\mathbb{Z}_4$ clock and shift algebra $Z_j X_j = \text{i} X_j Z_j$ and Pauli matrices

$$\sigma_j^x = \text{Id}_4 \otimes \begin{pmatrix} 0 & 1 \\ 1 & 0 \end{pmatrix} \,, \qquad \sigma_j^x = \text{Id}_4 \otimes \begin{pmatrix} 1 & 0 \\ 0 & -1 \end{pmatrix} \,, \tag{A.8}$$

that satisfy $\sigma_j^x \sigma_j^z = -\sigma_j^z \sigma_j^x$. The action on states is given by

$$\begin{aligned}
X_j |(p_j \,, q_j)\rangle &= |(p_j + 1 \,, q_j)\rangle \,, & Z_j |(p_j \,, q_j)\rangle &= \text{i}^{p_j} |(p_j \,, q_j)\rangle \,, \\
\sigma_j^x |(p_j \,, q_j)\rangle &= |(p_j \,, q_j + 1)\rangle \,, & \sigma_j^z |(p_j \,, q_j)\rangle &= (-1)^{q_j} |(p_j \,, q_j)\rangle \,,
\end{aligned} \tag{A.9}$$

where the summation for $p$ and $q$ is modulo 4 and 2 respectively. The symmetry action on states is by lines in $\mathcal{S} = \mathcal{C}_{\mathcal{M}}^* = \text{Vec}(\mathbb{Z}_4 \times \mathbb{Z}_2)$ acting from below as in (2.7) implemented by the operator

$$\mathcal{U}_{(p \,, q)} = \prod_j X_j^p (\sigma_j^x)^q \,. \tag{A.10}$$

which acts on states as

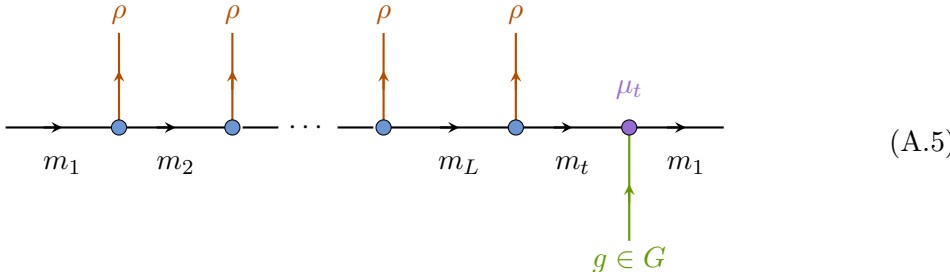

$$\text{(A.11)}$$

**Gapped phases.** There are 9 gapped phases realized in $\mathbb{Z}_4 \times \mathbb{Z}_2$ symmetric quantum systems, labelled by $(H, \beta)$ with $H \subseteq \mathbb{Z}_4 \times \mathbb{Z}_2$ and $\beta \in H^2(H, U(1))$. There are 2 phases each for $H = \mathbb{Z}_4 \times \mathbb{Z}_2$ and $\mathbb{Z}_2 \times \mathbb{Z}_2$, which we label as $(\mathbb{Z}_4 \times \mathbb{Z}_2, \pm)$ and $(\mathbb{Z}_2 \times \mathbb{Z}_2, \pm)$ respectively. The phases labelled by '$-$' are non-trivial SPTs. There is 1 phase corresponding to $H = \mathbb{Z}_4$ labelled as $(\mathbb{Z}_4, *)$ and 3 phases where $\mathbb{Z}_4 \times \mathbb{Z}_2$ is broken down to $\mathbb{Z}_2$ labelled as $(\mathbb{Z}_2^b, *), (\mathbb{Z}_2^{a^2}, *)$ and $(\mathbb{Z}_2^{a^2 b}, *)$. Lastly there is the fully symmetry broken phase labelled as $(\mathbb{Z}_1, *)$.

Fixed-point Hamiltonians for each of these gapped phases can be constructed using the procedure outlined in Sec. 3.1 by picking the Frobenius algebra $A_{(H,\beta)}$. At the level of objects,

$$A_{(H,\beta)} = \oplus_{h \in H} h, \tag{A.12}$$

while the product structure $m : A \otimes A \to A$ and coproduct structure $\Delta : A \to A \otimes A$ in the algebra are determined by $\beta$ as

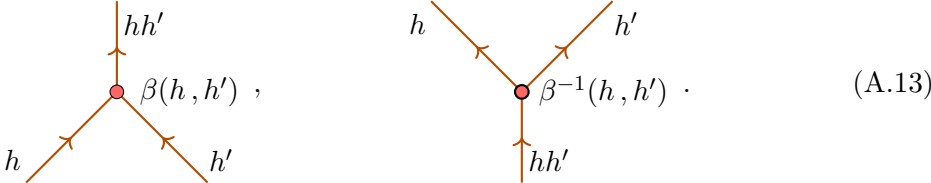

$$\tag{A.13}$$

The order parameters for the different gapped phases are most conveniently understood in terms of the SymTFT construction described in Sec. 3.1. The SymTFT for the present case is $\mathcal{Z}(\mathsf{Vec}_{\mathbb{Z}_4 \times \mathbb{Z}_2})$, that is the $\mathbb{Z}_4 \times \mathbb{Z}_2$ Dijkgraaf-Witten theory. The topological lines in the SymTFT are dyons $d = (g, \hat{g})$ that carry a flux $g \in G$ and charge $\hat{g} \in \mathsf{Rep}(\mathbb{Z}_4 \times \mathbb{Z}_2) \cong \mathbb{Z}_4 \times \mathbb{Z}_2$. We denote the pure flux line that corresponds to $(p, q)$ as $m_4^p m_2^q$. A pure charge corresponding to $(p, q)$ is denoted as $e_4^p e_2^q$.

Within the SymTFT picture, the symmetry and input boundary are

$$\mathfrak{B}^{\mathrm{sym}} = \mathfrak{B}^{\mathrm{inp}} = \langle e_4, e_2 \rangle = \bigoplus_{(p,q)} e_4^p e_2^q. \tag{A.14}$$

The fusion category of lines on both boundaries is $\mathcal{C} = \mathcal{S} = \mathsf{Vec}_{\mathbb{Z}_4 \times \mathbb{Z}_2}$, where the simple objects are provided by the projections of the bulk flux-lines. The two boundaries are separated by an interface that hosts the regular $\mathcal{C}$ module category. The set of possible order parameters corresponds to the set of Bosonic lines in the SymTFT. These are the lines $(g, \hat{g})$ for which $\hat{g}(g) = 1$. The order parameter are realized in the spin chain model for $g \in G$ and $\hat{g} = (p, q)$ by bringing the bulk line onto the boundary such that the two ends are on either side of $\mathcal{M}$ at the $j^{\mathrm{th}}$ site and then shrinking the line. Doing so one finds the following concrete operators

$$\mathcal{O}_{(g,\hat{g}),j} = \mathcal{T}_{g,j} Z_j^{\hat{p}} (\sigma_j^z)^{\hat{q}}, \tag{A.15}$$

where $\mathcal{T}_{g,j}$ acts between symmetry twisted sectors as

$$\mathcal{T}_{g,j}|(\vec{p},\vec{q})\rangle_{g_0} = |(\vec{p},\vec{q})\rangle_{gg_0}. \tag{A.16}$$

We now describe the fixed-point Hamiltonians, ground states and order parameters for each of the gapped phases.

- **Trivial (paramagnetic) phase:** In the SymTFT, this gapped phase is obtained by choosing

$$\mathfrak{B}^{\mathrm{phys}} = \langle m_4\,, m_2 \rangle = \bigoplus_{(p,q)} m_4^p m_2^q, \tag{A.17}$$

where all the bulk SymTFT flux lines have been condensed. To obtain this physical boundary, one has to gauge $H = \mathbb{Z}_4 \times \mathbb{Z}_4$ on the input boundary which corresponds to choosing $(H,\beta) = (\mathbb{Z}_4 \times \mathbb{Z}_2, +)$. The fixed-point Hamiltonian has the form

$$\mathcal{H}_{(\mathbb{Z}_4 \times \mathbb{Z}_2, +)} = -\frac{1}{8}\sum_j \sum_{(p,q)} \left[\begin{array}{c} \overset{(p,q)}{\longrightarrow} \end{array}\right]_j, \tag{A.18}$$

An operator in this Hamiltonian acts on a state as

$$\tag{A.19}$$

which can be expressed in terms of the local operators in (A.7) and (A.8) as

$$\left[\begin{array}{c} \overset{(p,q)}{\longrightarrow} \end{array}\right]_j = X_j^p (\sigma_j^x)^q. \tag{A.20}$$

This Hamiltonian has a unique ground state in each symmetry twisted sector

$$|\mathrm{GS}_{(\mathbb{Z}_4 \times \mathbb{Z}_2, +)}\rangle_g = \frac{1}{|G|^{L/2}} \prod_j \sum_{\vec{p},\vec{q}} |(\vec{p},\vec{q})\rangle_g. \tag{A.21}$$

The multiplet of $|G| = 8$ ground states are mapped into one another by the action of the order parameter $\mathcal{O}_{(g,1),j} = \mathcal{T}_{g,j}$ as

$$\mathcal{T}_{g,j}|\mathrm{GS}_{(\mathbb{Z}_4 \times \mathbb{Z}_2, +)}\rangle_{g'} = |\mathrm{GS}_{(\mathbb{Z}_4 \times \mathbb{Z}_2, +)}\rangle_{gg'}. \tag{A.22}$$

- **$\mathbb{Z}_4 \times \mathbb{Z}_2$ preserving SPT phase:** In the SymTFT, this gapped phase is obtained by choosing

$$\mathfrak{B}^{\text{phys}} = \langle m_4 e_2 \,, m_2 e_4^2 \rangle = \bigoplus_{(p,q)} (m_4 e_2)^p \left( m_2 e_4^2 \right)^q . \tag{A.23}$$

Notice that the physical boundary is generated by dyons instead of pure charges or pure fluxes. This is typical of Abelian group SPTs. This phase is obtained from the input boundary $\mathfrak{B}^{\text{inp}}$ by gauging the full $\mathbb{Z}_4 \times \mathbb{Z}_2$ symmetry on it with a choice of non-trivial discrete torsion. Such a gauging is equivalent to summing over a network of Frobenius algebra objects $A_{(H,\beta)}$ with $H = \mathbb{Z}_4 \times \mathbb{Z}_2$ and the non-trivial 2-cocycle $\beta \in H^2(H, U(1))$ which can be chosen to be

$$\beta((p_1, q_1)\,, (p_2, q_2)) = (-1)^{p_1 q_2} . \tag{A.24}$$

The Hamiltonian has the same form as (A.18), except that the Frobenius algebra product and coproduct are twisted by $\beta$ and $\beta^{-1}$ respectively,

$$\mathcal{H}_{(\mathbb{Z}_4 \times \mathbb{Z}_2, -)} = -\frac{1}{8} \sum_j \sum_{(p,q)} \left[ \begin{array}{c} \vcenter{\hbox{}} \end{array} \right]_j , \tag{A.25}$$

therefore an operator in the Hamiltonian acts on a state as

$$= (-1)^{q(\Delta p)_{j-1/2} + p(\Delta q)_{j+1/2}} \times \tag{A.26}$$

where $(\Delta p)_{j-1/2} := p_j = p_{j-1}$. In terms of (A.7) and (A.8)

$$\left[ \vcenter{\hbox{}} \right]_j = X_j \sigma_j^z \sigma_{j+1}^z \,, \qquad \left[ \vcenter{\hbox{}} \right]_j = Z_{j-1}^2 Z_j^2 \sigma_j^x . \tag{A.27}$$

The operator representations of the remaining choices of $(p,q)$ can be obtained by taking products of these operators. The fixed-point Hamiltonian in the gapped phase $(\mathbb{Z}_4 \times \mathbb{Z}_2 \,, -)$ therefore is

$$\mathcal{H}_{(\mathbb{Z}_4 \times \mathbb{Z}_2 \,, -)} = -\frac{1}{8} \sum_j \sum_{p,q} \left[ X_j \sigma_j^z \sigma_{j+1}^z \right]^p \left[ Z_{j-1}^2 Z_j^2 \sigma_j^x \right]^q . \tag{A.28}$$

Since, the terms in the Hamiltonian all mutually commute, the ground state can be readily obtained and is the state with eigenvalue $+1$ for the stabilizers $X_j \sigma_j^z \sigma_{j+1}^z$ and

$Z_{j-1}^2 Z_j^2 \sigma_j^x$ for all $j$. It has the form

$$|\text{GS}_{(\mathbb{Z}_4 \times \mathbb{Z}_2, -)}\rangle_{(0,0)} = \prod_j \frac{1 + Z_{j-1}^2 Z_j^2 \sigma_j^x}{2} \big| \{X_j = 1, \sigma_j^z = 1\} \big\rangle_{(0,0)},$$ (A.29)

where the state

$$\begin{aligned}
X_j \big| \{X_j = 1, \sigma_j^z = 1\} \big\rangle_{(0,0)} &= \big| \{X_j = 1, \sigma_j^z = 1\} \big\rangle_{(0,0)}, \\
\sigma_j^z \big| \{X_j = 1, \sigma_j^z = 1\} \big\rangle_{(0,0)} &= \big| \{X_j = 1, \sigma_j^z = 1\} \big\rangle_{(0,0)},
\end{aligned}$$ (A.30)

for all $j$. The fact that the ground state is symmetric and uncharged follows from

$$\mathcal{U}_{(1,0)} = \prod_j X_j \sigma_j^z \sigma_{j+1}^z, \qquad \mathcal{U}_{(0,1)} = \prod_j Z_{j-1}^2 Z_j^2 \sigma_j^x.$$ (A.31)

The defining property of this SPT are the charge of its ground state in the symmetry twisted sector. Note that the presence of a symmetry defect corresponding to the $(1,0)$ or $(0,1)$ element in $\mathbb{Z}_4 \times \mathbb{Z}_2$ alters the sign of a single term in the Hamiltonian.

(A.32)

It follows that the fixed-point Hamiltonians in the symmetry twisted sectors corresponding to a group elements $(1,0)$ and $(0,1)$ with the twist inserted at a site $j_0$ are

$$\begin{aligned}
\mathcal{H}_{(\mathbb{Z}_4 \times \mathbb{Z}_2, -);((1,0),j_0)} = &-\frac{1}{8} \sum_{p,q} \sum_{j \neq j_0+1} \big[X_j \sigma_j^z \sigma_{j+1}^z\big]^p \big[Z_{j-1}^2 Z_j^2 \sigma_j^x\big]^q \\
&-\frac{1}{8} \sum_{p,q} \big[X_j \sigma_j^z \sigma_{j+1}^z\big]^p \big[-Z_{j_0}^2 Z_{j_0+1}^2 \sigma_{j_0+1}^x\big]^q, \\
\mathcal{H}_{(\mathbb{Z}_4 \times \mathbb{Z}_2, -);((0,1),j_0)} = &-\frac{1}{8} \sum_{p,q} \sum_{j \neq j_0} \big[X_j \sigma_j^z \sigma_{j+1}^z\big]^p \big[Z_{j-1}^2 Z_j^2 \sigma_j^x\big]^q \\
&-\frac{1}{8} \sum_{p,q} \big[-X_{j_0} \sigma_{j_0}^z \sigma_{j_0+1}^z\big]^p \big[Z_{j-1}^2 Z_j^2 \sigma_j^x\big]^q.
\end{aligned}$$ (A.33)

The corresponding ground states are

$$|\text{GS}_{(\mathbb{Z}_4 \times \mathbb{Z}_2, -)}\rangle_{(1,0)} = \prod_j \frac{1 + (-1)^{\delta_{j,j_0+1}} Z_{j-1}^2 Z_j^2 \sigma_j^x}{2} \big| \{X_j = 1, \sigma_j^z = 1\} \big\rangle_{(1,0)}$$

$$= \sigma_{j_0+1}^z \prod_j \frac{1 + Z_{j-1}^2 Z_j^2 \sigma_j^x}{2} \big| \{X_j = 1, \sigma_j^z = 1\} \big\rangle_{(1,0)},$$

$$|\text{GS}_{(\mathbb{Z}_4 \times \mathbb{Z}_2, -)}\rangle_{(0,1)} = \prod_j \frac{1 + Z_{j-1}^2 Z_j^2 \sigma_j^x}{2} \big| \{X_{j \neq j_0} = 1, X_{j_0} = -1, \sigma_j^z = 1\} \big\rangle_{(0,1)} \tag{A.34}$$

$$= Z_{j_0}^2 \prod_j \frac{1 + Z_{j-1}^2 Z_j^2 \sigma_j^x}{2} \big| \{X_j = 1, \sigma_j^z = 1\} \big\rangle_{(0,1)}.$$

From which it follows that

$$\mathcal{U}_{(1,0)} |\text{GS}_{(\mathbb{Z}_4 \times \mathbb{Z}_2, -)}\rangle_{(0,1)} = -|\text{GS}_{(\mathbb{Z}_4 \times \mathbb{Z}_2, -)}\rangle_{(0,1)},$$

$$\mathcal{U}_{(0,1)} |\text{GS}_{(\mathbb{Z}_4 \times \mathbb{Z}_2, -)}\rangle_{(1,0)} = -|\text{GS}_{(\mathbb{Z}_4 \times \mathbb{Z}_2, -)}\rangle_{(1,0)}. \tag{A.35}$$

From (A.23), we read-off that the order paramaters for this SPT phase. These correspond to the SymTFT lines that have been condensed on the physical boundary. These are

$$\mathcal{O}_{[(p,q), \hat{\beta}_{(p,q)}], j} = \mathcal{O}_{((p,q),(\hat{0},\hat{0})), j} \left(Z_j^2\right)^q \left(\sigma_j^z\right)^p, \tag{A.36}$$

where we have used that

$$\hat{\beta}_{(p,q)}((p', q')) = (-1)^{pq' + qp'} \Rightarrow \hat{\beta}_{(p,q)} = (\hat{2}q, \hat{p}), \tag{A.37}$$

These order parameters act on the multiple of ground states as

$$\mathcal{O}_{((p,q), \hat{\beta}_{(p,q)}), j} |\text{GS}_{(\mathbb{Z}_4 \times \mathbb{Z}_2, -)}\rangle_{(p',q')} = |\text{GS}_{(\mathbb{Z}_4 \times \mathbb{Z}_2, -)}\rangle_{(p+p', q+q')}. \tag{A.38}$$

- $(\mathbb{Z}_2 \times \mathbb{Z}_2, +)$ **phase:** Next we describe the gapped phase that spontaneously breaks the global symmetry to $\mathbb{Z}_2 \times \mathbb{Z}_2$. Each ground state is in the trivial SPT phase for the preserved $\mathbb{Z}_2 \times \mathbb{Z}_2$. The physical SymTFT boundary corresponding to this gapped phase is

$$\mathfrak{B}^{\text{phys}} = \langle m_4^2, e_4^2, m_2 \rangle. \tag{A.39}$$

This phase is obtained by gauging the $\mathbb{Z}_2 \times \mathbb{Z}_2$ symmetry on the input boundary with a choice of trivial discrete torsion. Such a gauging is given by the Frobenius algebra $A_{\mathbb{Z}_2 \times \mathbb{Z}_2, +}$. The fixed-point Hamiltonian in the phase $(\mathbb{Z}_2 \times \mathbb{Z}_2, +)$ has the form

$$\mathcal{H}_{(\mathbb{Z}_2 \times \mathbb{Z}_2, +)} = -\frac{1}{4} \sum_j \sum_{p,q=0,1} \left[ \begin{array}{c} \text{(2p, q)} \\ \bullet \longrightarrow \bullet \end{array} \right]_j,$$

$$= -\sum_j \left[\frac{1 + Z_{j-1}^2 Z_j^2}{2}\right] \left[\frac{1 + Z_j^2 Z_{j+1}^2}{2}\right] \left[\frac{1 + X_j^2}{2}\right] \left[\frac{1 + \sigma_j^x}{2}\right]. \tag{A.40}$$

There are two ground states each in the $g$-twisted sectors for $g \in \{(0,0),(2,0),(0,1),(2,1)\} \simeq \mathbb{Z}_2 \times \mathbb{Z}_2$. These ground states break the global symmetry down to $\mathbb{Z}_2 \times \mathbb{Z}_2$

$$
\begin{aligned}
|\mathrm{GS}_{(\mathbb{Z}_2 \times \mathbb{Z}_2,+)}, 0\rangle_g &= \frac{1}{2^L} \prod_j \sum_{p_j, q_j = 0,1} |(2p_1, q_1), \dots, (2p_L, q_L)\rangle_g, \\
|\mathrm{GS}_{(\mathbb{Z}_2 \times \mathbb{Z}_2,+)}, 1\rangle_g &= \frac{1}{2^L} \prod_j \sum_{p_j, q_j = 0,1} |(2p_1 + 1, q_1), \dots, (2p_L + 1, q_L)\rangle_g.
\end{aligned}
\tag{A.41}
$$

The order parameters that characterize this phase are

$$
\mathcal{O}_{[g,\hat{g}],j}, \qquad g \in \mathbb{Z}_2 \times \mathbb{Z}_2, \hat{g} \in \{(0,0),(2,0)\}.
\tag{A.42}
$$

which act on the ground state subspace as

$$
\begin{aligned}
\mathcal{O}_{[g,(0,0)],j}|\mathrm{GS}_{(\mathbb{Z}_2 \times \mathbb{Z}_2,+)}, p\rangle_{g^*} &= |\mathrm{GS}_{(\mathbb{Z}_2 \times \mathbb{Z}_2,+)}, 0\rangle_{gg^*}, \\
\mathcal{O}_{[g,(2,0)],j}|\mathrm{GS}_{(\mathbb{Z}_2 \times \mathbb{Z}_2,+)}, p\rangle_{g^*} &= (-1)^p |\mathrm{GS}_{(\mathbb{Z}_2 \times \mathbb{Z}_2,+)}, 0\rangle_{gg^*}.
\end{aligned}
\tag{A.43}
$$

- $(\mathbb{Z}_2 \times \mathbb{Z}_2, -)$ **phase:** This phase breaks the global $\mathbb{Z}_4 \times \mathbb{Z}_2$ symmetry down to the $\mathbb{Z}_2 \times \mathbb{Z}_2$ subgroup such that each ground state realizes a non-trivial $\mathbb{Z}_2 \times \mathbb{Z}_2$ SPT. The physical boundary is given by

$$
\mathfrak{B}^{\mathrm{phys}} = \langle m_2 e_4, m_4^2 e_2 \rangle = \bigoplus_{p,q} (m_2 e_4)^p (m_4^2 e_2)^q.
\tag{A.44}
$$

The physical boundary is obtained from the input boundary by gauging $H = \mathbb{Z}_2 \times \mathbb{Z}_2 \subseteq \mathbb{Z}_4 \times \mathbb{Z}_2$ with a choice of discrete torsion $\beta \in H^2(H, U(1))$. We may choose the following representative for $\beta$

$$
\beta((p_1, q_1),(p_2, q_2)) = \mathrm{i}^{p_1 q_2 - p_2 q_1}.
\tag{A.45}
$$

The fixed-point Hamiltonian has the form

$$
\mathcal{H}_{(\mathbb{Z}_2 \times \mathbb{Z}_2, -)} = -\frac{1}{4} \sum_j \sum_{p,q=0,1} \left[ \vcenter{\hbox{}} \right]_j,
\tag{A.46}
$$

Operators appearing in the Hamiltonian can be expressed in terms of the local operators as

$$
\left[ \vcenter{\hbox{}} \right]_j = \left[\frac{1 + Z_{j-1}^2 Z_j^2}{2}\right] \left[\frac{1 + Z_j^2 Z_{j+1}^2}{2}\right] [\sigma_j^z X_j^2 \sigma_{j+1}^z]^p \left[\frac{Z_{j-1}\sigma_j^x Z_j^\dagger + Z_{j-1}^\dagger \sigma_j^x Z_j}{2}\right]^q.
\tag{A.47}
$$

There are two ground states in the untwisted sector that spontaneously break the symmetry down to $\mathbb{Z}_2 \times \mathbb{Z}_2$. These are

$$|\text{GS}_{(\mathbb{Z}_2 \times \mathbb{Z}_2, -)}, p\rangle_{(0,0))} = \prod_j \left[ \frac{1 + \sigma_j^z X_j^2 \sigma_{j+1}^z}{2} \right] \sum_{q_j = 0,1} |(p, q_1), (p, q_2), \ldots, (p, q_L)\rangle_{(0,0)} ,$$

(A.48)

where $p = 0, 1$. Both these ground states are invariant (uncharged) under $\mathbb{Z}_2 \times \mathbb{Z}_2$ generated by

$$\mathcal{U}_{(2,0)} = \prod_j X_j^2 , \qquad \mathcal{U}_{(0,1)} = \prod_j \sigma_j^x , \tag{A.49}$$

and have a $(-1)^p$ eigenvalue under $Z_j^2$ which serves as a symmetry breaking order parameter. To diagnose the SPT nature of this phase, we need to inspect the ground states in the symmetry twisted sectors. To do so, we note that in the presence of a symmetry twist, a single term (crossing the twist) in the Hamiltonian gets modified. For instance,

(A.50)

which leads to a change of sign of the operator $Z_{j-1} Z_j^\dagger \sigma_j^x$ at the location of the $(2,0)$ symmetry twist. The corresponding ground state can be created by inserting a local charge $\sigma_j^z$ with respect to the untwisted ground state and then acting with all the stabilizers. Similarly, the symmetry twist $(0,1)$ has the effect

(A.51)

which alters the sign of $\sigma_j^z X_j^2 \sigma_{j+1}^z$ at the location of the symmetry defect. Again, the corresponding ground state is created by inserting a charge $Z_j$ with respect to the untwisted ground state. The ground states in the twisted Hilbert spaces are

$$|\text{GS}_{(\mathbb{Z}_2 \times \mathbb{Z}_2, -)}, p\rangle_{(2p_0, q_0))}$$
$$= Z_j^{q_0} \left( \sigma_j^z \right)^{p_0} \prod_j \left[ \frac{1 + \sigma_j^z X_j^2 \sigma_{j+1}^z}{2} \right] \sum_{q_j = 0,1} |(p, q_1), \ldots, (p, q_L)\rangle_{(2p_0, q_0)} ,$$

(A.52)

The order parameters for this SPT phase are the symmetry breaking order parameter $Z_j^2$ and the string order parameter

$$\mathcal{O}_{[(2p,q),\hat{\beta}_{(2p,q)}],j} = \mathcal{O}_{[(2p,q),(-\hat{q},\hat{p})],j} = \mathcal{O}_{[(2p,q),(\hat{0},\hat{0})],j} Z_j^{-q} \left(\sigma_j^z\right)^p . \tag{A.53}$$

- $(\mathbb{Z}_2^{a^2}, *)$ **phase:** This phase breaks the global symmetry down to a $\mathbb{Z}_2$ subgroup generated by $a^2 \equiv (2,0)$. The Hamiltonian has the form

$$\mathcal{H}_{(\mathbb{Z}_2^{a^2},*)} = -\frac{1}{2} \sum_j \left\{ \left[ \; \overset{(0,0)}{\longrightarrow} \; \right]_j + \left[ \; \overset{(2,0)}{\longrightarrow} \; \right]_j \right\} , \tag{A.54}$$

which is stabilized by the set of operators

$$\left\{ \sigma_{j-1}^z \sigma_j^z , \quad Z_{j-1}^2 Z_j^2 , \quad X_j^2 \right\} . \tag{A.55}$$

The ground states lie in the sectors twisted by $g \in \{(0,0), (2,0)\} \simeq \mathbb{Z}_2^{a^2}$ and in each of these twisted sectors there are 4 symmetry broken ground states

$$|\text{GS}_{(\mathbb{Z}_2^{a^2},*)}, (p,q)\rangle_g = \frac{1}{2^{L/2}} \prod_j \sum_{p_j=0,1} |(p+2p_1,q_1),\ldots,(p+2p_L,q_L)\rangle_g . \tag{A.56}$$

The set of order parameters is

$$\left\{ Z_j^2 , \quad \sigma_j^z , \quad \mathcal{O}_{[(2,0),(0,0)],j} \right\} , \tag{A.57}$$

which act on the ground states as

$$Z_j^2 |\text{GS}_{(\mathbb{Z}_2^{a^2},*)}, (p,q)\rangle_g = (-1)^p |\text{GS}_{(\mathbb{Z}_2^{a^2},*)}, (p,q)\rangle_g ,$$
$$\sigma_j^z |\text{GS}_{(\mathbb{Z}_2^{a^2},*)}, (p,q)\rangle_g = (-1)^q |\text{GS}_{(\mathbb{Z}_2^{a^2},*)}, (p,q)\rangle_g , \tag{A.58}$$
$$\mathcal{O}_{[(2,0),(0,0)],j} |\text{GS}_{(\mathbb{Z}_2^{a^2},*)}, (p,q)\rangle_g = |\text{GS}_{(\mathbb{Z}_2^{a^2},*)}, (p,q)\rangle_{(2,0)\cdot g} .$$

The properties of the remaining gapped phases can be determined similarly.

# B  Example: non-Abelian symmetry $\mathcal{S} = S_3$

In this appendix, we present the $S_3$ symmetric anyon chain model and study the corresponding $S_3$ protected gapped phases. We present the finite group $S_3$ as

$$S_3 = \langle a, b \mid a^3 = 1, b^2 = 1, bab = a^2 \rangle . \tag{B.1}$$

**Setup.** To construct the $S_3$ symmetric lattice model, we choose

$$\mathcal{C} = \mathcal{M} = \mathsf{Vec}_{S_3}, \quad \rho = \bigoplus_{g \in S_3} g. \tag{B.2}$$

The Hilbert space is spanned by the basis states $|\vec{g}\rangle = |g_1, g_2, \ldots, g_L\rangle$, with $g_j \in G$.

$$\tag{B.3}$$

This Hilbert space admits a tensor decomposition into local Hilbert spaces associated to the vertices of the lattice. The Hilbert space $\mathcal{V}_j$ assigned to the $j^{\text{th}}$ vertex is isomorphic to the group algebra

$$\mathcal{V}_j = \mathbb{C}[S_3] \cong \mathrm{Span}_{\mathbb{C}} \left\{ |g\rangle \ \middle| \ g \in S_3 \right\}. \tag{B.4}$$

We define the following operators acting on $\mathcal{V}_j$

$$L_j^h |g_j\rangle = |hg_j\rangle, \quad R_j^h |g_j\rangle = |g_j h\rangle, \tag{B.5}$$

and for each irreducible representation $\Gamma \in \mathrm{Rep}(S_3)$, we define operators that act diagonally on the basis states as

$$(Z_{IJ}^{\Gamma})_j |g_j\rangle = \left[\mathcal{D}^{\Gamma}(g_j)\right]_{IJ} |g_j\rangle, \tag{B.6}$$

where $I, J = 1, \ldots, \dim(\Gamma)$ and $\mathcal{D}_{\Gamma}$ is the matrix representation of $\Gamma$. When $\dim(\Gamma) = 1$, we will suppress the indices $I, J$. The lattice systems defined as anyon chains via the data (B.2) are $S_3$ symmetric where the $S_3$ symmetry is represented as

$$\mathcal{U}_g = \prod_j R_j^g. \tag{B.7}$$

The $S_3$ symmetry operators act on the basis states as

$$\tag{B.8}$$

and on the operators as

$$\begin{aligned} \mathcal{U}_g L_j^h \mathcal{U}_g^{-1} &= L_j^h \\ \mathcal{U}_g R_j^h \mathcal{U}_g^{-1} &= R_j^{ghg^{-1}} \\ \mathcal{U}_g (Z_{IJ}^{\Gamma})_j \mathcal{U}_g^{-1} &= (Z_{IK}^{\Gamma})_j \left[\mathcal{D}^{\Gamma}(g^{-1})\right]_{KJ} \end{aligned} \tag{B.9}$$

Hence the local operators $(Z^\Gamma_{IJ})_j$ for $J = 1, \ldots, \dim(\Gamma)$ form a $\dim(\Gamma)$ dimensional $S_3$ multiplet transforming in the $\Gamma$ representation. The group $S_3$ has three irreducible representations denoted as $1, P$ and $E$ for which $\dim(1) = \dim(P) = 1$ and $\dim(E) = 2$. The charged operators $Z^\Gamma$ transforming in these representations have the form (see also [49, 78])

$$
\begin{aligned}
Z^1 &= \mathbb{I}, \\
Z^P &= |1\rangle\langle 1| + |a\rangle\langle a| + |a^2\rangle\langle a^2| - |b\rangle\langle b| - |ab\rangle\langle ab| - |a^2b\rangle\langle a^2b|, \\
Z^E_{11} &= |1\rangle\langle 1| + \omega|a\rangle\langle a| + \omega^2|a^2\rangle\langle a^2|, \\
Z^E_{12} &= |b\rangle\langle b| + \omega|ab\rangle\langle ab| + \omega^2|a^2b\rangle\langle a^2b|, \\
Z^E_{21} &= |b\rangle\langle b| + \omega^2|ab\rangle\langle ab| + \omega|a^2b\rangle\langle a^2b|, \\
Z^E_{22} &= |1\rangle\langle 1| + \omega^2|a\rangle\langle a| + \omega|a^2\rangle\langle a^2|.
\end{aligned}
\tag{B.10}
$$

Since we are studying $S_3$ symmetric models, it is natural to consider Hilbert spaces twisted by elements $g \in S_3$. For each $g \in G$, there is a symmetry twisted Hilbert space spanned by states $|\vec{g}\rangle_g := |g_1, g_2, \ldots, g_L\rangle_g$, corresponding to fusion trees

$$\tag{B.11}$$

We denote the $g$-twisted Hilbert space as $\mathcal{V}_g$. Then the full Hilbert space is the direct sum

$$
\mathcal{V} = \bigoplus_g \mathcal{V}_g.
\tag{B.12}
$$

The symmetry action on basis states in symmetry twisted sectors is given by

$$
\mathcal{U}_h|g_1, g_2, \ldots, g_L\rangle_g = |g_1 h, g_2 h, \ldots, g_L h\rangle_{h^{-1}gh},
\tag{B.13}
$$

which can be understood diagramatically as

$$\tag{B.14}$$

We also consider twisted sector operators. These map between different symmetry twisted sectors as

$$
\mathcal{T}_h|g_1, g_2, \ldots, g_L\rangle_g = |g_1, g_2, \ldots, g_L\rangle_{hg}.
\tag{B.15}
$$

Under $S_3$ action, the twisted sector operators transform as

$$
\mathcal{U}_g \mathcal{T}_h \mathcal{U}_g^{-1} = \mathcal{T}_{g^{-1}hg}.
\tag{B.16}
$$

**SymTFT and $S_3$ charges.** The SymTFT for the present case is the $S_3$ Dijkgraaf-Witten theory $\mathcal{Z}(\mathsf{Vec}_{S_3})$. The bulk topological lines of the SymTFT are given in eq. (2.76). From these, the bosonic lines are

$$([\mathrm{id}], 1), \quad ([\mathrm{id}], 1_-), \quad ([\mathrm{id}], \mathrm{E}), \quad ([\mathrm{a}], 1), \quad ([\mathrm{b}], +). \tag{B.17}$$

In order to construct the $\mathsf{Vec}_{S_3}$ model, we need to specify an input and symmetry topological boundary of the SymTFT. For the present case, these are the same

$$\mathfrak{B}^{\mathrm{sym}} = \mathfrak{B}^{\mathrm{inp}} = ([\mathrm{id}], 1) \oplus ([\mathrm{id}], 1_-) \oplus 2([\mathrm{id}], \mathrm{E}). \tag{B.18}$$

The category of topological lines on these topological boundaries is given by $\mathsf{Vec}_{S_3}$ whose isomorphism classes of simple lines are labelled by group elements in $S_3$. The bulk (pure flux) lines that carry labels of conjugacy classes in $S_3$ project onto the boundary as

$$
\begin{aligned}
([a], 1) &\longmapsto a \oplus a^2, \\
([b], +) &\longmapsto b \oplus ab \oplus a^2 b.
\end{aligned}
\tag{B.19}
$$

The set of SymTFT bosonic lines (B.17) aids in the organization of local operators in the lattice models into $S_3$ multiplets. More precisely the different $S_3$ multiplets can be labelled by bosonic lines in the SymTFT. Here we describe the structure of the different $S_3$ multiplets in the lattice model. These will play the role of order parameters for different gapped phases in what follows. Firstly the pure charges $([\mathrm{id}], 1_-)$ and $([\mathrm{id}], \mathrm{E})$ become the local (untwisted sector) operators that tranform in the $P$ and $E$ representation respectively. More concretely, the $([\mathrm{id}], 1_-)$ line becomes

$$\mathcal{O}_{P,j} = Z_j^P, \tag{B.20}$$

while the $([\mathrm{id}], \mathrm{E})$ line gives rise to two 2-dimensional multiplets as it has 2 ends on both the input and symmetry boundaries. Specifically the two ends on the input boundary correspond to the multiplet labels while the ends on the symmetry boundary fix the dimensionality of the multiplet. These multiplets are

$$\mathcal{O}_{E_1,j} \cong \{Z_{11,j}^E, Z_{12,j}^E\}, \qquad \mathcal{O}_{E_1,j} \cong \{Z_{21,j}^E, Z_{22,j}^E\}. \tag{B.21}$$

It can be checked that these multiplets satisfy composition rules that are consistent with $\mathsf{Rep}(S_3)$ fusion rules.

$$
\begin{aligned}
\mathcal{O}_{P,j} \times \mathcal{O}_{P,j} &= 1, \\
\mathcal{O}_{E_1,j} \times \mathcal{O}_{P,j} &= (Z_{11,j}^E \oplus Z_{12,j}^E) \times \mathcal{O}_{P,j} = (Z_{11,j}^E \oplus -Z_{12,j}^E) \cong \mathcal{O}_{E_1,j}, \\
\mathcal{O}_{E_1,j} \times \mathcal{O}_{E_1,j} &= (Z_{11,j}^E \oplus Z_{12,j}^E) \times (Z_{11,j}^E \oplus Z_{12,j}^E) \\
&= (Z_{22,j}^E \oplus Z_{21,j}^E) \oplus 0 \oplus 0 = \mathcal{O}_{E_2,j} \oplus 0 \oplus 0,
\end{aligned}
\tag{B.22}
$$

In the last line the transformation properties of $0 \oplus 0$ are consistent with those of $\mathcal{O}_{1,j} \oplus \mathcal{O}_{P,j}$. The order parameters corresponding to the remaining two SymTFT bosonic lines i.e., $([a], 1)$ and $([b], +)$ are twisted sector operators. They map between different twisted sectors and are of the form

$$
\begin{aligned}
\mathcal{O}_{a,j} &= \mathcal{T}_{a,j} \oplus \mathcal{T}_{a^2,j}\,, \\
\mathcal{O}_{b,j} &= \mathcal{T}_{b,j} \oplus \mathcal{T}_{ab,j} \oplus \mathcal{T}_{a^2b,j}\,.
\end{aligned}
\tag{B.23}
$$

Given the transformation properties of twisted sector operators under the $S_3$ symmetry in (B.16), it follwos that they form multiplets that contain operators labelled by elements in conjugacy classes. The composition rules of all these multiplets are compatible with the fusions of lines in the SymTFT.

$S_3$ **Gapped Phases.** We now describe the different gapped phases for the lattice model with $S_3$ symmetry. The different gapped phases are classified by Frobenius algebras in $\mathsf{Vec}_{S_3}$ for which there are four choices corresponding to the four subgroups of $S_3$

$$
A_H = \bigoplus_{h \in H} h\,, \qquad H \subseteq S_3\,.
\tag{B.24}
$$

In the SymTFT, this gapped phase is obtained by gauging $H \subseteq S_3$ on $\mathfrak{B}^{\mathrm{inp}}$ to obtain the physical boundary $\mathfrak{B}^{\mathrm{phys}}$. We are interested in characterizing these gapped phases via properties encoded in their fixed-point Hamiltonians and ground states thereof. The untwisted sector fixed-point Hamiltonian in the gapped phase corresponding to $A_H$ is denoted $\mathcal{H}_H$. In general all such Hamiltonians, being $S_3$ symmetric can also be defined in the presence of $S_3$-symmetry defects. A collection of static $S_3$ defects on the lattice is an assignment of $S_3$ group elements to the edges of the lattice, i.e., an $S_3$ background gauge field $A$. We also consider Hamiltonians in the presence of such defects denoted as $\mathcal{H}_H(A)$.

The general form of an operator in the Hamiltonians we consider is

$$
\mathcal{O}_H = -\frac{1}{2} \sum_j \sum_{h, h_L, h_R} \left[ \begin{array}{c} \overset{h}{\underset{h_L \qquad h_R}{\bullet\!\!-\!\!\!\longrightarrow\!\!\!-\!\!\bullet}} \end{array} \right]_j = -\frac{1}{|H|} P^{(H)}_{j-\frac{1}{2}} L^h_j P^{(H)}_{j+\frac{1}{2}}\,, ,
\tag{B.25}
$$

where $h, h_L, h_R \in H$. In terms of the local lattice operators defined previously, this has the form

$$
\mathcal{H}_H = -\frac{1}{|H|} \sum_j \sum_{h \in H} P^{(H)}_{j-\frac{1}{2}} L^h_j P^{(H)}_{j+\frac{1}{2}}\,,
\tag{B.26}
$$

where $P_{j+\frac{1}{2}}$ is an operator that projects onto a subspace of states for which $g_{j+1} g_j^{-1} \in H$. For different choices of subgroup $H$, this projector takes the form (similar Hamiltonians were

discssed in [51])

$$P^{(S_3)}_{j+\frac{1}{2}} = \mathbb{I}_{j,j+1}\,,$$

$$P^{(\mathbb{Z}_3)}_{j+\frac{1}{2}} = \frac{1}{2}\sum_{\Gamma=1,P} Z^\Gamma_j (Z^\Gamma_{j+1})^\dagger\,,$$

$$P^{(\mathbb{Z}_2)}_{j+\frac{1}{2}} = \frac{1}{3}\left[\mathbb{I}_{j,j+1} + \sum_{IJ}\left(Z^E_j \cdot (Z^E_{j+1})^\dagger\right)_{IJ}\right]\,,$$

$$P^1_{j+\frac{1}{2}} = \sum_\Gamma \frac{\dim(\Gamma)}{|G|}\mathrm{Tr}\left(Z^\Gamma_j \cdot (Z^\Gamma_{j+1})^\dagger\right)\,. \tag{B.27}$$

With the purpose of defining models in the presence of symmetry defects, we also define symmetry twisted projectors. For a symmetry twist $A_{j+\frac{1}{2}} \in S_3$ (which in our convention is a $A_{j+\frac{1}{2}}$ symmetry defect on the anyon chain at the site $j$), there are the following twisted projectors

$$P^{(S_3)}_{j+\frac{1}{2}}(A_{j+1/2}) = \mathbb{I}_{j,j+1}$$

$$P^{(\mathbb{Z}_3)}_{j+\frac{1}{2}}(A_{j+1/2}) = \frac{1}{2}\sum_{\Gamma=1,P} Z^\Gamma_j \mathcal{D}^\Gamma(A_{j+1/2})(Z^\Gamma_{j+1})^\dagger$$

$$P^{(\mathbb{Z}_2)}_{j+\frac{1}{2}}(A_{j+1/2}) = \frac{1}{3}\left[\mathbb{I}_{j,j+1} + \sum_{IJ}\left(Z^E_j \cdot \mathcal{D}^E(A_{j+1/2}) \cdot (Z^E_{j+1})^\dagger\right)_{IJ}\right]$$

$$P^1_{j+\frac{1}{2}}(A_{j+1/2}) = \sum_\Gamma \frac{\dim(\Gamma)}{|G|}\mathrm{Tr}\left(Z^\Gamma_j \cdot \mathcal{D}^\Gamma(A_{j+1/2}) \cdot (Z^\Gamma_{j+1})^\dagger\right) \tag{B.28}$$

Using these $S_3$-twisted projectors, the Hamiltonians coupled to a background $S_3$ gauge field $A$ has the form

$$\mathcal{H}_H(A) = -\frac{1}{|H|}\sum_j \sum_{h\in H} L^h_j P^{(H)}_{j-\frac{1}{2}}(A_{j-1/2})P^{(H)}_{j+\frac{1}{2}}(A_{j+1/2})\,. \tag{B.29}$$

**Trivial phase:** The trivial phase corresponds to choosing $H = S_3$ and therefore gauging the full $\mathsf{Vec}_{S_3}$ symmetry on the input boundary which furnishes

$$\mathfrak{B}^{\mathrm{phys}}_{S_3} = ([\mathrm{id}], 1) \oplus ([a], 1) \oplus ([b], +)\,. \tag{B.30}$$

The fixed-point Hamiltonian in the untwisted sector simplifies to

$$\mathcal{H}_{S_3} = -\frac{1}{6}\sum_j \sum_h L^h_j\,, \tag{B.31}$$

where $L^h_j$ was defined in (B.5). The Hamiltonian remains invariant in any symmetry twisted sector because its dependence on $S_3$ symmetry twists is solely through the projection operators,

which do not appear in this fixed-point Hamiltonian. There is a unique product state ground state in each twisted sector

$$|\text{GS}\rangle_g = \frac{1}{|G|^{L/2}} \prod_j \sum_{g_j} |g_1, g_2, \ldots, g_L\rangle_g . \tag{B.32}$$

The $S_3$ symmetry is represented on the ground states as

$$\mathcal{U}_h |\text{GS}\rangle_g = |\text{GS}\rangle_{h^{-1}gh} . \tag{B.33}$$

The order parameters corresponding to the SymTFT lines on the physical boundary are $\mathcal{O}_{a,j}$ and $\mathcal{O}_{b,j}$. These act on the ground states as

$$\mathcal{T}_{g_1,j} |\text{GS}\rangle_{g_2} = |\text{GS}\rangle_{g_1 g_2} . \tag{B.34}$$

$\mathbb{Z}_2$ **SSB phase:** In the SymTFT, this gapped phase corresponds to gauging $H = \mathbb{Z}_3$ on the input boundary. Doing so, furnishes the physical boundary

$$\mathfrak{B}^{\text{phys}}_{\mathbb{Z}_3} = ([\text{id}], 1) \oplus ([\text{id}], 1_-) \oplus 2([a], 1) . \tag{B.35}$$

The fixed-point Hamiltonian becomes

$$\mathcal{H}_{\mathbb{Z}_3} = -\frac{1}{3} \sum_j \sum_h P^{(\mathbb{Z}_3)}_{j-\frac{1}{2}} L^j_j P^{(\mathbb{Z}_3)}_{j+\frac{1}{2}} , \tag{B.36}$$

where $h \in \{1, a, a^2\}$. The operators comprising the Hamiltonian mutually commute

$$\left[ P^{(\mathbb{Z}_3)}_{j+\frac{1}{2}}, \frac{1}{3} \sum L^h_{j'} \right] = 0 , \qquad \forall j, j' . \tag{B.37}$$

The ground states can hence be obtained by separately projecting onto the $+1$ eigen spaces of each of these operators. The simultaneous $+1$ eigenspace of $P^{(\mathbb{Z}_3)}_{j+\frac{1}{2}}$ decomposes into a direct sum of two vector spaces $V_0$ and $V_1$ where $V_q$ ($q = 0, 1$) is spanned by states $|\vec{g}\rangle$ for which $g_j = a^p b^q$. Meanwhile, $\frac{1}{3} \sum L^h_j$ serves to disorder within these two spaces. Therefore there are two ground states which are equal weight superpositions of basis states in $V_q$

$$|\text{GS}, q\rangle = \frac{1}{3^{L/2}} \prod_j \sum_{\vec{p}} |a^{p_1} b^q, a^{p_2} b^q, \ldots, a^{p_L} b^q\rangle . \tag{B.38}$$

These ground states are mapped into one another under the action of $\mathcal{U}_b$

$$\mathcal{U}_b : |\text{GS}, 0\rangle \longleftrightarrow |\text{GS}, 1\rangle , \tag{B.39}$$

while they are left invariant by $\mathbb{Z}_3 \subset S_3$ generated by $\mathcal{U}_a$, therefore this phase is referred to as the $\mathbb{Z}_2$ SSB phase.

Next we study the twisted sector ground states. Twisting by a group element in the $[a]$ conjugacy class leaves the Hamiltonian invariant since

$$P^{(\mathbb{Z}_3)}_{j+\frac{1}{2}}(a^p) = P^{(\mathbb{Z}_3)}_{j+\frac{1}{2}}(1)\,. \tag{B.40}$$

We again find two ground states in each corresponding twisted Hilbert space

$$
\begin{aligned}
|\mathrm{GS}, q\rangle_a &= \frac{1}{3^{L/2}} \sum_{\vec{p}} |a^{p_1}b^q, a^{p_2}b^q, \dots, a^{p_L}b^q\rangle_a\,, \\
|\mathrm{GS}, q\rangle_{a^2} &= \frac{1}{3^{L/2}} \sum_{\vec{p}} |a^{p_1}b^q, a^{p_2}b^q, \dots, a^{p_L}b^q\rangle_{a^2}\,.
\end{aligned}
\tag{B.41}
$$

Since $b : a \leftrightarrow a^2$, we obtain

$$\mathcal{U}_b : |\mathrm{GS}, q\rangle_a \longleftrightarrow |\mathrm{GS}, [q+1]_2\rangle_{a^2}\,, \tag{B.42}$$

Pictorially, the mapping of twisted sector ground states under the action of $\mathcal{U}_b$ follows from

$$\tag{B.43}$$

Next, consider $[b]$ twisted sectors. The corresponding symmetry twisted Hamiltonian has the following projection operator at say a single link $j_0$

$$P^{(\mathbb{Z}_3)}_{j_0+\frac{1}{2}}(a^p b) = \frac{1}{2}\left\{\mathbb{I}_{j_0,j_0+1} - Z^P_{j_0}Z^P_{j_0+1}\right\}\,. \tag{B.44}$$

There is no state that is in the $+1$ eigenspace of all projectors for such a symmetry twist and therefore there are no $[b]$-twisted ground states.

The order parameters that characterize this gapped phase are $\{\mathcal{O}_{P,j}, \mathcal{O}_{a,j}\}$. Their action on the ground states is

$$
\begin{aligned}
{}_{a^{p'}}\langle \mathrm{GS}, q'|Z^P_j|\mathrm{GS}, q\rangle_{a^p} &= \delta_{q,q'}\delta_{p,p'}(-1)^q\,, \\
{}_{a^{p'}}\langle \mathrm{GS}, q'|\mathcal{T}_{a,j}|\mathrm{GS}, q\rangle_{a^p} &= \delta_{q,q'}\delta_{p+1,p'}\,.
\end{aligned}
\tag{B.45}
$$

$\mathbb{Z}_3$ **SSB phase:** This phase corresponds to gauging $\mathbb{Z}_2^b$ on the input boundary of the SymTFT. Doing so gives

$$\mathfrak{B}^{\mathrm{phys}}_{\mathbb{Z}_3} = ([\mathrm{id}], 1) \oplus ([\mathrm{id}], E) \oplus ([b], +)\,. \tag{B.46}$$

The Hamiltonian can be solved to obtain three ground states

$$|\text{GS}, 0\rangle = \frac{1}{2^{L/2}} \prod_j \sum_{g_j \in \mathsf{M}_0} |g_1, g_2, \dots, g_L\rangle,$$

$$|\text{GS}, 1\rangle = \frac{1}{2^{L/2}} \prod_j \sum_{g_j \in \mathsf{M}_1} |g_1, g_2, \dots, g_L\rangle, \tag{B.47}$$

$$|\text{GS}, 2\rangle = \frac{1}{2^{L/2}} \prod_j \sum_{g_j \in \mathsf{M}_2} |g_1, g_2, \dots, g_L\rangle.$$

where $\mathsf{M}_0 = \{1, b\}$, $\mathsf{M}_1 = \{a, a^2 b\}$ and $\mathsf{M}_2 = \{a^2, ab\}$. The $S_3$ action on these ground states is

$$\mathcal{U}_b |\text{GS}, 0\rangle = |\text{GS}, 0\rangle$$

$$\mathcal{U}_b |\text{GS}, 1\rangle = |\text{GS}, 2\rangle$$

$$\mathcal{U}_b |\text{GS}, 2\rangle = |\text{GS}, 1\rangle \tag{B.48}$$

$$\mathcal{U}_a |\text{GS}, j\rangle = |\text{GS}, j + 1 \bmod 3\rangle.$$

These ground states can be distinguished by the expectation values of the operators transforming in the $E$-representation as

$$\langle \text{GS}, p | (Z^E_{IJ})_j | \text{GS}, p' \rangle \propto \delta_{p,p'} \omega^{pJ}. \tag{B.49}$$

Let us now move onto the twisted sector Hamiltonians. In the presence of $g$ symmetry twist at the $j$-th site, we need to define the Hamiltonian using the twisted projector in (B.28) which contains operators of the form

$$Z^E_j \cdot \mathcal{D}_E(g) \cdot (Z^E)^\dagger_{j+1} = \begin{pmatrix} Z^E_{11} & Z^E_{12} \\ Z^E_{21} & Z^E_{22} \end{pmatrix}_j \cdot \mathcal{D}_E(g) \cdot \begin{pmatrix} Z^E_{22} & Z^E_{12} \\ Z^E_{21} & Z^E_{11} \end{pmatrix}_{j+1} \tag{B.50}$$

Using the explicit form of $Z^E_{IJ}$ in (B.10) and the matrix representations for $E$, one finds that the twisted projectors have the following image on the degrees on the sites $(j, j+1)$

$$\text{im}\left[P^{(\mathbb{Z}_3)}_{j+\frac{1}{2}}(a^p)\right] = \left\{ |\vec{g}\rangle \mid g_j \in \mathsf{M}_q, g_{j+1} \in \mathsf{M}_{q+p \bmod 3} \right\}$$

$$\text{im}\left[P^{(\mathbb{Z}_3)}_{j+\frac{1}{2}}(b)\right] = \left\{ |\vec{g}\rangle \mid g_j, g_{j+1} \in \mathsf{M}_0 \right\} \bigcup \left\{ |\vec{g}\rangle \mid g_j \in \mathsf{M}_{1,2}, g_{j+1} \in \mathsf{M}_{2,1} \right\},$$

$$\text{im}\left[P^{(\mathbb{Z}_3)}_{j+\frac{1}{2}}(ab)\right] = \left\{ |\vec{g}\rangle \mid g_j, g_{j+1} \in \mathsf{M}_1 \right\} \bigcup \left\{ |\vec{g}\rangle \mid g_j \in \mathsf{M}_{0,2}, g_{j+1} \in \mathsf{M}_{2,0} \right\}, \tag{B.51}$$

$$\text{im}\left[P^{(\mathbb{Z}_3)}_{j+\frac{1}{2}}(a^2 b)\right] = \left\{ |\vec{g}\rangle \mid g_j, g_{j+1} \in \mathsf{M}_2 \right\} \bigcup \left\{ |\vec{g}\rangle \mid g_j \in \mathsf{M}_{0,1}, g_{j+1} \in \mathsf{M}_{1,0} \right\}.$$

We immediately see that the ground states in the $a^p$ twisted sector $p \neq 0$ have higher energy as compared with the untwisted sector ground states since there is no way to simultaneosly

satisfy the projectors on all the sites. Equivalently the union of the images of all the projectors is empty. Meanwhile there is a single twisted sector ground state in each of the $[b]$-twisted sectors. These are

$$|\mathrm{GS}\rangle_{a^p b} = \frac{1}{2^{L/2}} \prod_j \sum_{g_j \in \mathsf{M}_p} |g_1, g_2, \ldots, g_L\rangle_{a^p b}. \tag{B.52}$$

The $S_3$ action on the twisted sector ground states takes the form

$$\mathcal{U}_g |\mathrm{GS}\rangle_{a^p b} = |\mathrm{GS}\rangle_{g^{-1}(a^p b)g} \tag{B.53}$$

The order parameters are expected to be in the multiplets $\mathcal{O}_{E_I}$ and $\mathcal{O}_b$ as the corresponding lines are condensed on the physical boundary in the SymTFT. We find

$$_{a^p b}\langle \mathrm{GS}, p | (Z_{IJ}^E)_j | \mathrm{GS}, \rangle_{a^{p'} b} \propto \delta_{p,p'} \omega^{pJ}. \tag{B.54}$$

Similarly the operators in the multiplet $\mathcal{O}_b$ map between twisted and untwisted sector ground states as

$$\mathcal{T}_{a^p b} |\mathrm{GS}, p\rangle = |\mathrm{GS}\rangle_{a^p b}. \tag{B.55}$$

$S_3$ **SSB phase:**   This phase corresponds to the algebra

$$\mathcal{A}_1 = 1, \quad \mathfrak{B}^{\mathrm{phys}} = \mathfrak{B}^{\mathrm{inp}}, \tag{B.56}$$

with corresponding Hamiltonian

$$\mathcal{H}_{S_3} = -\sum_j \left[ \raisebox{-1em}{\includegraphics{placeholder}} \right]_j . \tag{B.57}$$

The resulting gapped phase can be equivalently produced using the Hamiltonian

$$\tilde{\mathcal{H}}_1 = -\sum_j \sum_{\Gamma \in \mathsf{Rep}(S_3)} \frac{\dim(\Gamma)}{6} \mathrm{Tr}\left( Z_j^\Gamma \cdot (Z_{j+1}^\Gamma)^\dagger \right) \tag{B.58}$$

which favors an ordering in the $S_3$ degrees of freedom, i.e. $g_i = g_{i+1} = g$. Therefore, we get 6 ground states in the untwisted sector labelled by $g \in S_3$

$$|\mathrm{GS}, g\rangle = |g, \ldots, g\rangle. \tag{B.59}$$

The action of the $S_3$ generators on these ground states is

$$\mathcal{U}_h |\mathrm{GS}, g\rangle = \mathcal{U}_h |\mathrm{GS}, gh\rangle \tag{B.60}$$

and we see that the full $S_3$ symmetry is spontaneously broken. All the local operators $Z^P$ and $Z_{IJ}^E$ described in (B.10) have a non-trivial vev in these ground states and act as order parameters for the gapped phase. Notice that since these ground states are not invariant under any element of $S_3$, we cannot twist by any element, and there is no state in a non-trivial twisted sector.

# C    Example: $\mathcal{S} = \mathsf{Rep}(S_3)$ with $\mathcal{M} = \mathsf{Vec}$

### $\mathsf{Rep}(S_3)$ chain definition

Before specializing to the case of $G = S_3$, we describe the construction of a model with a $\mathsf{Rep}(G)$ symmetry for a general finite non-Abelian group $G$. To construct such a model, we pick

$$\mathcal{C} = \mathsf{Vec}_G, \quad \mathcal{M} = \mathsf{Vec}, \quad \rho = \bigoplus_{g \in G} g. \tag{C.1}$$

This choice can be understood as starting from a model with $G$ finite symmetry and gauging the full symmetry group to obtain a dual model with $\mathsf{Rep}(G)$ symmetry. The degrees of freedom are assigned to the morphism spaces, i.e., which we identify as the edges of a one-dimensional lattice. The Hilbert space is spanned by basis states

$$\tag{C.2}$$

where $g_{j+1/2} \in G$ and $g$ is the $G$ holonomy around the spatial cycle, i.e.,

$$g \equiv g_{1/2} \cdot g_{3/2} \cdots g_{L-1/2}. \tag{C.3}$$

The symmetry of this model is given by $\mathcal{C}^*_{\mathcal{M}}$ which is the dual category to $\mathcal{C}$ with respect to the module category $\mathcal{M}$. For the present case $\mathcal{C}^*_{\mathcal{M}} = \mathsf{Rep}(G)$. The simple objects in $\mathsf{Rep}(G)$ are finite-dimensional unitary irreducible representations of $G$ with the fusion structure given by the tensor product of representations and the additive structure given by the direct sum. More precisely an object $\Gamma \in \mathsf{Rep}(G)$ is the pair $(\mathcal{D}_\Gamma, V_\Gamma)$ where $V_\Gamma$ is a finite dimensional complex vector space and $\mathcal{D}_\Gamma$ is the homomorphism from $G$ to the unitary operators on $V_\Gamma$. A morphism between two representations $\Gamma_1$ and $\Gamma_2$ is a linear map $\mathcal{I} : V_{\Gamma_1} \to V_{\Gamma_2}$ which intertwines the two representations, i.e., $\mathcal{I} \circ \mathcal{D}_{\Gamma_1}(g) = \mathcal{D}_{\Gamma_2}(g) \circ \mathcal{I}$.

As with any global symmetry, we may also consider the $\mathsf{Rep}(G)$ twisted sectors. The Hilbert space has a direct sum decomposition into $\mathsf{Rep}(G)$ twisted sectors as

$$\mathcal{V} = \bigoplus_\Gamma \mathcal{V}_\Gamma, \qquad \mathcal{V}_\Gamma \cong \mathbb{C}[G]^{\otimes L} \otimes V_\Gamma. \tag{C.4}$$

The $\Gamma$ twisted sector $\mathcal{V}_\Gamma$ is spanned by basis states

$$\tag{C.5}$$

where $v \in V_\Gamma$. The twisted sector states satisfy the property

$$
\text{(C.6)}
$$

using which the action on states can be readily obtained. In general the action of non-invertible symmetries is significantly more complex than their invertible counterparts. A reason for this is that the definition of symmetry operators depends on various choices of branchings and junctions as we will see below. In the simplest case, consider a symmetry operator for $\Gamma \in \mathsf{Rep}(G)$ wrapping the spatial cycle. Its action on an untwisted sector state is evaluated as

$$
\mathcal{U}_\Gamma |\vec{g}, g\rangle = \quad = \sum_i \quad \text{(C.7)}
$$

$$
= \sum_{i,j} \mathcal{D}_\Gamma(g)_{ij} \quad = \chi_\Gamma(g)|\vec{g}, g\rangle \,,
$$

where $\chi_\Gamma(g) = \text{Tr}[\mathcal{D}_\Gamma(g)]$ is the character of $\Gamma$ and $\{v_i^*\}$ is a basis vector in the dual representation space $V_\Gamma^* = \text{Hom}(V_\Gamma, \mathbb{C})$ with a canonical pairing $(v_i, v_j^*) = \delta_{ij}$. In going to the second equality, we use the the map $\mathbb{C} \to V \otimes V^*$, under which $1 \mapsto \sum_i v_i \otimes v_i^*$.

In contrast to invertible symmetries, non-invertible symmetry operators can map between different twisted Hilbert spaces. The simplest example of an operator implementing such a map contains a trivalent junction $\mathcal{I} : \Gamma_2 \to \Gamma_1^\vee \otimes \Gamma_1$[5]

$$
\mathcal{U}_{\Gamma_1}(\Gamma_2; \mathcal{I})|\vec{g}, g\rangle = \quad \text{(C.8)}
$$

This $\mathsf{Rep}(G)$ action can be evaluated using the intertwiner $1 \mapsto \sum_i v_i^* \otimes v_i$ and (C.6) as

$$
\mathcal{U}_{\Gamma_1}(\Gamma_2; \mathcal{I})|\vec{g}\rangle = \sum_{i,j} \mathcal{D}_{\Gamma_1}(g)_{ij} \quad = \sum_{i,j} \mathcal{D}_{\Gamma_1}(g)_{ij}
$$

$$
= \sum_{i,j} \mathcal{D}_{\Gamma_1}(g)_{ij} |\vec{g}, g\rangle_{(\Gamma_2, \mathcal{I}^{-1}(v_i^*, v_j))} \,.
$$

$$
\text{(C.9)}
$$

[5]Here $\Gamma^\vee$ is the dual representation defined via the pair $(\mathcal{D}_\Gamma^*, V_\Gamma^*)$ satisfying $\mathcal{D}_\Gamma^*(g)(v^*) = v^* \circ \mathcal{D}_\Gamma(g^{-1})$.

Similarly twisted sector states may also be mapped into untwisted sector states via $\mathsf{Rep}(G)$ action as

$$\mathcal{U}_{\Gamma_1}(\mathcal{I})|\vec{g},g\rangle_{(\Gamma_2,v)} = \quad \raisebox{-2em}{} \quad = \sum_{i,j}\;\mathcal{D}_{\Gamma_1}(g)_{ij}\;\raisebox{-2em}{} \tag{C.10}$$

$$= \sum_{i,j}\mathcal{D}_{\Gamma_1}(g)_{ij}\left[\mathcal{I}^{-1}(v,v_j)\right]_i|\vec{g},g\rangle$$

where $\mathcal{I}:\Gamma_1\to\Gamma_2\otimes\Gamma_1$ and $v_i\in V_{\Gamma_1}$, $v\in V_{\Gamma_2}$. Next we consider the following $\mathsf{Rep}(G)$ action mapping a twisted sector state to a twisted sector state

$$\mathcal{U}_{\Gamma_1}(\Gamma_3,\Gamma_4;\mathcal{I}_1,\mathcal{I}_2)|\vec{g},g\rangle_{(\Gamma_2,v)} = \quad \raisebox{-3em}{} \tag{C.11}$$

where $\mathcal{I}_2:\Gamma_4\to\Gamma_3\otimes\Gamma_1$, $\mathcal{I}_1:\Gamma_3\to\Gamma_1^\vee\otimes\Gamma_2$ and $v\in V_{\Gamma_2}$. Following similar steps as the previous calculations one finds

$$\mathcal{U}_{\Gamma_1}(\Gamma_3,\Gamma_4;\mathcal{I}_1,\mathcal{I}_2)|\vec{g},g\rangle_{(\Gamma_2,v)} = \sum_{i,j}\mathcal{I}_1^{-1}(v_i^*,v)_j\mathcal{I}_2^{-1}(v_j,v_k)_l\mathcal{D}_{\Gamma_1}(g)_{ik}|\vec{g},v_l\rangle_{\Gamma_4}, \tag{C.12}$$

where $v_i^*\in V_{\Gamma_1}^\vee$, $v_j\in V_{\Gamma_3}$, $v_k\in V_{\Gamma_1}$ and $v_\ell\in V_{\Gamma_4}$.

So far we have described the general structure of how states transform under $\mathsf{Rep}(G)$ action. Now we specialize to the group $\mathsf{Rep}(S_3)$ which has three simple objects

$$1,\ P,\ E, \tag{C.13}$$

where $P$ is the one dimensional sign representation

$$\mathcal{D}_P(a)=1,\qquad \mathcal{D}_P(b)=-1, \tag{C.14}$$

and $E$ is the two dimensional representation such that

$$\mathcal{D}_E(a)=\begin{pmatrix}\omega & 0\\ 0 & \omega^2\end{pmatrix},\qquad \mathcal{D}_E(b)=\begin{pmatrix}0 & 1\\ 1 & 0\end{pmatrix}, \tag{C.15}$$

where $\omega = \exp\{2\pi i/3\}$. We denote the basis vectors spanning $V_E$ as $v_1 \sim (1,0)$ and $v_2 \sim (0,1)$. The vectors generating $V_1$ and $V_P$ are denoted as $v_{\mathrm{id}}$ and $v_P$ respectively. The $\mathsf{Rep}(S_3)$ fusion rules are

$$P \otimes P = 1\,,$$
$$P \otimes E = E \otimes P = E\,, \qquad\qquad\qquad (\text{C.16})$$
$$E \otimes E = 1 \oplus P \oplus E\,.$$

To compute the $\mathsf{Rep}(S_3)$ action, we require the intertwiners between representations. We work with the following choice of intertwiners

$$\mathcal{I}^1_{EE} : V_E \otimes V_E \longrightarrow V_1\,, \quad (v_1\,,v_2) \longmapsto v_{\mathrm{id}}\,, \quad (v_2\,,v_1) \longmapsto v_{\mathrm{id}}\,,$$
$$\mathcal{I}^P_{EE} : V_E \otimes V_E \longrightarrow V_P\,, \quad (v_1\,,v_2) \longmapsto v_P\,, \quad (v_2\,,v_1) \longmapsto -v_P\,, \qquad (\text{C.17})$$
$$\mathcal{I}^E_{EE} : V_E \otimes V_E \longrightarrow V_E\,, \quad (v_1\,,v_1) \longmapsto v_2\,, \quad (v_2\,,v_2) \longmapsto v_1\,.$$

The remaining intertwiners can be obtained by rotation and the identification $v_1^* \cong v_2$ and $v_2^* \cong v_1$. Since all the fusion multiplicities for the fusion of simple objects in $\mathsf{Rep}(S_3)$ are either 0 or 1, the junction intertwiners are uniquely determined by the choice of lines. We therefore drop the junction labels in what follows. First let us consider the action under $\mathcal{U}_E(\Gamma)$ given in (C.9)

$$\mathcal{U}_E(\Gamma)|\vec{g}, g\rangle = \qquad\qquad\qquad\qquad\qquad\qquad (\text{C.18})$$

which has the action

$$\mathcal{U}_E(P)|\vec{g}, g\rangle = [\mathcal{D}_E(g)_{22} - \mathcal{D}_E(g)_{11}]\,|\vec{g}, g\rangle_P\,,$$
$$\mathcal{U}_E(E)|\vec{g}, g\rangle = \mathcal{D}_E(g)_{12}|\vec{g}, g\rangle_{(E,v_1)} + \mathcal{D}_E(g)_{21}|\vec{g}, g\rangle_{(E,v_2)}. \qquad (\text{C.19})$$

It follows from (C.15) that only the states with $g \in [a]$ conjugacy class transform non-trivially between the untwisted and $P$-twisted sectors.

$$|\vec{g}, a\rangle \xrightarrow{\ \mathcal{U}_E(P)\ } (\omega^2 - \omega)|\vec{g}, a\rangle_P\,,$$
$$|\vec{g}, a^2\rangle \xrightarrow{\ \mathcal{U}_E(P)\ } (\omega - \omega^2)|\vec{g}, a^2\rangle_P\,, \qquad\qquad (\text{C.20})$$

Similarly, only the states with $g \in [b]$ conjugacy class transform non-trivially between the untwisted and $E$-twisted sectors.

$$|\vec{g}, b\rangle \xrightarrow{\ \mathcal{U}_E(E)\ } |\vec{g}, b\rangle_{(E,v_1)} + |\vec{g}, b\rangle_{(E,v_2)}\,,$$
$$|\vec{g}, ab\rangle \xrightarrow{\ \mathcal{U}_E(E)\ } \omega|\vec{g}, ab\rangle_{(E,v_1)} + \omega^2|\vec{g}, ab\rangle_{(E,v_2)}\,, \qquad (\text{C.21})$$
$$|\vec{g}, a^2b\rangle \xrightarrow{\ \mathcal{U}_E(E)\ } \omega^2|\vec{g}, a^2b\rangle_{(E,v_1)} + \omega|\vec{g}, a^2b\rangle_{(E,v_2)}\,,$$

Next let us describe the $\mathsf{Rep}(S_3)$ action on the twisted sector states. The $P$ action in the $P$ twisted Hilbert space is simply

$$\mathcal{U}_P(1,P)|\vec{g},g\rangle_P = \chi_P(g)|\vec{g},g\rangle_P\,. \tag{C.22}$$

While $\mathcal{U}_E$ can map a $P$-twisted state to an untwisted sector state. The amplitude of such an action is the $P$-twisted trace in the $E$-representation, i.e.,

$$\mathcal{U}_E|\vec{g},g\rangle_P = [\mathcal{D}_{11}(g) - \mathcal{D}_{22}(g)]\,|\vec{g}\rangle\,. \tag{C.23}$$

Clearly such an action is non-trivial only if $g$ is in the $[a]$ conjugacy class. Lastly, there are $\mathcal{U}_E(E,E)$ and $\mathcal{U}_E(E,P)$ operators as defined in (C.11) (recall that junction labels are suppressed) which act as

$$\begin{aligned}
\mathcal{U}_E(E,E)|\vec{g},g\rangle_P &= \mathcal{D}_E(g)_{12}|\vec{g},g\rangle_{(E,v_1)} - \mathcal{D}_E(g)_{21}|\vec{g},g\rangle_{(E,v_2)}\,, \\
\mathcal{U}_E(E,P)|\vec{g},g\rangle_P &= -\chi_E(g)|\vec{g},g\rangle_P\,.
\end{aligned} \tag{C.24}$$

The action of $\mathcal{U}_E(E,E)$ is non-trivial only when $g$ is in the $[b]$ conjugacy class while that of $\mathcal{U}_E(E,P)$ is non-trivial only in the other two conjugacy classes i.e., in $[1]$ and $[a]$.

Finally the $\mathsf{Rep}(S_3)$ action on the $E$-twisted sector states is

$$\begin{aligned}
\mathcal{U}_P(E,E)|\vec{g},g\rangle_{(E,v)} &= -\chi_P(g)|\vec{g},g\rangle_{(E,v)}\,, \\
\mathcal{U}_E(1,E)|\vec{g},g\rangle_{(E,v_i)} &= \sum_j \mathcal{D}_E(g)_{ij}|\vec{g},g\rangle_{(E,v_j)}\,, \\
\mathcal{U}_E(P,E)|\vec{g},g\rangle_{(E,v_i)} &= \sum_j \mathcal{D}_E(g)_{ij}(-1)^{\delta_{i,j}}|\vec{g},g\rangle_{(E,v_j)}\,, \\
\mathcal{U}_E(E,1)|\vec{g},g\rangle_{(E,v_i)} &= \mathcal{D}_E(g)_{i+1,i}|\vec{g},g\rangle\,, \\
\mathcal{U}_E(E,P)|\vec{g},g\rangle_{(E,v_i)} &= \mathcal{D}_E(g)_{i+1,i}(-1)^i|\vec{g},g\rangle_P\,, \\
\mathcal{U}_E(E,E)|\vec{g},g\rangle_{(E,v_i)} &= \mathcal{D}_E(g)_{i+1,i+1}|\vec{g},g\rangle_{(E,v_i)}\,.
\end{aligned} \tag{C.25}$$

**SymTFT setup.** Before moving on to the description of the $\mathsf{Rep}(S_3)$ multiplets and gapped phases realized in this model, we remind the reader of the SymTFT, which provides a natural construction of this spin model. We have the SymTFT for $\mathsf{Rep}(S_3)$ which is $\mathcal{Z}(\mathsf{Vec}_{S_3}) \cong \mathcal{Z}(\mathsf{Rep}(S_3))$. The topological line defects in this TFT are summarized in the main text in Sec. 2.3. For the present model we choose the input and physical boundary as

$$\begin{aligned}
\mathfrak{B}^{\mathrm{sym}} &= ([\mathrm{id}],1) \oplus ([a],1) \oplus ([b],+)\,, \\
\mathfrak{B}^{\mathrm{inp}} &= ([\mathrm{id}],1) \oplus ([\mathrm{id}],1_-) \oplus 2([\mathrm{id}],E)\,.
\end{aligned} \tag{C.26}$$

The interface between them is provided by the Vec module category. The fusion category of lines on the input and symmetry boundaries are $\mathsf{Vec}_{S_3}$ and $\mathsf{Rep}(S_3)$ respectively. On the input boundary, the bulk lines $([a], 1)$ projects to $a \oplus a^2$ while $([b], +)$ projects to $b \oplus ab \oplus a^2 b$. On the symmetry boundary $([\mathrm{id}], 1_-)$ projects to $P$ and $([\mathrm{id}], E)$ projects to $E$. The projections of the remaining SymTFT can be obtained by consistency requirements. Moreover these lines and their projections play a special role as they are Bosonic and deliver the $\mathsf{Rep}(S_3)$ charges or symmetry multiplets.

$\mathsf{Rep}(S_3)$ **order parameters:** Following the general theory in Sec. 2.1, the possible order parameters for any given gapped phase are in one-to-one correspondence with bosonic lines in the SymTFT for $\mathsf{Rep}(S_3)$ which is $\mathcal{Z}(\mathsf{Vec}_{S_3}) \cong \mathcal{Z}(\mathsf{Rep}(S_3))$ which are.

$$([\mathrm{id}], 1), \quad ([\mathrm{id}], \mathrm{P}), \quad ([\mathrm{id}], \mathrm{E}),$$
$$([a], 1), \quad ([a], \omega), \quad ([a], \omega^2), \tag{C.27}$$
$$([b], +), \quad ([b], -),$$

Among these, the bosonic lines are

$$([\mathrm{id}], 1), \quad ([\mathrm{id}], \mathrm{P}), \quad ([\mathrm{id}], \mathrm{E}), \quad ([a], 1), \quad ([b], +). \tag{C.28}$$

Corresponding to each of these lines, one obtains a mulitplet of operators that transform irreducibly under the action of $\mathsf{Rep}(S_3)$. The identity line $([\mathrm{id}], 1)$ corresponds to the identity operator while the charge line carrying the 1-dimensional representation $P$ is a symmetry twist/string operator that acts on states as

$$\mathcal{O}_{P, j+\frac{1}{2}} : |\vec{g}, g\rangle_{(\Gamma, v)} \longrightarrow |\vec{g}, g\rangle_{P \otimes (\Gamma, v)}, \tag{C.29}$$

where $v \in V_\Gamma$ and for simplicity, we assume that the $(\Gamma, v)$ twist line in the state $|\vec{g}, g\rangle_{(\Gamma, v)}$ is located at the site $j$. More general cases can be treated similarly, however one needs to account for how the states transform when the twist lines are transported.

The line carrying the $E$-representation gives rise to a doublet of string operators that can be labelled by basis vectors $v_1, v_2$ spanning $V_E$. Their action on states is similarly given by

$$\mathcal{O}_{(E, v_i), +\frac{1}{2}} : |\vec{g}, g\rangle_{(\Gamma, v)} \longrightarrow |\vec{g}, g\rangle_{(E, v_i) \otimes (\Gamma, v)}. \tag{C.30}$$

The SymTFT line $([a], 1)$ has quantum dimension 2 and corresponds to a doublet of operators,

$$\mathcal{O}_{([a], 1), j+\frac{1}{2}} = \left( \mathcal{O}^+_{a, j+\frac{1}{2}}, \mathcal{O}^-_{a, j+\frac{1}{2}} \right), \tag{C.31}$$

one of which is a local operator, i.e., it acts within a given symmetry twisted sector of the Hilbert space while the other is non-local or a string operator that maps between different twisted sectors. We emphasize that this feature of symmetry multiplets comprising of a combination of local and string operators is a feature is unique to non-invertible symmetries. The action of the $\mathcal{O}_{([a],1),j}$ multiplet on states is

$$
\begin{aligned}
\mathcal{O}^+_{a,j+\frac{1}{2}} &= L^a_{j+\frac{1}{2}} + L^{a^2}_{j+\frac{1}{2}} , \\
\mathcal{O}^+_{a,j} &= \left[ L^a_{j+\frac{1}{2}} - L^{a^2}_{j+\frac{1}{2}} \right] \mathcal{O}_{P,j+\frac{1}{2}} ,
\end{aligned}
\tag{C.32}
$$

where $L^h_{j+\frac{1}{2}}$ is a local operator that implements left multiplication by the group element on the degree of freedom at $j + \frac{1}{2}$

$$
L^h_{j+\frac{1}{2}} |g_{j+\frac{1}{2}}\rangle = |hg_{j+\frac{1}{2}}\rangle .
\tag{C.33}
$$

Lastly the SymTFT line $([b],+)$ has quantum dimension 3 and corresponds to a multiplet of three operators, two of which are twisted sector operators

$$
\mathcal{O}_{([b],+),j+\frac{1}{2}} = \left( \mathcal{O}_{b,j+\frac{1}{2}} , \mathcal{O}^1_{b,j+\frac{1}{2}} , \mathcal{O}^2_{b,j+\frac{1}{2}} \right) ,
\tag{C.34}
$$

defined as

$$
\begin{aligned}
\mathcal{O}_{b,j+\frac{1}{2}} &= L^b_{j+\frac{1}{2}} + L^{ab}_{j+\frac{1}{2}} + L^{a^2 b}_{j+\frac{1}{2}} , \\
\mathcal{O}^1_{b,j+\frac{1}{2}} &= \left[ L^b_{j+\frac{1}{2}} + \omega L^{ab}_{j+\frac{1}{2}} + \omega^2 L^{a^2 b}_{j+\frac{1}{2}} \right] \mathcal{O}_{(E,v_1),j+\frac{1}{2}} , \\
\mathcal{O}^1_{b,j+\frac{1}{2}} &= \left[ L^b_{j+\frac{1}{2}} + \omega^2 L^{ab}_{j+\frac{1}{2}} + \omega L^{a^2 b}_{j+\frac{1}{2}} \right] \mathcal{O}_{(E,v_2),j+\frac{1}{2}} .
\end{aligned}
\tag{C.35}
$$

## Rep($S_3$) gapped phases

In this section we describe the structure of gapped phases realized in Rep($S_3$) symmetric systems. There are four gapped phases whose fixed-point Hamiltonians are obtained by picking a Frobenius algebra in Vec$_{S_3}$. Recall that a Frobenius algebra in Vec$_{S_3}$ is labelled by a subgroup $H$ of $S_3$. Correspondingly the fixed-point Hamiltonian in the Rep($S_3$) anyon model is

$$
\mathcal{H}^{(\mathsf{Rep}(S_3))}_H = -\frac{1}{|H|} \sum_j \left[ \begin{array}{c} \mathcal{A}_H \quad \mathcal{A}_H \\ \mathcal{A}_H \\ \mathcal{A}_H \quad \mathcal{A}_H \end{array} \right]_j ,
\tag{C.36}
$$

which can be expressed in terms of local $S_3$ spin operators acting as

$$
\mathcal{H}^{(\mathsf{Rep}(S_3))}_H = -\frac{1}{|H|} \sum_j \sum_{h \in H} \Pi^{(H)}_{j-\frac{1}{2}} \Pi^{(H)}_{j+\frac{1}{2}} L^h_{j-\frac{1}{2}} L^{h^{-1}}_{j+\frac{1}{2}} .
\tag{C.37}
$$

Here $\Pi^H_{j+\frac{1}{2}}$ is a projection operator at the edge $j+1/2$ that projects on the subspace of $\mathcal{V}_{j+1/2}$ spanned by $|h\rangle$ for $h \in H$. Concretely, these projectors are

$$\Pi^{(S_3)}_{j+\frac{1}{2}} = \mathbb{I}_{j+\frac{1}{2}}$$

$$\Pi^{(\mathbb{Z}_3)}_{j+\frac{1}{2}} = \frac{1}{2} \sum_{\Gamma=1,P} Z^\Gamma_{j+\frac{1}{2}}$$

$$\Pi^{(\mathbb{Z}_2)}_{j+\frac{1}{2}} = \frac{1}{3} \left[ \mathbb{I}_{j+\frac{1}{2}} + \sum_{IJ} \left( Z^E_{j+\frac{1}{2}} \right)_{IJ} \right] \tag{C.38}$$

$$\Pi^{(1)}_{j+\frac{1}{2}} = \sum_\Gamma \frac{\dim(\Gamma)}{|G|} \mathrm{Tr}\left( Z^\Gamma_{j+\frac{1}{2}} \right) .$$

expressed in terms of the local operators (B.6) acting on the on-site Hilbert space associated to half-integers. A more economic fixed-point Hamiltonian, i.e., one that involves interactions between fewer degrees of freedom while being in the same gapped phase is given by

$$\widetilde{\mathcal{H}}^{(\mathsf{Rep}(S_3))}_H = -\frac{1}{|H|} \sum_j \sum_{h \in H} L^h_{j-\frac{1}{2}} L^{h^{-1}}_{j+\frac{1}{2}} - \sum_j \Pi^{(H)}_{j+\frac{1}{2}} . \tag{C.39}$$

Since these Hamiltonians are all $\mathsf{Rep}(S_3)$ symmetric, their action on the twisted Hilbert space, i.e., in the presence of $\mathsf{Rep}(S_3)$ defects can be considered. The presence of $\mathsf{Rep}(S_3)$ defect leaves the projectors $\Pi^{(H)}_{j+\frac{1}{2}}$ unaltered, while the other operators act as

$$L^h_{j-\frac{1}{2}} L^{h^{-1}}_{j+\frac{1}{2}} |g_{j-\frac{1}{2}}, (\Gamma, v)_j, g_{j+\frac{1}{2}}\rangle = |g_{j-\frac{1}{2}} h, (\Gamma, \mathcal{D}_\Gamma(h) \cdot v)_j, h^{-1} g_{j+\frac{1}{2}}\rangle . \tag{C.40}$$

Therefore we can define a Hamiltonian in the phase labelled by $\mathcal{A}_H$ acting on the twisted Hilbert space $\mathcal{V}_\Gamma$ with the twist defect on site $j_0$ as

$$\widetilde{\mathcal{H}}^{(\mathsf{Rep}(S_3))}_{H,\Gamma} = -\frac{1}{|H|} \sum_{j \neq j_0} \sum_{h \in H} L^h_{j-\frac{1}{2}} L^{h^{-1}}_{j+\frac{1}{2}} - \frac{1}{|H|} \sum_{j_0} \sum_{h \in H} L^h_{j_0-\frac{1}{2}} \mathcal{D}_\Gamma(h)_{j_0} L^{h^{-1}}_{j_0+\frac{1}{2}} - \sum_j \Pi^{(H)}_{j+\frac{1}{2}} , \tag{C.41}$$

where the operator $\mathcal{D}_\Gamma(h)_{j_0}$ acts on the vector space $V_\Gamma$ inserted at $j_0$. We now specialize to different choices of $\mathcal{A}_H$ and describe the characterization of the corresponding gapped phases by the $\mathsf{Rep}(S_3)$ action on their ground state multiplets and the existence of order parameters.

$\mathsf{Rep}(S_3)$ **Trivial phase:** The trivial phase is one with a single $\mathsf{Rep}(S_3)$ invariant ground state in the untwisted Hilbert space. This phase corresponds to the choice of algebra with a single (identity) object

$$\mathcal{A}_1 = 1 . \tag{C.42}$$

Choosing this algebra, we expect the gapped phase for which the physical boundary is the same as the input boundary of the SymTFT

$$\mathfrak{B}^{\mathrm{inp}} = \mathfrak{B}^{\mathrm{phys}} = ([\mathrm{id}], 1) \oplus ([\mathrm{id}], 1_-) \oplus 2([\mathrm{id}], E) . \tag{C.43}$$

Using (C.39), the Hamiltonian has the form

$$\widetilde{\mathcal{H}}_1^{(\mathsf{Rep}(S_3))} = -\sum_j \Pi_{j+\frac{1}{2}}^{(1)} = -\sum_j \sum_{\Gamma=1,P,E} \frac{\dim(\Gamma)}{|S_3|} \mathrm{Tr}\left[Z_{j+1/2}^{\Gamma}\right]. \tag{C.44}$$

The form of the Hamiltonian is insensitive to the presence of $\mathsf{Rep}(S_3)$ defects. Consequently, there is a single ground state in each $\mathsf{Rep}(S_3)$ twisted sector. These ground states are

$$|\mathrm{GS}\rangle_1 = |\vec{1},1\rangle_1, \quad |\mathrm{GS}\rangle_P = |\vec{1},1\rangle_P, \quad |\mathrm{GS}\rangle_{E,v_i} = |\vec{1},1\rangle_{(E,v_i)}, \tag{C.45}$$

where $\vec{1} = (1,1,\ldots,1)$ and $i = 1,2$. The $\mathsf{Rep}(S_3)$ action on this multiplet can be straightforwardly computed using the procedure described in Sec. 2.3. On the untwisted and $P$-twisted sector it takes the form

$$\mathcal{U}_\Gamma|\mathrm{GS}\rangle_1 = \dim(\Gamma)|\mathrm{GS}\rangle_1,$$
$$\mathcal{U}_P|\mathrm{GS}\rangle_P = |\mathrm{GS}\rangle_P, \tag{C.46}$$
$$\mathcal{U}_E|\mathrm{GS}\rangle_P = -2|\mathrm{GS}\rangle_P,$$

while on the $E$-twisted sector ground states

$$\mathcal{U}_P(E,E)|\mathrm{GS}\rangle_{(E,v_i)} = -|\mathrm{GS}\rangle_{(E,v_i)},$$
$$\mathcal{U}_E(P,E)|\mathrm{GS}\rangle_{(E,v_i)} = -|\mathrm{GS}\rangle_{(E,v_i)}, \tag{C.47}$$
$$\mathcal{U}_E(X,E)|\mathrm{GS}\rangle_{(E,v_i)} = +|\mathrm{GS}\rangle_{(E,v_i)}, \quad X = 1, E.$$

Note that no two distinct twisted sector ground states map into each other under $\mathsf{Rep}(S_3)$ action. The order parameters for this gapped phase are $\mathcal{O}_P$ and $\mathcal{O}_{(E,v_i)}$ which correspond to the SymTFT lines $([\mathrm{id}],1_-)$ and $([\mathrm{id}],E)$. This is compatible with the fact that the Lagrangian algebra defining the topological boundary condition for the $\mathsf{Rep}(S_3)$ trivial phase is

$$\mathcal{L} = ([\mathrm{id}],1) \oplus ([\mathrm{id}],1_-) \oplus 2([\mathrm{id}],E). \tag{C.48}$$

The action of these order parameters on the multiplet of ground states is

$$\mathcal{O}_P|\mathrm{GS}\rangle_{(\Gamma,v)} = |\mathrm{GS}\rangle_{P\otimes(\Gamma,v)},$$
$$\mathcal{O}_{(E,v_i)}|\mathrm{GS}\rangle_{(\Gamma,v)} = |\mathrm{GS}\rangle_{(E,v_i)\otimes(\Gamma,v)}. \tag{C.49}$$

$\mathbb{Z}_2$ **SSB phase:** Next, we consider the gapped phase corresponding to the Frobenius algebra

$$\mathcal{A}_{\mathbb{Z}_2} = 1 \oplus b. \tag{C.50}$$

Within the SymTFT such a gapped phase corresponds to gauging $\mathbb{Z}_2^b$ on the input boundary thus delivering

$$\mathfrak{B}^{\mathrm{phys}} = ([\mathrm{id}],1) \oplus ([b],+) \oplus ([\mathrm{id}],E). \tag{C.51}$$

A fixed-point Hamiltonian in the untwisted sector in this gapped phase is obtained by setting $H = \mathbb{Z}_2^b$ in (C.39)

$$\widetilde{\mathcal{H}}_{\mathbb{Z}_2}^{(\mathsf{Rep}(S_3))} = -\frac{1}{2}\sum_j \left[\mathbb{I} + L_{j-\frac{1}{2}}^b L_{j+\frac{1}{2}}^b\right] - \sum_j \frac{1}{3}\left[\mathbb{I} + \sum_{IJ}\left(Z_{j+\frac{1}{2}}^E\right)_{IJ}\right]. \tag{C.52}$$

The second term in the Hamiltonian restricts to the subspace with degrees of freedom restriced to $|1\rangle, |b\rangle \in \mathcal{V}_{j+\frac{1}{2}}$, while the first term combines terms with the same holonomy (i.e., either 1 or $b$) into a single orbit under the Hamiltonian action. Therefore one finds a two-dimensional ground state space spanned by

$$|\Psi_1\rangle \propto \prod_j \sum_{g_{j+\frac{1}{2}}=1,b} \delta_{g,1}|\vec{g},g\rangle, \qquad |\Psi_b\rangle \propto \prod_j \sum_{g_{j+\frac{1}{2}}=1,b} \delta_{g,b}|\vec{g},g\rangle. \tag{C.53}$$

However these are not the thermodynamic ground states/vacua of the theory. The ground states are given by linear combinations of these states

$$|\Psi_\pm\rangle = \frac{1}{2^{L/2}}\prod_j \sum_{g_j=1,b}(-1)^{\pi(g)}|\vec{g},g\rangle, \tag{C.54}$$

where $\pi : S_3 \to \mathbb{Z}_2$ such that $\pi(1) = 0$ and $\pi(b) = 1$. Next, we consider the $P$-twisted sector. The Hamiltonian with a $P$ symmetry twist at site $j_0$ is (see (C.41))

$$\widetilde{\mathcal{H}}_{\mathbb{Z}_2,P}^{(\mathsf{Rep}(S_3))} = -\frac{1}{2}\sum_{j \neq j_0}\left[\mathbb{I} + L_{j-\frac{1}{2}}^b L_{j+\frac{1}{2}}^b\right] - \frac{1}{2}\left[\mathbb{I} - L_{j_0-\frac{1}{2}}^b L_{j_0+\frac{1}{2}}^b\right]$$
$$- \sum_j \frac{1}{3}\left[\mathbb{I} + \sum_{IJ}\left(Z_{j+\frac{1}{2}}^E\right)_{IJ}\right] \tag{C.55}$$

The orbit of any state $|\vec{g},g\rangle$ with $g_{j+1/2} = 1, b$ under Hamiltonian action vanishes. Therefore there are no $P$ twisted states in the ground state space of the model. Equivalently, the lowest energy eigenstates in the $P$-twisted sector are higher up in energy as compared with the untwisted sector ground states and therefore do not participate in the infra red physics.

Now let us consider the $E$-twisted sector for which the Hamiltonian is

$$\widetilde{\mathcal{H}}_{\mathbb{Z}_2,E}^{(\mathsf{Rep}(S_3))} = -\frac{1}{2}\sum_{j \neq j_0}\left[\mathbb{I} + L_{j-\frac{1}{2}}^b L_{j+\frac{1}{2}}^b\right] - \frac{1}{2}\left[\mathbb{I} + L_{j_0-\frac{1}{2}}^b \mathcal{D}_E(b) L_{j_0+\frac{1}{2}}^b\right]$$
$$- \sum_j \frac{1}{3}\left[\mathbb{I} + \sum_{IJ}\left(Z_{j+\frac{1}{2}}^E\right)_{IJ}\right]. \tag{C.56}$$

There are two $E$-twisted sector ground states

$$|\Psi_1\rangle_E = \frac{1}{2^{L/2}}\prod_j \sum_{g_{j+\frac{1}{2}}=1,b}\delta_{g,1}\left[|\vec{g},g\rangle_{(E,v_1)} + |\vec{g},g\rangle_{(E,v_2)}\right],$$
$$|\Psi_b\rangle_E = \frac{1}{2^{L/2}}\prod_j \sum_{g_{j+\frac{1}{2}}=1,b}\delta_{g,b}\left[|\vec{g},g\rangle_{(E,v_1)} + |\vec{g},g\rangle_{(E,v_2)}\right]. \tag{C.57}$$

Let us now describe the $\mathsf{Rep}(S_3)$ action on the four ground states

$$\{|\Psi_+\rangle, |\Psi_-\rangle, |\Psi_1\rangle_E, \ |\Psi_b\rangle_E\}\,. \tag{C.58}$$

On the untwisted states, the $\mathsf{Rep}(S_3)$ symmetry lines act

$$\begin{aligned}
\mathcal{U}_P|\Psi_1\rangle &= |\Psi_1\rangle\,, \\
\mathcal{U}_P|\Psi_b\rangle &= -|\Psi_b\rangle\,, \\
\mathcal{U}_E|\Psi_1\rangle &= 2|\Psi_1\rangle\,, \\
\mathcal{U}_E|\Psi_b\rangle &= 0\,.
\end{aligned} \tag{C.59}$$

Hence,

$$\mathcal{U}_P|\Psi_\pm\rangle = |\Psi_\mp\rangle\,, \qquad \mathcal{U}_E|\Psi_\pm\rangle = |\Psi_+\rangle + |\Psi_+\rangle\,, \tag{C.60}$$

which satisfy the $\mathsf{Rep}(S_3)$ fusion rules. Since the $\mathcal{U}_P$ symmetry operator which generates the $\mathbb{Z}_2 \in \mathsf{Rep}(S_3)$ exchanges the two ground states, we refer to this phase as the $\mathbb{Z}_2$ SSB phase.

Next, we consider the $\mathsf{Rep}(S_3)$ action that maps between the twisted and untwisted sector ground states. Using (C.19) and (C.25),

$$\begin{aligned}
\mathcal{U}_E(E)|\Psi_\pm\rangle &= \pm|\Psi_b\rangle_E\,, \\
\mathcal{U}_E(E,1)|\Psi_1\rangle_E &= 0\,, \\
\mathcal{U}_E(E,1)|\Psi_b\rangle_E &= |\Psi_+\rangle - |\Psi_-\rangle\,.
\end{aligned} \tag{C.61}$$

$\mathsf{Rep}(S_3)/\mathbb{Z}_2$ **SSB phase:** Next, we consider the gapped phase corresponding to the Frobenius algebra

$$\mathcal{A}_{\mathbb{Z}_3} = 1 \oplus a \oplus a^2\,. \tag{C.62}$$

Within the SymTFT such a gapped phase corresponds to gauging $\mathbb{Z}_3$ on the input boundary thus delivering

$$\mathfrak{B}^{\text{phys}} = ([\text{id}], 1) \oplus ([\text{id}], 1_-) \oplus 2([a], 1)\,. \tag{C.63}$$

A fixed-point Hamiltonian in the untwisted sector in this gapped phase is obtained by setting $H = \mathbb{Z}_3$ in (C.39)

$$\widetilde{\mathcal{H}}_{\mathbb{Z}_3}^{(\mathsf{Rep}(S_3))} = -\frac{1}{3}\sum_j \left[\mathbb{I} + L^a_{j-\frac{1}{2}}L^{a^2}_{j+\frac{1}{2}} + L^{a^2}_{j-\frac{1}{2}}L^a_{j+\frac{1}{2}}\right] - \frac{1}{2}\sum_j \sum_{\Gamma=1,P} Z^{\Gamma}_{j+\frac{1}{2}}\,. \tag{C.64}$$

The second term in the Hamiltonian constrains each degree of freedom such that $g_{j+\frac{1}{2}} \in \{1, a, a^2\}$. The first term disorders these degrees of freedom within a definite $g$ sector, i.e., the action of the Hamiltonian does not alter the holonomy $\prod_j g_{j+\frac{1}{2}}$. We thus find a three

dimensional untwisted sector ground state space spanned by states that have holonomies $1$, $a$ and $a^2$ respectively.

$$|\Psi, q\rangle = \frac{1}{3^{(L-1/2)}} \prod_j \sum_{g_j=1,a,a^2} \delta_{g,a^q} |\vec{g}, g\rangle. \tag{C.65}$$

These are not the thermodynamic ground states which are obtained as a linear combinations

$$|\text{GS}, p\rangle = \frac{1}{\sqrt{3}} \sum_{q=0,1,2} \omega^{pq} |\Psi, q\rangle. \tag{C.66}$$

Using (C.7), it follows that $\text{Rep}(S_3)$ acts on this multiplet of ground states as

$$\begin{aligned} \mathcal{U}_P |\text{GS}, p\rangle &= |\text{GS}, p\rangle, \\ \mathcal{U}_P |\text{GS}, p\rangle &= |\text{GS}, p+1 \bmod 3\rangle + |\text{GS}, p+2 \bmod 3\rangle. \end{aligned} \tag{C.67}$$

This is the $\text{Rep}(S_3)/\mathbb{Z}_2$ SSB phase as the $P$ symmetry acts identically in each ground state and is therefore preserved while the $E$ symmetry maps between ground states and is spontaneously broken.

Notice that the presence of a $P$-twist leaves that Hamiltonian invariant. Therefore there are three isomorphic $P$-twisted sector ground states

$$|\text{GS}, p\rangle_P, \quad p = 0, 1, 2. \tag{C.68}$$

Next, we can see that there in no ground state in the $E$-twisted sector. To see this, let us consider an $E$ twist with vector $v_1 \in V_E$ at the site $j_0$. Now if there was was an $E$-twisted sector ground state, it would need to be in the $+1$ eigenspace of the following operators

$$\begin{aligned} &\frac{1}{3}\left[\mathbb{I} + L_{j-\frac{1}{2}}^a L_{j+\frac{1}{2}}^{a^2} + L_{j-\frac{1}{2}}^{a^2} L_{j+\frac{1}{2}}^a\right], \quad j \neq j_0, \\ &\frac{1}{3}\left[\mathbb{I} + \omega L_{j_0-\frac{1}{2}}^a L_{j_0+\frac{1}{2}}^{a^2} + \omega^2 L_{j_0-\frac{1}{2}}^{a^2} L_{j_0+\frac{1}{2}}^a\right]. \end{aligned} \tag{C.69}$$

No such state exists and therefore there are no $E$-twisted states in the IR. This is consistent with the fact that the charges condensed on the physical boundary in this phase do not contain any $E$-twisted sector operators in their multiplets.

$\text{Rep}(S_3)/\mathbb{Z}_2$ **SSB phase.** Next, we consider the gapped phase corresponding to the Frobenius algebra

$$\mathcal{A}_{S_3} = \bigoplus_{g \in S_3} g. \tag{C.70}$$

Within the SymTFT such a gapped phase corresponds to gauging the full $S_3$ symmetry on the input boundary thus delivering

$$\mathfrak{B}^{\text{phys}} = ([\text{id}], 1) \oplus ([a], 1) \oplus (b, +). \tag{C.71}$$

A fixed-point Hamiltonian in the untwisted sector in this gapped phase is obtained by setting $H = S_3$ in (C.39)

$$\mathcal{H}_{\mathbb{Z}_2}^{(\mathsf{Rep}(S_3))} = -\frac{1}{6} \sum_j \sum_{g \in S_3} L^g_{j-\frac{1}{2}} L^{g^{-1}}_{j+\frac{1}{2}} .$$ 

(C.72)

The ground state space is three dimensional spanned by states with holonomies in the different $S_3$ conjugacy classes. These states are

$$
\begin{aligned}
|\Psi_1\rangle &= \frac{1}{\sqrt{6^{L-1}}} \sum_{\vec{g}} \delta_{g,1} |\vec{g}, g\rangle , \\
|\Psi_a\rangle &= \frac{1}{\sqrt{2.6^{L-1}}} \sum_{\vec{g}} \delta_{g,[a]} |\vec{g}, g\rangle , \\
|\Psi_b\rangle &= \frac{1}{\sqrt{3.6^{L-1}}} \sum_{\vec{g}} \delta_{g,[b]} |\vec{g}, g\rangle .
\end{aligned}
$$ 

(C.73)

The thermodynamic can be found to be the following linear combinations

$$
\begin{aligned}
|\mathrm{GS}, 1\rangle &= \frac{1}{\sqrt{6}} \left[ |\Psi_1\rangle + \sqrt{2}|\Psi_a\rangle + \sqrt{3}|\Psi_b\rangle \right] , \\
|\mathrm{GS}, 2\rangle &= \frac{1}{\sqrt{6}} \left[ |\Psi_1\rangle + \sqrt{2}|\Psi_a\rangle - \sqrt{3}|\Psi_b\rangle \right] , \\
|\mathrm{GS}, 3\rangle &= \frac{1}{\sqrt{6}} \left[ 2|\Psi_1\rangle - \sqrt{2}|\Psi_a\rangle \right] .
\end{aligned}
$$ 

(C.74)

Under $P \in \mathsf{Rep}(S_3)$ symmetry these ground states transform as

$$U_P|\mathrm{GS}, 1\rangle_1 = |\mathrm{GS}, 2\rangle_1 , \quad U_P|\mathrm{GS}, 2\rangle_1 = |\mathrm{GS}, 1\rangle_1 , \quad U_P|\mathrm{GS}, 3\rangle_1 = |\mathrm{GS}, 3\rangle_1 ,$$ 

(C.75)

and under $E \in \mathsf{Rep}(S_3)$,

$$
\begin{aligned}
\mathcal{U}_E|\mathrm{GS}, 1\rangle_1 &= \mathcal{U}_E|\mathrm{GS}, 2\rangle_1 = |\mathrm{GS}, 3\rangle_1 , \\
\mathcal{U}_E|\mathrm{GS}, 3\rangle_1 &= |\mathrm{GS}, 1\rangle_1 + |\mathrm{GS}, 2\rangle_1 + |\mathrm{GS}, 3\rangle_1 .
\end{aligned}
$$ 

(C.76)

The twisted sector ground states as well the $\mathsf{Rep}(S_3)$ action within and between the twisted and untwisted sector states can be computed using the methods described in this section.

$\mathsf{Rep}(S_3)$ **Trivial to $\mathbb{Z}_2$ SSB Phase Transition.** We now consider the lattice construction of the gSPT phase for $\mathsf{Rep}(S_3)$ corresponding to the transition from the $\mathsf{Rep}(S_3)$ trivial phase to the $\mathbb{Z}_2$ SSB phase. In the club sandwich set-up, this is realized by starting from $\mathfrak{Z}(S_3)$ and condensing the non-Lagrangian algebra $\mathcal{A}_{\mathrm{phys}} = ([\mathrm{id}], 1) \oplus ([\mathrm{id}], E)$. This produces an interface

to the reduced topological order $\mathfrak{Z} = \mathfrak{Z}(\mathbb{Z}_2)$ given by the Toric Code.

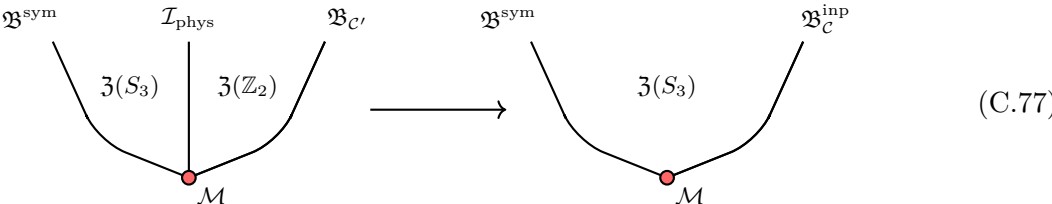

$$(C.77)$$

The interface provides a map from the topological lines of the toric code topological order to those of $\mathfrak{Z}(S_3)$.

$$
\begin{aligned}
1 &\longmapsto ([\mathrm{id}], 1) \oplus ([\mathrm{id}], E)\,, & m &\longmapsto ([b], +)\,, \\
e &\longmapsto ([\mathrm{id}], 1_-) \oplus ([\mathrm{id}], E)\,, & f &\longmapsto ([b], -)\,.
\end{aligned}
\tag{C.78}
$$

We pick the topological boundary for $\mathfrak{Z}(\mathbb{Z}_2)$ to be

$$
\mathfrak{B}_{\mathcal{C}'} = 1 \oplus e\,,
\tag{C.79}
$$

on which the symmetry is $\mathcal{C}' = \mathbb{Z}_2 = \{1, P\}$. The $P$ line on $\mathfrak{B}_{\mathcal{C}'}$ is obtained by the projection of the bulk line $m$, while the bulk $e$ line projects to the identity in $\mathcal{C}'$. Now we may consider a lattice system constructed on the input boundary $\mathfrak{B}_{\mathcal{C}'}$ with input $\rho' = 1 \oplus P$. Specifically, let us consider a Hamiltonian that realizes the $\mathbb{Z}_2$ symmetric transverse field Ising model described in (4.25). This model realizes a $\mathbb{Z}_2$ symmetric trivial phase (Triv), a $\mathbb{Z}_2$ SSB phase and a $\mathbb{Z}_2$ transition between the two phases in the Ising universality class at $\lambda = 1$.

Upon compactifying the interval containing $\mathfrak{Z}(\mathbb{Z}_2)$ between $\mathcal{I}_{\mathrm{phys}}$ and $\mathfrak{B}_{\mathcal{C}'}$ as depicted in (C.77), we obtain the following topological boundary condition $\mathfrak{B}_{\mathcal{C}}^{\mathrm{inp}}$ of the SymTFT $\mathfrak{Z}(S_3)$ using (C.78)

$$
\mathfrak{B}_{\mathcal{C}}^{\mathrm{inp}} = ([\mathrm{id}], 1) \oplus ([\mathrm{id}], 1_-) \oplus 2([\mathrm{id}], E)\,.
\tag{C.80}
$$

This is the input boundary for the $\mathsf{Rep}(S_3)$ symmetric anyon model with $\mathcal{M} = \mathsf{Vec}$. Under the compactification of $\mathfrak{Z}(\mathbb{Z}_2)$, the input of the $\mathbb{Z}_2$ model maps as

$$
1 \oplus P \longmapsto 1 + b\,.
\tag{C.81}
$$

Therefore to summarize, the club sandwich after compactifying $\mathfrak{Z}(\mathbb{Z}_2)$ produces an anyon model with input

$$
\mathcal{C} = \mathsf{Vec}_{S_3}\quad,\quad \mathcal{M} = \mathsf{Vec}\quad,\quad \rho = 1 \oplus b\,.
\tag{C.82}
$$

The Hamiltonian (4.25) in terms of the spin operators in $L^g_{j+\frac{1}{2}}$ and $\Pi^{(H)}_{j+\frac{1}{2}}$ for this choice of input data takes the form

$$
\mathcal{H}_{\mathrm{TFI}}^{(\mathsf{Rep}(S^3))} = -\sum_j \left[ \Pi^{(1)}_{j+\frac{1}{2}} + \frac{\lambda}{2} \Pi^{(\mathbb{Z}_2)}_{j-\frac{1}{2}} \left( 1 + L^b_{j-\frac{1}{2}} L^b_{j+\frac{1}{2}} \right) \Pi^{(\mathbb{Z}_2)}_{j+\frac{1}{2}} \right]\,.
\tag{C.83}
$$

The first term is a projector onto the $g_{j+\frac{1}{2}} = 1$, while $\Pi^{(\mathbb{Z}_2)}_{j+\frac{1}{2}}$ is a projector onto $g_{j+\frac{1}{2}} = 1, b$. Therefore the low energy physics of this model lies in the subspace of states with $g_{j+\frac{1}{2}} = 1, b$ for all $j$. We define effective Pauli spin operators on this space such the $g_{j+\frac{1}{2}} = 1, b$ are $\sigma^z$ eigenstates with eigenvalues $+1$ and $-1$ respectively. In terms of the Pauli operators,

$$\Pi^{(1)}_{j+\frac{1}{2}} \simeq \frac{1 + \sigma^z_{j+\frac{1}{2}}}{2}, \quad \Pi^{(\mathbb{Z}_2)}_{j+\frac{1}{2}} \simeq \mathbb{I}, \quad L^b_{j+\frac{1}{2}} \simeq \sigma^x_{j+\frac{1}{2}}. \tag{C.84}$$

Therefore the Hamiltonian (C.83) simplifies to

$$\mathcal{H}^{(\mathsf{Rep}(S_3))}_{\mathrm{TFI-eff}} = -\frac{1}{2} \sum_j \left[ (1 + \sigma^z_{j+\frac{1}{2}}) + \lambda(1 + \sigma^x_{j-\frac{1}{2}} \sigma^x_{j+\frac{1}{2}}) \right]. \tag{C.85}$$

Let us describe the $\mathsf{Rep}(S_3)$ action on this model. $\mathcal{U}_P$ is realized in a standard spin parity measuring operator which is the $\mathbb{Z}_2$ symmetry of the Transverse field Ising model,

$$\mathcal{U}_P = \prod_j \sigma^z_{j+\frac{1}{2}}. \tag{C.86}$$

On the other hand recall that the $E$ symmetry acts as the character on a given basis state, i.e.

$$\mathcal{U}_E |\vec{g}, g\rangle = \chi_E(g) |\vec{g}, g\rangle, \tag{C.87}$$

on the whole space. In the restricted low energy space of the present model, $g \in \{1, b\}$, therefore it follows that

$$\mathcal{U}_E = 1 + \mathcal{U}_P = 1 + \prod_j \sigma^z_{j+\frac{1}{2}}. \tag{C.88}$$

The phase diagram parametrized by $\lambda$ has the form

$$\tag{C.89}$$

The order parameter for this phase transition is $\sigma^x_{j+\frac{1}{2}}$ which has a vanishing expectation value in the trivial and non-vanishing expectation value in the SSB phase. This order parameter is charged under $\mathcal{U}_P$ and becomes the spin operator at the Ising transition.

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
