# Peer review of "Lattice Models for Phases and Transitions with Non-Invertible Symmetries"

_SciPost Physics_

## Round 5 · Author Response

We thank the referees for their positive assessment of our work and for recommending publication. We also appreciate their helpful suggestions. Below we have provided a list of changes made in response to their comments and questions.

---

## Round 5 · List of Changes

Response to Referee 1:

We thank the referee for their positive assessment of our work and for recommending publication. We also appreciate their helpful suggestion. Following their advice, we have added a separate paragraph summarizing the lattice models we constructed, in the introduction.

Response to Referee 2:

We thank the referee for their positive evaluation of our work and for recommending publication. Below we respond to their comments and questions.

  1. Thank you for pointing this out. We have clarified this by explicitly stating that the input datum refers to boundaries of the SymTFT.

  2. We have added a footnote stating "The generalized charges are defined as the irreducible representations of the Tube algebra $Tube(S)$ of the symmetry category $S$" where generalized charges first mentioned.

  3. It is not true that the lines in Z(S) need to be Bosonic. The Bosonic lines however have the property that their coprresponding generalized charges can condense, i.e., serve as order parameters for a certain phase. Mathematically these lines can appear in condensable algebras which describe topological interfaces (and boundaries) of the SymTFT.

  4. We in fact reworded this. It was brought to our attention by Ananda Roy, that the anyon chains are numerically very well analyzable. It would be interesting to explore this further, and we believe this is being done in Roy's group.

  5. Eq 2.4 is the coordinatized expression of the fact that any (2-local) operator can be constructed as a endomorphism from rho \otimes \rho. This is true since the Hilbert space is defined by fixing an array of rho's. We use (2.18) to compute the symmetry action on the generalized charges. We have added a forward reference to the equations where eq 2.18 was used.

  6. No, the lattice would not disappear as such since it describes an interface between two distinct boundaries along with a configuration of lines ending on this interface. We note that while the rho's themselves are comprised of topological lines, their ends however are not generically topological.

  7. Thanks, this has now been fixed.

  8. We have changed the phrase "Being dual to" to "Since P is dual to the Z2b symmetry being gauged,"

  9. Thanks for pointing this out. There was actually a typo in this equation which is now fixed. We also added a crossref to eq 2.52 which is used in deriving this.

  10. We have added to references above this equation. The Lagrangian algebra mentioned in the referees question is a specific example of a Lagrangian algebra that corresponds to the Dirichlet boundary condition. In general there are others.

  11. We added a reference to App. B of SciPost Phys. 18, 032 (2025) which clarifies these details.

  12. Thanks for pointing this out. It was a typo that has now been fixed \vec{g},g should be \vec{p},\vec{q}.

  13. In (2.89), we describe a general operator constructed from a string diagram in Vec(S_3) (i.e., the input fusion category in the construction of the lattice model). Delta and m have been explicitly defined above 2.89. When constructing Hamiltonians for gapped phases one invokes Frobenius algebras in Vec(G) which have the product and co-product as part of their definition. This is described (along with references) in Sec. 3 of our paper.

  14. Yes it is an oversight which has now been fixed.

  15. Here we are labeling boundary conditions by their Lagrangian algebras. The boundary condition 1\oplus e therefore condenses the e line and the m line projects onto this boundary to become the generator of the non-anomalous Z2 symmetry. Therefore this boundary condition is perhaps more natural for the reasons mentioned by the referee.

  16. The product and coproduct structure on A=1 is trivial while on A=Z2 is given be the description given above eq.(2.89). When rho=A, the Hilbert space obtained is directly that of the IR TQFT describing the corresponding gapped phase. For example when rho=A=1, we are in the symmetry breaking phase and as correctly pointed out by the referee, there are two states in the Hilbert space that are the symmetry breaking vacua.

  17. Done.

  18. Yes \Prod_j\sum_{h_j\in H} = \sum_{h_1,\dots,h_L}.

  19. We changed the sentence to "The defining property of SPTs is that their ground states symmetry twisted boundary conditions transform in non-trivial representations of the global symmetry."

  20. We intended for Sec 4.1 to serve as a description of the general structure, goal etc. In Sec 4.2 we provide the initial data required for the Z4--> Z2 transition between eq.(4.19) and (4.24).

Response to Referee 3:

We thank the referee for their positive evaluation of our work and for recommending publication. Below we respond to their comments and questions.

  1. In the abstract we already state: “The general theory of such phases in (1+1)d has been studied using the Symmetry Topological Field Theory (SymTFT), also known as topological holography. This has unearthed the infrared (IR) structure of these phases and transitions. In this paper, we describe how the SymTFT information can be converted into an ultraviolet (UV) anyonic chain lattice model realizing, in the IR limit, these phases and transitions.” This makes explicit reference to both (1+1)d and to “chains,” which refer specifically to (1+1)d models.

  2. Given the length of the paper, we aimed to keep the introduction concise. The mentioned paragraphs were intended only as a broad overview of the contents, while a detailed discussion of the SymTFT configurations and how they translate to lattice configurations is provided in Sec. 2.

  3. We thank the referee for this question. Let us clarify that the paper already contains Abelian, non-Abelian, and non-invertible examples. We begin with Abelian cases to illustrate the structure in a familiar setting, and then proceed to the more intricate example of Rep(S3), which is non-invertible. Further details on S3, the simplest non-Abelian case, are provided in the appendix. This example serves to highlight the key features of our construction in the simplest such setting.

  4. We added the following sentence in the Outlook paragraph--"In this paper we focus on models defined on the circle with periodic or twisted boundary conditions, while the study of open chains with boundary conditions is deferred to future work."

---

## Editorial Decision

in_refereeing